# Towards the Explainability of Temporal Graph Networks via Memory Backtracking and Topological Attribution

Yazheng Liu [1]   Xi Zhang [2]   Sihong Xie [* 1]   Hui Xiong [1]

## Abstract

Temporal graphs are ubiquitous in real-world applications and Temporal Graph Networks (TGNs) have achieved superior predictive accuracy. Understanding which historical events drive model predictions can enhance trustworthiness of TGNs. Existing explanation methods overlook the memory module, the core component that records and updates node histories, leaving the influence of past events unexplored. To address this, we attribute TGNs predictions through the topology attribution tree and memory backtracking tree. The topology attribution tree captures the influence of neighbors and their memory vectors, then the memory backtracking tree quantifies how historical events shape node memory vectors. We apply the LRP in TGNs, ensuring that the total contribution of events equals the model's logits. Finally, top-k selection may be unfaithful due to the nonlinear mapping from logits to probabilities, we design optimization objectives to identify the important events. Experiments on nine temporal graph datasets, spanning node property prediction, link prediction tasks and graph classification tasks, show that our method provides faithful explanations and outperforms state-of-the-art baselines. The code is available at https://github.com/yazhengliu/MemExplainer.

## 1. Introduction

Temporal Graph Networks (TGNs) (Rossi et al., 2020) have gained increasing attention in real-world applications such as fraud detection (Kim et al., 2024) and healthcare forecasting (Hancox et al., 2024; Lin et al., 2025). TGNs take as input a temporal graph. A temporal graph can be represented as a sequence of timestamped events on the graph. Each event is represented as $e_k = (v, u, t_k)$, indicating that source node $v$ and destination node $u$ have an interaction event at timestamp $t_k$. In TGNs, each node maintains a memory vector to store its historical information. For example, in a social network, a node's memory vector might represent the history of its interactions with other users, while in a recommendation system, it could store the user's past preferences and activities.

TGNs process temporal graphs in two stages: the memory module and the embedding module. In the memory module (Figure 1 (b)), events are processed in batches to improve computational efficiency. Target node updates its memory based on the received message. This message includes the source node memory, the destination node memory, and the event feature. The updated memory is passed to subsequent batches for further processing. In the embedding module (Figure 1 (c)), each node $u$ obtains its neighboring events $\mathcal{N}_u[0, t] = \{e_k = (v, u, t_k)|e_k \in \mathcal{E}(t)\}$, where $t$ is the current time and $\mathcal{E}(t)$ is the set of events observed before timestamp. The memory vectors are used to generate these neighboring event embeddings. The event embedding also includes the source node memory, the destination node memory and the event feature. Each node aggregates these neighboring event embeddings to form its own node embedding. To predict the presence of an edge between two target nodes, the embeddings of these nodes are used for the prediction. The memory and embedding modules enable TGN to capture both structural and temporal dependencies within the temporal graph and enhance prediction accuracy.

Despite their strong predictive performance, TGNs remain black-box models. TGNs offer little transparency into how predictions depend on historical events. Enhancing explainability is critical to improving the trustworthiness of TGNs and ensuring their safe deployment in high-stakes areas such as fraud detection and healthcare. Various explanation methods on static graph neural networks have been proposed, such as GNNExplainer (Ying et al., 2019), PGExplainer (Luo et al., 2020), and FlowX (Gui et al., 2023). These methods typically identify a small subset of

[1]Artificial Intelligence Trust, The Hong Kong University of Science and Technology (Guangzhou) [2]Key Laboratory of Trustworthy Distributed Computing and Service (MoE), Beijing University of Posts and Telecommunications. Correspondence to: Sihong Xie <sihongxie@hkust-gz.edu.cn>.

*Proceedings of the 43rd International Conference on Machine Learning*, Seoul, South Korea. PMLR 306, 2026. Copyright 2026 by the author(s).

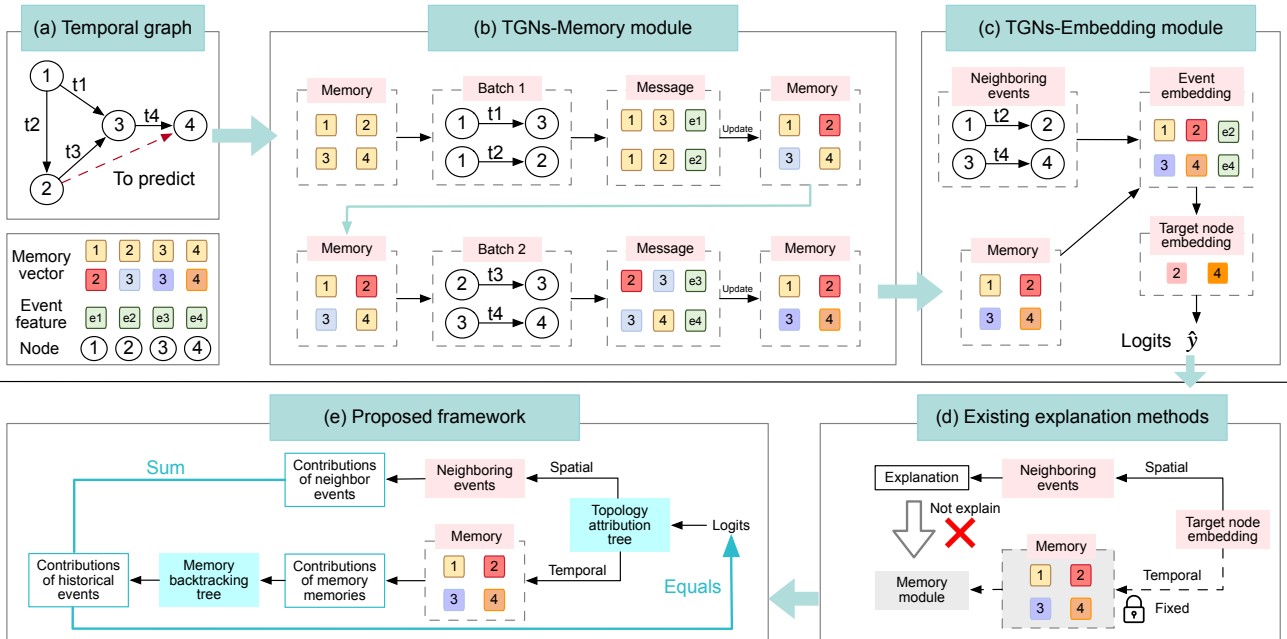

*Figure 1.* Overview of TGNs and our explainability framework. (**a**) Temporal graph: each node has a memory vector and each event has features. (**b**) The memory module in TGNs: events are processed in batches. In the batch 1, the node 3 and 2 update their memory, because they receive the message from event $e_1$ and $e_2$. For event $e_1 = (1, 3, t_1)$, the message is constructed from the memories of source node 1, destination node 3 and the feature of $e_1$. The updated memory is passed to batch 2 for further processing. (**c**) The embedding module in TGNs: for target nodes 2 and 4, the neighboring events are $e_2$ and $e_4$. The embedding of $e_4$ is based on memories of nodes 3 and 4, alongside the event feature. The event embeddings are aggregated to generate embeddings for node 4, which are used for predictions. (**d**) Existing explanation methods. Existing explanation methods fix the memories such that they ignore how historical events update the memory of node. As a result, they may obtain unfaithful explanations. (**e**) The proposed framework: we construct a topology attribution tree to quantify the spatial contribution of neighboring events and the temporal contribution of the node memories (Section 4.2). Then we construct memory backtracking tree to track the long term impact of historical events on node memories (Section 4.3). The total contributions of the neighboring events and historical events equals the logits.

important edges, nodes, or subgraphs that contribute the most to the prediction on static graphs. There have been some recent attempts at TGNs explainability, such as T-GNNExplainer (Xia et al., 2022) and TempME (Chen & Ying, 2023). However, all of the above methods fix the final memory vector (Figure 1 (d)). Fixing the memory vector means they ignore how historical events update the memory of node. In TGNs, the target node embeddings capture both temporal information from node memories and spatial information from neighboring events. Fixing the memory vector thus restricts the explanation method to the topological structure of the temporal graph, neglecting the critical role of the memory module. As a result, these explanation methods lead to inaccurate attributions and unfaithful explanations.

To address the challenge of temporal graph explanations, we propose a framework that attributes the predictions of TGNs to neighboring events and historical events, considering both spatial interactions among neighbors and the updates of node memories (Figure 1 (e)). First, we construct a topology attribution tree to quantify the spatial contribu-

tion of neighboring events and the temporal contribution of the node memories. Then we construct memory backtracking tree to track the long-term impact of historical events on the node memories. We apply Layer-wise Relevance Propagation (LRP) (Bach et al., 2015) method to the TGN model and ensure that the sum contributions of neighbor events and node memories equals the logits when constructing the topology attribution tree. In memory backtracking trees, the sum contributions of events equals the total memory contributions of all nodes. Thus, our method achieves conservation, where the sum of all event contributions equals the logits. This conservation property allows us to derive KL divergence and formulate an optimization problem to select important events as explanations. Experiments on nine temporal graph datasets, spanning node property prediction, link prediction, and graph classification demonstrate the effectiveness of our method, which consistently outperforms four state-of-the-art baselines, highlighting the advantage of tracing node memories. A t-test between our method and the second-best baseline shows statistical significance in **77**% of cases for Fidelity$_{\text{KL}}$ and **74**% for Fidelity$_{\text{prob}}$.

## 2. Related work

**GNNs Explainability** Explainability methods for Graph Neural Networks can be broadly categorized into instance-level and model-level approaches (Yuan et al., 2022). Instance level explanation methods focus on explaining model predictions by identifying important subgraphs that strongly correlate with predictions. For example, the Gradient/Features-based techniques, such as CAM and GradCAM (Pope et al., 2019) identify important nodes using gradients. Perturbation-based methods, such as GNNexplainer (Ying et al., 2019), PGExplainer (Luo et al., 2020), learn edge masks by maximizing mutual information to explain the predicted class distribution. Decomposition-based methods, such as GNN-LRP (Schnake et al., 2020), extend the original LRP (Bach et al., 2015) algorithm to GNNs and attribute the importance to graph walks. Surrogate-based methods, like GraphLime (Huang et al., 2020), build a surrogate model with kernel-based feature selection to provide node feature explanations. Model level explanation methods (Yuan et al., 2020) produce a high-level explanation about the general behaviors of GNNs. However, these methods are designed for static graphs and cannot explain temporal graph models. They fail to capture the temporal dependency mixed with the graph topology.

**TGNs Explainability** TGNNExplainer (Xia et al., 2022) is the first explainer tailored for TGNs, which relies on the MCTS algorithm to search for a combination of the explanatory events. TempME (Chen & Ying, 2023) extracts the most interaction-related motifs based on the information bottleneck principle. Recent work (He et al., 2022) utilizes the probabilistic graphical model to generate explanations for discrete time series on the graph, leaving the continuous-time setting underexplored. However, these methods overlook the memory module in TGNs, which is responsible for maintaining and updating node states over time. As a result, they fail to capture how historical interactions accumulate in memory and directly shape future predictions, leaving the core mechanism of TGNs unexplained.

## 3. Preliminaries on TGNs

A temporal graph is defined as a function of timestamp $t$, denoted by $\mathcal{G}(t) = \{\mathcal{V}(t), \mathcal{E}(t)\}$, where $\mathcal{V}(t)$ and $\mathcal{E}(t)$ represent the set of nodes and events observed before timestamp $t$. Each event $e_k = (v, u, t_k) \in \mathcal{E}(t)$ indicates an interaction between source node $v$ and destination node $u$ at timestamp $t_k$, where $t_k < t$. Let $\mathbf{x}_u \in \mathbb{R}^{1 \times d_m}$ and $\mathbf{x}_{e_k} \in \mathbb{R}^{1 \times d_e}$ denote feature vectors for node $u$ and event $e_k$, respectively. In TGNs, each node $u$ maintains a memory vector $\mathbf{s}_u^t \in \mathbb{R}^{1 \times d_m}$ at timestamp $t$. $\mathbf{s}_u^{t-} \in \mathbb{R}^{1 \times d_m}$ denotes the memory vector of node $u$ before timestamp $t$.

Temporal Graph Networks (Rossi et al., 2020) can be viewed as an encoder-decoder framework. The encoder maps the temporal graph $\mathcal{G}(t)$ into time-aware node embeddings, while the decoder takes one or more embeddings to obtain task-specific predictions, such as node property prediction or link prediction. The memory of node is updated whenever the node participates in an event. TGNs compute node embedding $\mathbf{z}_u^t \in \mathbb{R}^{d_m}$ in four steps:

$$\mathbf{m}_u^t = f_{\text{message}}\left(\mathbf{s}_v^{t-}, \mathbf{s}_u^{t-}, \mathbf{x}_{e_k}, t - t_k,\right), \qquad (1)$$

$$\bar{\mathbf{m}}_u^t = f_{\text{agg}}\left(\mathbf{m}_u^{t_1}, \ldots, \mathbf{m}_u^{t_b}\right), \qquad (2)$$

$$\mathbf{s}_u^t = f_{\text{update}}\left(\bar{\mathbf{m}}_u^t, \mathbf{s}_u^{t-}\right), \qquad (3)$$

$$\mathbf{z}_u^t = \sum_{e_k \in \mathcal{N}_u^n([0,t])} f_{\text{emb}}\left(\mathbf{s}_u^t, \mathbf{s}_v^t, \mathbf{x}_{e_k}, \mathbf{x}_u, \mathbf{x}_v\right). \qquad (4)$$

Here, the **message function** $f_{\text{message}}$ computes the message of destination node $u$ based on the memory $\mathbf{s}_u^{t-}$, $\mathbf{s}_v^{t-}$, the event features $\mathbf{x}_{e_k}$, and the elapsed time $t - t_k$. $f_{\text{message}}$ can be either identity map or MLP. If multiple messages are received in a batch, the **message aggregator** $f_{\text{agg}}$ combines these messages, either by selecting the most recent message or by averaging, and $t_1, \cdots, t_b \leq t$. The **memory updater** $f_{\text{update}}$ updates the node's memory, typically using an LSTM or GRU. Let $\mathcal{N}_u[0,t] = \{e_k | e_k = (v, u, t_k) \in \mathcal{E}(t)\}$ denote the neighborhood of node $u$ in time interval $t_k$ and $\mathcal{N}_u^n[0,t]$ denote the set of the most recent $n$ interactions from $\mathcal{N}_u[0,t]$, where $t$ is the current time. For each event, the **embedding** function $f_{\text{emb}}$ combines the memory vectors of the target node and its neighbors, along with the associated event and node features to produce an event-specific representation. $f_{\text{emb}}$ can be implemented as a temporal graph attention or a graph sum function.

The node embeddings are fed into an MLP for task-specific predictions. For link prediction, given two nodes $v$ and $u$ at time $t$, the prediction score is obtained as $\hat{y}_{v,u} = \sigma(f_{\text{mlp}}([\mathbf{z}_v^t || \mathbf{z}_u^t]))$, where $[\cdot \| \cdot]$ denotes vector concatenation, $f_{\text{mlp}}$ is the multi-layer perceptron, and $\sigma$ is the sigmoid function. For node property prediction, the embedding of a node $u$ is mapped to its label distribution as : $\hat{y}_u = \text{softmax}(f_{\text{mlp}}([\mathbf{z}_u^t]))$. For graph classification, the average pooling of $\mathbf{z}_u^t$ across all nodes in temporal graph $\mathcal{G}(t)$ produces a single vector representation for classification.

## 4. Method

### 4.1. Overview

The pipeline of our method is shown in Figure 1 (e), with the detailed process depicted in Figure 2. Taking link prediction as an example, given a temporal graph and a prediction between nodes $u$ and $v$ at time $t$, we proceed in three steps. Let $\mathcal{T}_{\text{top}}(x)$ denote the topology attribution tree and the $\mathbf{C}_x^t \in \mathbb{R}^{|\mathcal{E}(t)| \times d_m}$ represent the contributions of all events to $\mathbf{z}_x^t$, where node $x \in \{u, v\}$. The $\mathbf{M}_{p_0 \to u}^t \in \mathbb{R}^{d_m \times d_m}$

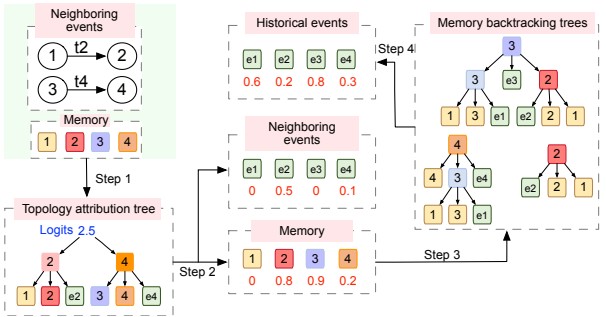

*Figure 2.* The detailed process of Figure 1 (e). Assuming the logits are 2.5, the numbers in red represent the contribution scores. The meaning of the different colored circles and squares is consistent with those in Figure 1. The number with red colors denote the contribution scores. The meanings of the different colored circles and squares are consistent with those in Figure 1. Given the node memories and neighboring events $e_2$ and $e_4$, we construct the topology attribution tree to attribute the logits to the memories of nodes 1, 2, 3, 4 and the features of neighboring events $e_2$ and $e_4$. For nodes 2, 3, 4, we construct the memory backtracking tree to attribute the contribution of node memories to the historical events $e_1$, $e_2$, $e_3$ and $e_4$. In both trees, the parent node is the target node in the event, with three child nodes: the source node, the target node, and the event triggering the interaction. The total contributions from historical and neighboring events equal the logits.

represents the contribution of node $p_0$'s memory vector to $z_u^t$. First, we construct $\mathcal{T}_{\text{top}}(u)$ and $\mathcal{T}_{\text{top}}(v)$ and attribute the $z_u^t$, $z_v^t$ to the historical events and to the node memory vectors (Section 4.2). We prove that these contribution values satisfy

$$\sum_{p_0 \in \text{leaf}(\mathcal{T}_{\text{top}}(x))} \mathbf{1}^\top \mathbf{M}_{p_0 \to x}^t + \mathbf{1}^\top \mathbf{C}_x^t = z_x^t, \quad x \in \{u, v\} \quad (5)$$

Where $\text{leaf}(\mathcal{T}_{\text{top}}(x))$ represent the leaf nodes of $\mathcal{T}_{\text{top}}(x)$ and $\mathbf{1}^\top$ represents the transpose of a vector of ones.

Next, we further decompose the contributions of memory vectors $\mathbf{M}_{p_0 \to u}^t$ to the contributions of events $\mathbf{C}_u^t$ through a memory backtracking tree, revealing which past interactions most strongly influence these vectors (Section 4.3). We also prove that the contribution values satisfy

$$\mathbf{1}^\top \mathbf{C}_u^t = z_u^t, \quad \mathbf{1}^\top \mathbf{C}_v^t = z_v^t \quad (6)$$

Finally, in the MLP function for link prediction, we compute the contribution of $z_u^t$ and $z_v^t$ to the final logits, update the $\mathbf{C}_u^t$ and $\mathbf{C}_v^t$, and obtain the total final contribution of events:

$$\mathbf{C}^t = \mathbf{C}_u^t + \mathbf{C}_v^t, \quad \mathbf{1}^\top \mathbf{C}^t = \text{logits} \quad (7)$$

Using this conservation property, we derive the KL divergence and design objective functions to select the important events that explain the TGNs prediction (Section 4.4).

**4.2. Spatial Decomposition via topology attribution tree**

We construct the topology attribution tree and apply the Layer-wise Relevance Propagation (LRP) method (Bach

et al., 2015) to Eq. (4), decomposing the target node embedding $z_u^t$ into contributions from the target memory vector $s_u^t$, the neighbor memory vector $s_v^t$, and the feature vectors $\mathbf{x}_u$, $\mathbf{x}_v$, and $\mathbf{x}_{e_k}$. Next, we introduce the original LRP method.

### 4.2.1. ORIGINAL LRP METHOD

LRP attributes the prediction score to the input neurons. Let the activation of a neuron at layer $l+1$ be $h^{l+1} \in \mathbb{R}$, computed as $h^{l+1} = f([h_1^l, \ldots, h_n^l])$, where $f$ may be a linear function or a composition of a linear function with a nonlinear activation such as ReLU. Given the relevance $R^{l+1} \in \mathbb{R}$, LRP distributes it $h^{l+1}$ to the inputs according to the following equation:

$$R_i^l = \frac{h_i^l w_i^l}{\sum_{i'} h_{i'}^l w_{i'}^l} R^{l+1}, \quad (8)$$

where $w_i^l$ is weight from the neuron $h_i^l$ to the neuron $h^{l+1}$.

For a multi-layer network, LRP redistributes the relevance score from the output layer back to the input layer by recursively applying the Eq. (8). Let the final prediction logit be assigned as relevance score $R^L$, where $L$ denotes the output layer. The relevance of an input neuron $h_i^0$ is then obtained by successively propagating through all intermediate layers, while preserving the conservation property: $\sum_i R_i^0 = R^L$

$$R_i^0 = \sum_{j,\ldots,k} \frac{h_i^0 w_i^0}{\sum_{i'} h_{i'}^0 w_{i'}^0} \cdots \frac{h_k^{L-1} w_k^{L-1}}{\sum_{k'} h_{k'}^{L-1} w_{k'}^{L-1}} \cdot R^{(L)}. \quad (9)$$

**Lemma 4.1.** *Let* $\mathbf{x} \in \mathbb{R}^{1 \times d_1}$, $\mathbf{W} \in \mathbb{R}^{d_1 \times d_2}$ *and* $\mathbf{y} = \mathbf{x} \cdot W \in \mathbb{R}^{1 \times d_2}$. *Suppose that the relevance contribution assigned to* $\mathbf{y}$ *is* $\mathbf{R}(\mathbf{y}) \in \mathbb{R}^{d_2 \times d_3}$. *Let* $\cdot$ *denote the matrix multiplication,* $\text{diag}(\cdot)$ *denote the diagonal matrix, and* $\text{diag}(\cdot)^{-1}$ *be the inverse of the diagonal matrix. Using the LRP method, the relevance contribution assigned to* $\mathbf{x}$ *is obtained from the following formula and satisfies the conservation property:*

$$\mathbf{P} = \text{diag}(\mathbf{x}) \cdot \mathbf{W} \cdot \text{diag}(\mathbf{y})^{-1}, \quad \mathbf{R}(\mathbf{x}) = \mathbf{P} \cdot \mathbf{R}(\mathbf{y}),$$
$$\mathbf{1}^\top \cdot \mathbf{R}(\mathbf{x}) = \mathbf{1}^\top \cdot \mathbf{R}(\mathbf{y}).$$

The proof of Lemma 4.1 is in Appendix A.2. Lemma 4.1 represents the matrix form of LRP. The matrix $\mathbf{P}$ can be interpreted as a proportional matrix. The core of the LRP algorithm is to design rules to compute this proportional matrix $\mathbf{P}$, ensuring that it satisfies the conservation property $\mathbf{1}^\top \cdot \mathbf{P} = \mathbf{1}^\top$. Consequently, we have $\mathbf{1}^\top \cdot \mathbf{R}(\mathbf{x}) = \mathbf{1}^\top \cdot \mathbf{P} \cdot \mathbf{R}(\mathbf{y}) = \mathbf{1}^\top \cdot \mathbf{R}(\mathbf{y})$. We will now proceed to derive the explicit expression for the proportional matrix $\mathbf{P}$ when the LRP method is applied to $f_{\text{emb}}$.

### 4.2.2. LRP ON $f_{\text{emb}}$

The $\mathbf{h}_u^{t,l}$ denotes the node embedding for $u$ in graph layer $l$ and timestamp $t$, with $\mathbf{h}_u^{t,0} = s_u^t$. For simplicity, we drop

the superscript $t$ and write $\mathbf{h}_u^l \triangleq \mathbf{h}_u^{t,l}$. We use the temporal graph sum function as an example to illustrate how LRP attributes the logits of TGNs to $\mathbf{s}_u^t$, $\mathbf{s}_v^t$, $\mathbf{x}_u$, $\mathbf{x}_v$, and $\mathbf{x}_{e_k}$. The derivation for the graph attention function is in Appendix A.4, with the resulting formulas in Eq. (26). For the graph sum function, Eq. (4) becomes:

$$\hat{\mathbf{h}}_u^l = \sum_{e_k=(v,u,t_k)\in\mathcal{N}_u^n} \left( \mathbf{h}_v^{l-1} \| \mathbf{x}_{e_k} \| \phi(t-t_k) \right) \mathbf{W}_1^l$$

$$\tilde{\mathbf{h}}_u^l = \mathrm{ReLU}(\hat{\mathbf{h}}_u^l), \quad \mathbf{h}_u^l = \left( \mathbf{h}_u^{l-1} \| \tilde{\mathbf{h}}_u^l \right) \mathbf{W}_2^l. \quad (10)$$

The function $\phi(\cdot)$ is a time encoding. Let $d_t$ denote the output dimension of $\phi(\cdot)$. $\mathbf{W}_2 = [\mathbf{A}^l \quad \mathbf{B}^l] \in \mathbb{R}^{d_m \times 2d_m}$ and $\mathbf{W}_1^l = [\mathbf{C}^l \quad \mathbf{D}^l \quad \mathbf{E}^l] \in \mathbb{R}^{d_m \times (d_m + d_e + d_t)}$ are parameters, where $\mathbf{A}^l, \mathbf{B}^l, \mathbf{C}^l \in \mathbb{R}^{d_m \times d_m}$, $\mathbf{D}^l \in \mathbb{R}^{d_e \times d_m}$ and $\mathbf{E}^l \in \mathbb{R}^{d_t \times d_m}$. $(\cdot \| \cdot \| \cdot)$ denotes vector concatenation. The final embedding is $\mathbf{z}_u^t = \mathbf{h}_u^L \in \mathbb{R}^{1 \times d_m}$, where $L$ is the number of layers.

**Proposition 4.2.** *When the LRP method is applied to $f_{\mathrm{emb}}$, the proportional matrices are*

$$\mathbf{P}(\tilde{u}) = \mathrm{diag}\left( \tilde{\mathbf{h}}_u^l \right) \cdot \mathbf{B}^l \cdot \mathrm{diag}\left( \mathbf{h}_u^l \right)^{-1}$$

$$\mathbf{P}(v) = \mathrm{diag}\left( \mathbf{h}_v^{l-1} \right) \cdot \mathbf{C}^l \cdot \mathrm{diag}\left( \hat{\mathbf{h}}_u^l \right)^{-1} \cdot \mathbf{P}(\tilde{u})$$

$$\mathbf{P}(e_k) = \mathrm{diag}(\mathbf{x}_{e_k}) \cdot \mathbf{D}^l \cdot \mathrm{diag}\left( \hat{\mathbf{h}}_u^l \right)^{-1} \cdot \mathbf{P}(\tilde{u})$$

$$\mathbf{P}(\phi) = \mathrm{diag}(\phi(t-t_k)) \cdot \mathbf{E}^l \cdot \mathrm{diag}\left( \hat{\mathbf{h}}_u^l \right)^{-1} \cdot \mathbf{P}(\tilde{u})$$

$$\mathbf{P}(u) = \mathrm{diag}\left( \mathbf{h}_u^{l-1} \right) \cdot \mathbf{A}^l \cdot \mathrm{diag}\left( \mathbf{h}_u^l \right)^{-1} \quad (11)$$

*Let $\mathbf{R} = \mathbf{R}\left( \mathbf{h}_u^l \right)$, the contribution matrices are given by*

$$\mathbf{R}\left( \mathbf{h}_v^{l-1} \right) = \mathbf{P}(v) \cdot \mathbf{R}, \quad \mathbf{R}\left( \mathbf{h}_u^{l-1} \right) = \mathbf{P}(u) \cdot \mathbf{R}$$
$$\mathbf{R} = \mathbf{P}(e_k) \cdot \mathbf{R}, \quad \mathbf{R}\left( t-t_k \right) = \mathbf{P}(\phi) \cdot \mathbf{R} \quad (12)$$

*For $e_k = (v, u, t_k) \in \mathcal{N}_u^n$, these contribution matrices satisfies*

$$\mathbf{1}^\top \mathbf{R}\left( \mathbf{h}_u^l \right) = \mathbf{1}^\top \mathbf{R}\left( \mathbf{h}_u^{l-1} \right) + \mathbf{1}^\top \sum_v \mathbf{R}\left( \mathbf{h}_v^{l-1} \right) \quad (13)$$

$$+ \mathbf{1}^\top \sum_{e_k} \mathbf{R}\left( e_k \right) + \mathbf{1}^\top \sum_{t_k} \mathbf{R}\left( t-t_k \right)$$

The contribution matrices $\mathbf{R}\left( \mathbf{h}_v^{l-1} \right)$ and $\mathbf{R}\left( \mathbf{h}_u^{l-1} \right) \in \mathbb{R}^{d_m \times d_m}$, $\mathbf{R}(\mathbf{x}_{e_k}) \in \mathbb{R}^{d_e \times d_m}$ and $\mathbf{R}(t-t_k) \in \mathbb{R}^{d_t \times d_m}$. Each matrix element represents the contribution of the $i$-th neuron in $\mathbf{h}_v^{l-1}$, $\mathbf{h}_u^{l-1}$, $\mathbf{x}_{e_k}$, $\phi(t-t_k)$ to the $j$-th neuron in $\mathbf{z}_u^t$, respectively. The proof is in Appendix A.3. The example of the Proposition 4.2 is in Figure 3.

### 4.2.3. TOPOLOGY ATTRIBUTION TREE

We construct the topology attribution tree through hierarchical traversal, updating the contribution matrix at each

*Figure 3.* We obtain these proportional matrices according to the Proposition 4.2. Summing their elements column-wise results in a vector of ones. Based on this property, the total contribution of $\mathbf{R}\left( \mathbf{h}_v^{l-1} \right)$, $\mathbf{R}\left( \mathbf{h}_u^{l-1} \right)$, $\mathbf{R}(\mathbf{x}_{e_k})$ and $\mathbf{R}(t-t_k)$ equals to $\mathbf{R}\left( \mathbf{h}_u^l \right)$.

layer of the tree based on Eq. (12) and Eq. (26). In this process we attribute contributions to both the events and the node memories, as described in Algorithm 2. As outlined in Algorithm 2, we can compute the contribution matrix $\mathbf{C}_u^t$, where each row corresponds to an event, and the row vector represents the contribution to $\mathbf{z}_u^t$. Additionally, we obtain a set of matrices $\{\mathbf{M}_{p_0 \to u}^t | p_0$ is a leaf node of $\mathcal{T}_{\mathrm{topo}}(u)\}$. Here, $\mathbf{M}_{p_0 \to u}^t \in \mathbb{R}^{d_m \times d_m}$ denotes the contribution of node $p_0$'s memory vector to $\mathbf{z}_u^t$. By the conservation property (Proposition 4.2), the contribution values satisfy: $\sum_{p_0} \mathbf{1}^\top \mathbf{M}_{p_0 \to u}^t + \mathbf{1}^\top \mathbf{C}_u^t = \mathbf{z}_u^t$.

### 4.3. Temporal Decomposition via Memory Backtracking

In Algorithm 2, we obtain a set of the memory matrices $\{\mathbf{M}_{p_0 \to u}^t | p_0$ is a leaf node of $\mathcal{T}_{\mathrm{top}}(u)\}$ and the event contribution matrix $\mathbf{C}_u^t$. For each $\mathbf{M}_{p_0 \to u}^t$, we construct a memory backtracking tree to attribute the contributions of node memory to the historical events responsible for updating the target node. In this process, we will update the $\mathbf{C}_u^t$. After the memory backtracking process, the contribution matrix satisfies $\mathbf{1}^\top \mathbf{C}_u^t = \mathbf{z}_u^t$. We use $\mathbf{M}_{u \to u}^t$ as an example to demonstrate how to construct the memory backtracking tree and attribute the contributions to the historical events.

#### 4.3.1. LRP ON $f_{\mathrm{update}}$

To illustrate the backtracking construction, we set $f_{\mathrm{update}}$ as a GRU. If $f_{\mathrm{update}}$ is RNN function, the derivation is in Appendix A.6, with the resulting formulas in Eq. (30). For $f_{\mathrm{update}}$ as a GRU, Eq. (3) becomes:

$$\mathbf{r}_u^t = \sigma\left( \bar{\mathbf{m}}_u^t \mathbf{W}_r + \mathbf{s}_u^{t-} \mathbf{U}_r + \mathbf{b}_r \right),$$
$$\mathbf{g}_u^t = \sigma\left( \bar{\mathbf{m}}_u^t \mathbf{W}_g + \mathbf{s}_u^{t-} \mathbf{U}_g + \mathbf{b}_g \right),$$
$$\tilde{\mathbf{s}}_u^t = \tanh\left( \bar{\mathbf{m}}_u^t \mathbf{W}_h + \mathbf{r}_u^t \odot (\mathbf{s}_u^{t-} \mathbf{U}_h) + \mathbf{b}_h \right),$$
$$\mathbf{s}_u^t = (1 - \mathbf{g}_u^t) \odot \mathbf{s}_u^{t-} + \mathbf{g}_u^t \odot \tilde{\mathbf{s}}_u^t.$$

Where $\mathbf{W}_r, \mathbf{W}_g, \mathbf{W}_h, \mathbf{U}_r, \mathbf{U}_g, \mathbf{U}_h, \mathbf{b}_r, \mathbf{b}_g, \mathbf{b}_h$ are the parameters in the GRU.

**Proposition 4.3.** *Let* $\mathrm{diag}$ *denote the diagonalization op-*

*erator that maps a vector to a diagonal matrix. Let $\mathbf{L}^t = \text{diag}(\bar{\mathbf{m}}_u^t) \cdot \mathbf{W}_h \cdot \text{diag}(\tilde{\mathbf{s}}_u^t - \mathbf{b}_h)^{-1}$, $\mathbf{F}^t = \text{diag}\left(\frac{(1-\mathbf{g}_u^t)\odot \mathbf{s}_u^{t-}}{\mathbf{s}_u^t}\right)$, and $\mathbf{H}^t = \text{diag}(\mathbf{r}_u^t \odot \mathbf{s}_u^{t-}) \cdot \mathbf{U}_h \cdot \text{diag}(\bar{\mathbf{m}}_u^t \mathbf{W}_h + \mathbf{r}_u^t \odot (\mathbf{s}_u^{t-} \mathbf{U}_h))^{-1}$. When the LRP method is applied to $f_{\text{update}}$, the proportional matrices are*

$$\mathbf{P}\left(\mathbf{s}_u^{t-}\right) = \mathbf{F}^t + \mathbf{H}^t \cdot \text{diag}\left(\frac{\mathbf{g}_u^t \odot \tilde{\mathbf{s}}_u^t}{\mathbf{s}_u^t}\right),$$

$$\mathbf{P}\left(\bar{\mathbf{m}}_u^t\right) = \mathbf{L}^t \cdot \text{diag}\left(\frac{\mathbf{g}_u^t \odot \tilde{\mathbf{s}}_u^t}{\mathbf{s}_u^t}\right),$$

$$\mathbf{1}^\top = \mathbf{1}^\top \mathbf{P}\left(\mathbf{s}_u^{t-}\right) + \mathbf{1}^\top \mathbf{P}\left(\bar{\mathbf{m}}_u^t\right).$$

*Then, the contribution matrices are given by*

$$\mathbf{R}\left(\mathbf{s}_u^{t-}\right) = \mathbf{P}\left(\mathbf{s}_u^{t-}\right) \cdot \mathbf{R}\left(\mathbf{s}_u^t\right),$$

$$\mathbf{R}\left(\bar{\mathbf{m}}_u^t\right) = \mathbf{P}\left(\bar{\mathbf{m}}_u^t\right) \cdot \mathbf{R}\left(\mathbf{s}_u^t\right),$$

$$\mathbf{1}^\top \mathbf{R}\left(\mathbf{s}_u^t\right) = \mathbf{1}^\top \mathbf{R}\left(\mathbf{s}_u^{t-}\right) + \mathbf{1}^\top \mathbf{R}\left(\bar{\mathbf{m}}_u^t\right).$$

*Then we decompose $\mathbf{R}(\bar{\mathbf{m}}_u^t)$ as*

$$\mathbf{R}\left(\bar{\mathbf{m}}_u^t\right) = [\mathbf{R}\left(\mathbf{s}_v^{t-}\right), \mathbf{R}\left(\mathbf{s}_u^{t-}\right), \mathbf{R}(\mathbf{x}_{e_k}), \mathbf{R}(t - t_k)].$$

Where $\mathbf{R}(\mathbf{s}_u^{t-})$ and $\mathbf{R}(\mathbf{s}_v^{t-}) \in \mathbb{R}^{d_m \times d_m}$, $\mathbf{R}(\mathbf{x}_{e_k}) \in \mathbb{R}^{d_e \times d_m}$, and $\mathbf{R}(t - t_k) \in \mathbb{R}^{d_t \times d_m}$ denote the contribution of $\mathbf{s}_u^{t-}$, $\mathbf{s}_v^{t-}$, $\mathbf{x}_{e_k}$ and $\phi(t - t_k)$ to $\mathbf{z}_u^t$, respectively. The proof is in Appendix A.5.

The overview of proposition 4.3 is provided in Figure 7. we first decompose $\mathbf{R}(\mathbf{s}_u^t)$ into $\mathbf{R}(\mathbf{s}_u^{t-})$ and $\mathbf{R}(\tilde{\mathbf{s}}_u^t)$. Then, $\mathbf{R}(\tilde{\mathbf{s}}_u^t)$ is further decomposed into $\mathbf{R}(\mathbf{s}_u^{t-})$ and $\mathbf{R}(\bar{\mathbf{m}}_u^t)$. Finally, $\mathbf{R}(\bar{\mathbf{m}}_u^t)$ is attributed to $\mathbf{R}(\mathbf{s}_v^{t-})$, $\mathbf{R}(\mathbf{s}_u^{t-})$, $\mathbf{R}(t - t_k)$, and $\mathbf{R}(\mathbf{x}_{e_k})$. The toy running example is in Appendix A.9.

### 4.3.2. MEMORY BACKTRACKING TREE

The memory backtracking tree is organized chronologically: the root corresponds to the target node's memory at the prediction time, the first layer contains the most recent events that directly updated this memory, and deeper layers trace earlier historical events. According to the proposition 4.3, the contribution of each parent node in the tree can be recursively attributed to its child nodes.

Figure 4 presents an example of constructing memory backtracking trees. In Figure 4 (b), events are processed in batches, and memory updates are carried forward to subsequent batches. Figure 4 (c) shows the memory update process of nodes 2 and 3. For node 3, its memory is updated from $\mathbf{s}_3^{t_0}$ after receiving the message from event $e_1$, remains unchanged at $t_2$, is updated again by event $e_3$ to $\mathbf{s}_3^{t_3}$, and stays unchanged at $t_4$. Node 2 follows a similar process.

Figure 4 (d) shows the memory backtracking tree for node 3. Starting from the $\mathbf{R}(\mathbf{s}_3^{t_4})$, we first trace it back to the most recent event $e_3$, and obtain the tree in Figure 4 (d) (1). The

$\mathbf{R}(\mathbf{s}_3^{t_2})$, $\mathbf{R}(\mathbf{s}_2^{t_2})$ and $\mathbf{R}(e_3)$ can be obtained according to the proposition 4.3. Then $\mathbf{R}(\mathbf{s}_3^{t_2})$ is further traced back to event $e_1$ as shown in Figure 4 (d) (2), while $\mathbf{R}(\mathbf{s}_2^{t_2})$ is traced back to event $e_2$, as shown in Figure 4 (d) (3). By recursively expanding the tree in this way, we obtain the contributions of all relevant historical events.

To trace contribution of node memory vectors, we record which events triggered updates. Let $\mathcal{H}(u, t) = \{ e_k = (v, u, t_k) \in \mathcal{E}(t) : t_k < t \land \text{Update}(u, t_k) \}$, where $\text{Update}(u, t_k)$ is true if and only if $u$'s memory is updated at time $t_k$. For example, in Figure 4 (b), the memory of node 3 is updated by events $e_1$ and $e_3$, thus, the $\mathcal{H}(3, t_4) = \{e_1 = (1, 3, t_1), e_3 = (2, 3, t_3)\}$. Based on the $\mathcal{H}(u, t)$, we construct memory backtracking trees and assign event contributions. The recursive procedure is given in Algorithm 5, where the maximum depth $T_L$ controls how far the backtracking proceeds: larger $T_L$ allows deeper tracing into historical events.

### 4.4. Selecting Important Events

We derive the KL divergence and formulate an optimization problem to select important events that faithfully explain the model's behavior. For the link prediction task, the predicted probability is $\hat{y}_{v,u} = \sigma(f_{\text{mlp}}([\mathbf{z}_v^t || \mathbf{z}_u^t]))$. Using the Algorithm 2 and Algorithm 5, we can compute the event contributions $\mathbf{C}_u^t$ and $\mathbf{C}_v^t$. Then, for the MLP function used in link prediction, we compute the contributions according to Lemma 4.1 and update $\mathbf{C}_u^t$ and $\mathbf{C}_v^t$. Due to the conservation property of Proposition 4.2 and Proposition 4.3, we have $\mathbf{1}^\top \mathbf{C}_u^t + \mathbf{1}^\top \mathbf{C}_v^t = f_{\text{mlp}}([\mathbf{z}_v^t || \mathbf{z}_u^t]) = y_{v,u}$. The final contribution matrix is $\mathbf{C}^t = \mathbf{C}_u^t + \mathbf{C}_v^t$.

For the link prediction, let $p_1$ and $p_2$ denote arbitrary prediction probabilities. The KL divergence between $p_1$ and $p_2$ is defined as:

$$\text{KL}(p_2 \| p_1) = \sigma(z_2)(z_2 - z_1) - \text{sp}(z_2) + \text{sp}(z_1), \quad (14)$$

where $\text{sp}(z) = \frac{1}{1+e^{-z}}$, $\sigma$ is the sigmoid function. The derivation process is in Appendix A.7.

Let $\mathcal{E}^* \subset \mathcal{E}(t)$ be the selected important events, and we aim to minimize the KL divergence between $\hat{y}_{v,u}(\mathcal{E}(t))$ and $\hat{y}_{v,u}(\mathcal{E}^*)$. Let $d_c$ denote the number of events in $\mathbf{C}^t$ with non-zero contributions, and $\mathbf{d} \in \{0, 1\}^{d_c}$ be the selection vector, where the element $d_l$ in $\mathbf{d}$ indicates whether the $l$-th event $e_l$ is selected. We define the following problem:

$$\mathbf{d}^* = \underset{\substack{\mathbf{d} \in \{0,1\}^{d_c}, \\ \|\mathbf{d}\|_1 = n}}{\arg\min} \quad -\hat{y}_{v,u}\left(\mathcal{E}(t)\right) \sum_{l=1}^{d_c} d_l \mathbf{C}_{e_l}^t + \text{sp}(\sum_{l=1}^{d_c} d_l \mathbf{C}_{e_l}^t)$$

$$(15)$$

The derivation process is in Appendix A.7. By solving Eq. (15), we can obtain the most important events for the link prediction task. The Algorithm 1 shows the overall

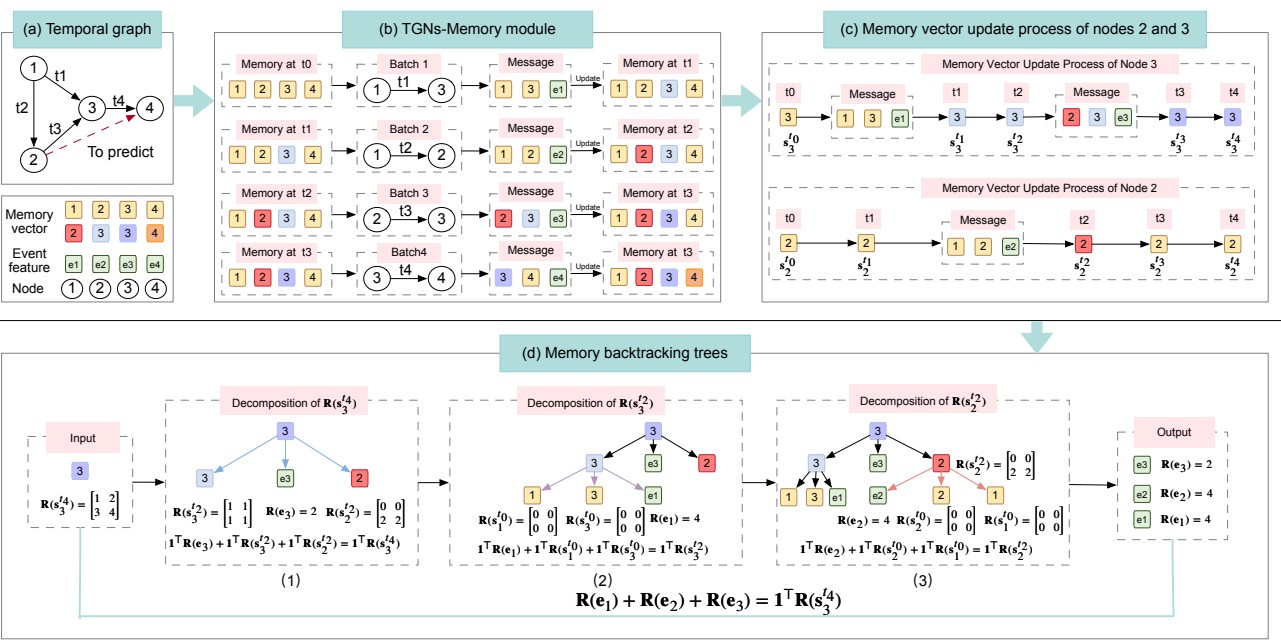

*Figure 4.* The example of building a memory backtracking tree.

process of selecting important events for link prediction task. The node property prediction is in Appendix A.12 and Algorithm 3, while the graph classification task is in Appendix A.13 and Algorithm 4.

---

**Algorithm 1** The overall framework for link prediction

---

1: **Input**: target interaction $(v, u)$, time $t$, and the max depth of memory backtracking tree $T_L$
2: Compute $\{\mathbf{M}^t_{p_0 \to u} | p_0$ is a leaf node of $\mathcal{T}_{\text{top}}(u)\}$, $\{\mathbf{M}^t_{p_0 \to v} | p_0$ is a leaf node of $\mathcal{T}_{\text{top}}(v)\}$, $\mathbf{C}^t_u$, and $\mathbf{C}^t_v$ using Algorithm 2.      ▷ topology attribution
3: **for** each $\mathbf{M}^t_{p \to u} \in \{\mathbf{M}^t_{p_0 \to u}\}$ **do**
4:     Update $\mathbf{C}^t_u$ using Algorithm 5 ▷ Memory attribution
5: **end for**
6: **for** each $\mathbf{M}^t_{p \to v} \in \{\mathbf{M}^t_{p_0 \to v}\}$ **do**
7:     Update $\mathbf{C}^t_v$ using Algorithm 5 ▷ Memory attribution
8: **end for**
9: Update $\mathbf{C}^t_u$ and $\mathbf{C}^t_v$ using Lemma 4.1, and $\mathbf{C}^t = \mathbf{C}^t_u + \mathbf{C}^t_v$
10: Select important events $\mathcal{E}^*$ using Eq. (15)
11: **Output:** The important events $\mathcal{E}^*$

---

### 4.5. Complexity Analysis

**Construct the topology attribution tree**: if $f_{\text{emb}}$ is the graph sum function, the complexity is $\mathcal{O}(N \cdot d_m + 2^L \cdot B \cdot n \cdot d_m)$. If the $f_{\text{emb}}$ is the graph attention function, the complexity is $\mathcal{O}(N \cdot d_m + 2^L \cdot B \cdot (n + n^2) \cdot d_m)$.
**Construct the memory backtracking tree**: the complexity

is $\mathcal{O}(2^{T_L} \cdot T_L \cdot d_m \cdot d_m)$. **Select the important events**: the complexity is $\mathcal{O}(d_c^3)$. Where $d_m$ is the dimension of the memory vector, $n$ is the number of most recent interactions considered, $L$ is the number of graph layers, $N = |\mathcal{V}(t)|$ is the number of nodes, $B$ is the batch size, $T_L$ is the depth of memory backtracking tree, and $d_c$ is the number of events in contribution matrix $\mathbf{C}^t$

## 5. Experiments

**Datasets**. We evaluate the effectiveness of our method, MemExplainer, on nine real-world temporal graph datasets: Wikipedia, Reddit, Enron and UCI (Hamilton et al., 2017; Jure, 2014; Poursafaei et al., 2022) for the link prediction task, tgbn-trade, tgbn-genre, and tgbn-reddit (Huang et al., 2023) for node property prediction, HMDB51 dataset (Kuehne et al., 2011) and Penn Action (Zhang et al., 2013) dataset for graph classification (human pose-based action classification) task. Detailed dataset statistics are provided in Appendix A.15.

**Baselines**. We compare the performance of MemExplainer with several baselines: **TGNNExplainer** (Xia et al., 2022), **TempME** (Chen & Ying, 2023), **GNNExplainer** (Ying et al., 2019), and **PGExplainer** (Luo et al., 2020). The GN-NExplainer and PGExplainer are the explanation methods for static graph models. We adapt it for TGNs following the same setting in prior work (Xia et al., 2022). TempME introduces a graph generation approach to capture temporal motifs for TGN predictions, while TGNNExplainer combines

a navigator with Monte Carlo Tree Search, guiding the sampling process to construct explanation subgraphs. We also include several ablation variants as baselines. **w/o memory** uses only neighboring-event contributions (no memory backtracking tree); **w/o topology** uses only historical-event contributions (no topology tree); **w/o selection** skips objective-based selection and directly returns the top-$k$ events by accumulated contribution.

**Experimental setup**. In the graph classification (pose-based action classification) task, for each video, we first run YOLOPose (Maji et al., 2022) to detect human keypoints and construct a skeleton graph whose nodes are body joints and whose edges follow the human kinematic structure. This produces a sequence of skeleton graphs for each frame, and then we feed these skeleton graphs into TGN to predict the action label. Notably, in Penn Action dataset, we use labeled data instead of extracting it using YOLOPose. We set the maximum depth of the memory backtracking tree to 5. Given the target events, nodes or graphs, we apply Algorithm 1, Algorithm 3 or Algorithm 4 to identify the important events $\mathcal{E}^*$.

**Evaluation metrics** We evaluate performance using fidelity and sparsity. Let $\mathcal{E}(t)$ represent the original temporal events. Let $\mathcal{E}^*$ denote the selected important events. The KL-based fidelity metrics are $\text{Fidelity}_{\text{KL}} = \text{KL}(\hat{y}_{v,u}(\mathcal{E}(t)) \| \hat{y}_{v,u}(\mathcal{E}^*))$ (link prediction) as defined in prior work (Liu et al., 2024). We use the probability-based fidelity metric: $\text{Fidelity}_{\text{prob}} = |\hat{y}_{v,u}(\mathcal{E}(t)) - \hat{y}_{v,u}(\mathcal{E}^*)|$ (link prediction) as defined in (Yuan et al., 2022). Lower values indicate better performance. Sparsity $= \frac{|\mathcal{E}^*|}{|\mathcal{E}(t)|}$, where $|\mathcal{E}^*|$ and $|\mathcal{E}(t)|$ denote the number of events in $\mathcal{E}^*$ and $\mathcal{E}(t)$. The metrics for node property prediction and graph classification are shown in Appendix A.11.

**Performance**. We report the $\text{Fidelity}_{\text{KL}}$ and the $\text{Fidelity}_{\text{prob}}$ results in Figures 5-6, respectively, when the $f_{\text{emb}}$ is graph sum function. Appendix A.17 provides the mean and standard deviation of these metrics in Tables 3 and 4. A t-test between our method and the second-best baseline shows statistical significance in **77%** of cases for $\text{Fidelity}_{\text{KL}}$ and **74%** for $\text{Fidelity}_{\text{prob}}$. In Figures 5-6, our method, MemExplainer, outperforms all baseline explainers across all datasets for both metrics, highlighting its ability to maintain high fidelity even with low sparsity. TempME can only capture temporal motifs and does not trace node memory vectors. TGNNExplainer relies on sampling; when many events are present, it only considers the sampled candidates, and if these have little influence, the resulting explanations have low fidelity. The performances of GNNExplainer and PGExplainer are poorer, as these methods are designed for static graphs and are not suitable for TGNs.

**Running time**. We partition the computation time into three components: **topology time**, for constructing the topology

attribution tree and computing neighbor contributions; **memory time**, for constructing the memory backtracking tree and computing contributions of historical events; and **selection time**, for solving Eq. (15) or Eq. (33). In Appendix A.16, Figure 8 reports the average computation time on all datasets. When $T_L = 10$, the total running time takes at most 6 seconds, which is still acceptable. The details of running time is provided in Appendix A.16.

**The performance of different $f_{\text{emb}}$ and $f_{\text{update}}$ in TGNs.** In Figure 9 and Figure 10, we show the performance of $\text{Fidelity}_{\text{prob}}$ and $\text{Fidelity}_{\text{KL}}$ when the $f_{\text{emb}}$ is graph attention model. In Figure 11 and Figure 12, we report $\text{Fidelity}_{\text{prob}}$ and $\text{Fidelity}_{\text{KL}}$ for the setting where the memory updater $f_{\text{update}}$ is implemented as an RNN. Our method, MemExplainer, significantly outperforms baseline explainers across all datasets for both metrics.

**Sensitivity analysis**. In Figure 13 and Figure 14, we report $\text{Fidelity}_{\text{prob}}$ and $\text{Fidelity}_{\text{KL}}$ when the number of events from neighboring samples is fixed to $n = 20$. Our method, MemExplainer, achieve the best performance on both metrics across all datasets. We also perform a sensitivity analysis on the maximum depth of the memory backtracking tree, varying the depth from 2 to 10, while maintaining the same number of events for each target node or edge across all depths. The results for $\text{Fidelity}_{\text{KL}}$ and $\text{Fidelity}_{\text{prob}}$ are shown in Figures 15 and 16. For tgbn-trade and tgbn-genre datasets, two metrics initially decrease and then increase with depth. Deeper backtracking traces more historical events, but also enlarges the search space, complicating event identification and reducing fidelity. Enron and UCI datasets perform better at lower depths. The optimal depth varies across datasets due to their distinct characteristics.

**Case study**. We show the explanation results on pose-based action classification task. Table 5, Table 6, Table 7, and Table 8 show case studies for the pull-up, climb, sit-up, and run classes (in Appendix A.20). Our method typically selects a much smaller subset of edges per frame, which leads to more visualized frames under the same total edge budget. Across actions, our method highlights the task-relevant kinematic chains: upper and lower limb chains for pull-ups, the hip–knee–ankle chain for running, the supporting limbs for climbing, and the torso–hip–knee chain for sit-ups. In contrast, across all four actions, PGExplainer, GNNExplainer, and TGNNExplainer usually select most skeletal links in one frame, making it hard to see which joints actually drive the prediction. TempME selects edges misaligned with the key biomechanics of the movement.

# 6. Conclusions and Limitations

We study the problem of explaining predictions in TGNs. Existing methods fix the memory vector and fail to account

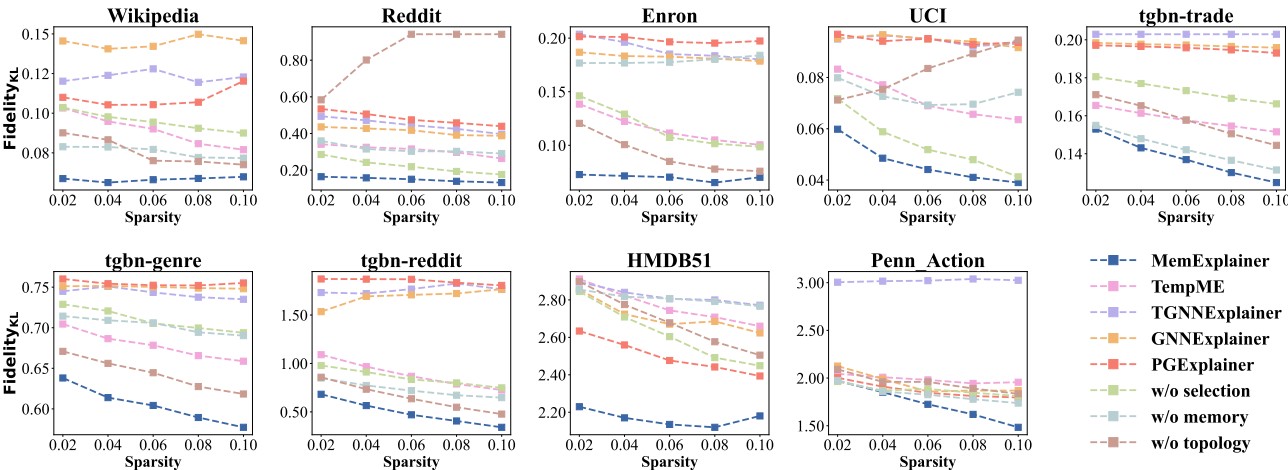

*Figure 5.* The performance of Fidelity$_{kl}$. Each figure corresponds to a different dataset. Lower value indicates better performance.

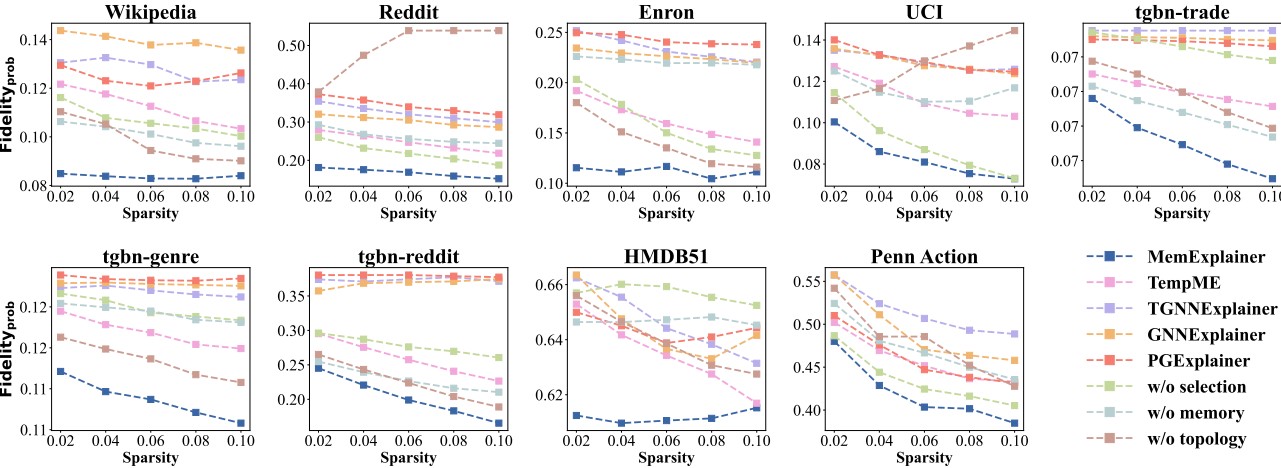

*Figure 6.* The performance of Fidelity$_{prob}$. Each figure corresponds to a different dataset. Lower value indicates better performance.

for the temporal evolution of node memories. To address this, we construct a topology attribution tree to quantify the spatial contribution of neighboring events and the temporal contribution of the node memories. We construct memory backtracking tree to track the long-term impact of historical events on the node memories. We design an objective function to select important events as explanations. Experiments show our method outperforms baselines across all datasets. One limitation is its computational cost on large temporal graphs, where the memory backtracking tree can become deep and wide, increasing runtime and memory usage. A practical solution is to limit the tree depth or prune branches with small relevance scores, which we leave for future work.

## Acknowledgements

Hui Xiong was supported in part by the National Key R&D Program of China (Grant No.2023YFF0725001), in part by the National Natural Science Foundation of China (Grant No.92370204), in part by the guangdong Basic and Applied Basic Research Foundation (Grant No.2023B1515120057), in part by the Key-Area Special Project of Guangdong Provincial Ordinary Universities(2024ZDZX1007). Sihong Xie was supported by the Department of Science and Technology of Guangdong Province (2023CX10X079), National Key R&D Program of China (Grant No.2023YFF0725001), the Guangzhou-HKUST(GZ) Joint Funding Program (Grant No.2023A03J0008), and Education Bureau Guangzhou Municipality. Xi Zhang was supported by the National Key Research and Development Program (Grant No. 2023YFC3303800), and the Natural Science Foundation of China (No. 62372057).

## Impact Statement

Our paper is a general algorithmic contribution, and we do not foresee direct negative societal impacts. On the positive side, our method can be used to help users understand TGNs.

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

# A. Appendix

## A.1. Notations

*Table 1.* Notations and their meanings

| Notations | Definitions and Descriptions |
|---|---|
| $t, t_k$ | Timestamp |
| $u, v$ | Node |
| $e_k = (v, u, t_k)$ | Event |
| $\mathbf{x}_u$ | The feature vectors of node $u$ |
| $\mathbf{x}_{e_k}$ | The feature vectors of event $e_k$ |
| $\mathbf{z}_v^t$ | The node embedding for node $u$ at timestamp t |
| $\mathbf{s}_u^t$ | Memory vector for node $u$ at timestamp $t$ |
| $\mathcal{N}_u[0, t] = \{e_k \mid e_k = (v, u, t_k) \in \mathcal{E}(t)\}$ | The neighborhood of node $u$ in time interval $t_k$ |
| $\mathcal{N}_u^n[0, t]$ | The set of the most recent $n$ interactions from $\mathcal{N}_u[0, t]$ |
| $\mathbf{R}\left(\mathbf{h}_u^{t,l-1}\right) \in \mathbb{R}^{d_m \times d_m}$ | Each matrix element represents the contribution of $i$-th neuron in $\mathbf{h}_u^{t,l-1}$ to the $j$-th in $\mathbf{z}_u^t$ |
| $\mathbf{R}\left(\mathbf{h}_v^{t,l-1}\right) \in \mathbb{R}^{d_m \times d_m}$ | Each matrix element represents the contribution of $i$-th neuron in $\mathbf{h}_v^{t,l-1}$ to the $j$-th in $\mathbf{z}_u^t$ |
| $\mathbf{R}\left(\mathbf{x}_{e_k}\right) \in \mathbb{R}^{d_e \times d_m}$ | Each matrix element represents the contribution of $i$-th neuron in $\mathbf{x}_{e_k}$ to the $j$-th in $\mathbf{z}_u^t$ |
| $\mathbf{R}(t - t_k) \in \mathbb{R}^{d_t \times d_m}$ | Each matrix element represents the contribution of $i$-th neuron in $\phi(t - t_k)$ to the $j$-th in $\mathbf{z}_u^t$ |
| $\mathbf{M}_{p_0 \to u}^t \in \mathbb{R}^{d_m \times d_m}$ | Each matrix element represents the contribution of $i$-th neurons in $\mathbf{s}_{p_0}^t$ to the $j$-th in $\mathbf{z}_u^t$ |
| $\mathbf{C}_u^t \in \mathbb{R}^{|\mathcal{E}(t)| \times d_m}$ | Each matrix element represents the contribution of $i$-th events in $\mathcal{E}(t)$ to the $j$-th in $\mathbf{z}_u^t$ |
| $\mathcal{T}_{top}(u)$ | The topology attribution tree of node $u$ |

## A.2. Proof of Lemma 4.1.

*Proof.* Let $\mathbf{x} = [x_1, \cdots x_{d_1}]$ and $\mathbf{y} = [y_1, \cdots y_{d_2}]$. We first broadcast the $\mathbf{x}^\top$ and $\mathbf{y}$ to align the shapes of $\mathbf{W}$.

$$\text{diag}(\mathbf{x}) = \begin{bmatrix} x_1 & 0 & \cdots & 0 \\ 0 & x_2 & \cdots & 0 \\ \vdots & \vdots & \ddots & \vdots \\ 0 & 0 & \cdots & x_{d_1} \end{bmatrix}, \quad \text{diag}(\mathbf{y})^{-1} = \begin{bmatrix} \frac{1}{y_1} & 0 & \cdots & 0 \\ 0 & \frac{1}{y_2} & \cdots & 0 \\ \vdots & \vdots & \ddots & \vdots \\ 0 & 0 & \cdots & \frac{1}{y_{d_2}} \end{bmatrix}.$$

Then, we can obtain the following equation:

$$\text{diag}(\mathbf{x}) \cdot \mathbf{W} \cdot \text{diag}(\mathbf{y})^{-1} = \begin{bmatrix} x_1 W_{1,1} & \cdots & x_1 W_{1,d_2} \\ x_2 W_{2,1} & \cdots & x_2 W_{2,d_2} \\ \vdots & \ddots & \vdots \\ x_{d_1} W_{d_1,1} & \cdots & x_{d_1} W_{d_1,d_2} \end{bmatrix} \cdot \begin{bmatrix} \frac{1}{y_1} & 0 & \cdots & 0 \\ 0 & \frac{1}{y_2} & \cdots & 0 \\ \vdots & \vdots & \ddots & \vdots \\ 0 & 0 & \cdots & \frac{1}{y_{d_2}} \end{bmatrix} = \begin{bmatrix} \frac{x_1 W_{1,1}}{y_1} & \cdots & \frac{x_1 W_{1,d_2}}{y_{d_2}} \\ \frac{x_2 W_{2,1}}{y_1} & \cdots & \frac{x_2 W_{2,d_2}}{y_{d_2}} \\ \vdots & \ddots & \vdots \\ \frac{x_{d_1} W_{d_1,1}}{y_1} & \cdots & \frac{x_{d_1} W_{d_1,d_2}}{y_{d_2}} \end{bmatrix} \quad (16)$$

Let $\mathbf{P} = \text{diag}(\mathbf{x}) \cdot \mathbf{W} \cdot \text{diag}(\mathbf{y})^{-1}$, and $\mathbf{y} = \mathbf{x} \cdot \mathbf{W}$, then,

$$\mathbf{1}^\top \cdot \mathbf{P} = \mathbf{1}^\top \cdot \text{diag}(\mathbf{x}) \cdot \mathbf{W} \cdot \text{diag}(\mathbf{y})^{-1} = \begin{bmatrix} \frac{\sum_i x_i W_{i,1}}{y_1} & \frac{\sum_i x_i W_{i,2}}{y_2} & \cdots & \frac{\sum_i x_i W_{i,d_2}}{y_{d_2}} \end{bmatrix} = \mathbf{1}^\top$$

Finally, the relevance contribution satisfies the conservation property.

$$\mathbf{R}(\mathbf{x}) = \mathbf{P} \cdot \mathbf{R}(\mathbf{y}), \quad \mathbf{1}^\top \cdot \mathbf{R}(\mathbf{x}) = \mathbf{1}^\top \cdot \mathbf{P} \cdot \mathbf{R}(\mathbf{y}) = \mathbf{1}^\top \cdot \mathbf{R}(\mathbf{y})$$

□

## A.3. Proof of proposition 4.2

*Proof.* Let $\mathbf{W}_2^l = \begin{bmatrix} \mathbf{A}^l \\ \mathbf{B}^l \end{bmatrix} \in \mathbb{R}^{2d_m \times d_m}$, where $\mathbf{A}^l, \mathbf{B}^l \in \mathbb{R}^{d_m \times d_m}$. Let $R\left(\mathbf{h}_u^{l-1}\right) \in \mathbb{R}^{d_m \times d_m}$ and $R\left(\tilde{\mathbf{h}}_u^l\right) \in \mathbb{R}^{d_m \times d_m}$ denote the contributions of $\mathbf{h}_u^{l-1}$ and $\tilde{\mathbf{h}}_u^l$ to $\mathbf{z}_u^t$, respectively. The element in the $i$-th row and $j$-th column of each matrix represents the contribution of the $i$-th neuron in $\mathbf{h}_u^{l-1}$ and $\tilde{\mathbf{h}}_u^l$ to the $j$-th neuron in $\mathbf{z}_u^t$, respectively. According to the Lemma 4.1, the matrices $\mathbf{R}\left(\mathbf{h}_u^{l-1}\right)$ and $\mathbf{R}\left(\tilde{\mathbf{h}}_u^l\right)$ are given by the following equations.

$$\mathbf{R}\big(\mathbf{h}_u^{l-1}\big) = \mathrm{diag}\big(\mathbf{h}_u^{l-1}\big) \cdot \mathbf{A}^l \cdot \mathrm{diag}\big(\mathbf{h}_u^l\big)^{-1} \cdot \mathbf{R}\big(\mathbf{h}_u^l\big),$$

$$\mathbf{R}\Big(\tilde{\mathbf{h}}_u^l\Big) = \mathrm{diag}\Big(\tilde{\mathbf{h}}_u^l\Big) \cdot \mathbf{B}^l \cdot \mathrm{diag}\big(\mathbf{h}_u^l\big)^{-1} \cdot \mathbf{R}\big(\mathbf{h}_u^l\big),$$

$$\mathbf{1}^\top \mathbf{R}\big(\mathbf{h}_u^l\big) = \mathbf{1}^\top \mathbf{R}\big(\mathbf{h}_u^{l-1}\big) + \mathbf{1}^\top \mathbf{R}\Big(\tilde{\mathbf{h}}_u^l\Big) \tag{17}$$

Where $\odot$ denotes the element-wise multiplication, $-$ represents element-wise division, and $\cdot$ denotes matrix multiplication.

Similarly, let $\hat{\mathbf{h}}_u^l = \sum_{e_k \in \mathcal{N}_u^n([0,t])} \Big(\mathbf{h}_v^{l-1} \,\|\, \mathbf{x}_{e_k} \,\|\, \phi(t - t_k)\Big) \mathbf{W}_1^l$. Let $\mathbf{W}_1^l = \begin{bmatrix} \mathbf{C}^l \\ \mathbf{D}^l \\ \mathbf{E}^l \end{bmatrix} \in \mathbb{R}^{(d_m + d_e + d_t) \times d_m}$, where $\mathbf{C}^l \in$

$\mathbb{R}^{d_m \times d_m}, \mathbf{D}^l \in \mathbb{R}^{d_e \times d_m}, \mathbf{E}^l \in \mathbb{R}^{d_t \times d_m}$. We apply LRP on the third equation in Eq. (10), then:

$$\mathbf{R}\big(\mathbf{h}_v^{l-1}\big) = \mathrm{diag}\big(\mathbf{h}_v^{l-1}\big) \cdot \mathbf{C}^l \cdot \mathrm{diag}\Big(\hat{\mathbf{h}}_u^l\Big)^{-1} \cdot \mathbf{R}\Big(\hat{\mathbf{h}}_u^l\Big)$$

$$\mathbf{R}(\mathbf{x}_{e_k}) = \mathrm{diag}(\mathbf{x}_{e_k}) \cdot \mathbf{D}^l \cdot \mathrm{diag}\Big(\hat{\mathbf{h}}_u^l\Big)^{-1} \cdot \mathbf{R}\Big(\hat{\mathbf{h}}_u^l\Big)$$

$$\mathbf{R}(t - t_k) = \mathrm{diag}(\phi(t - t_k)) \cdot \mathbf{E}^l \cdot \mathrm{diag}\Big(\hat{\mathbf{h}}_u^l\Big)^{-1} \cdot \mathbf{R}\Big(\hat{\mathbf{h}}_u^l\Big)$$

$$\mathbf{1}^\top \mathbf{R}\Big(\hat{\mathbf{h}}_u^l\Big) = \sum_{e_k = (v, u, t_k) \in \mathcal{N}_u^n([0,t])} \big[\mathbf{1}^\top \mathbf{R}\big(\mathbf{h}_v^{l-1}\big) + \mathbf{1}^\top \mathbf{R}(\mathbf{x}_{e_k}) + \mathbf{1}^\top \mathbf{R}(t - t_k)\big]$$

For the softmax function, we use the identity relevance propagation rule, i.e. $\mathbf{R}\Big(\hat{\mathbf{h}}_u^l\Big) = \mathbf{R}\Big(\tilde{\mathbf{h}}_u^l\Big)$, Thus,

$$\mathbf{R}\big(\mathbf{h}_v^{l-1}\big) = \mathrm{diag}\big(\mathbf{h}_v^{l-1}\big) \cdot \mathbf{C}^l \cdot \mathrm{diag}\Big(\hat{\mathbf{h}}_u^l\Big)^{-1} \cdot \mathrm{diag}\Big(\tilde{\mathbf{h}}_u^l\Big) \cdot \mathbf{B}^l \cdot \mathrm{diag}\big(\mathbf{h}_u^l\big)^{-1} \cdot \mathbf{R}\Big(\tilde{\mathbf{h}}_u^l\Big),$$

$$\mathbf{R}(\mathbf{x}_{e_k}) = \mathrm{diag}(\mathbf{x}_{e_k}) \cdot \mathbf{D}^l \cdot \mathrm{diag}\Big(\hat{\mathbf{h}}_u^l\Big)^{-1} \cdot \mathrm{diag}\Big(\tilde{\mathbf{h}}_u^l\Big) \cdot \mathbf{B}^l \cdot \mathrm{diag}\big(\mathbf{h}_u^l\big)^{-1} \cdot \mathbf{R}\Big(\tilde{\mathbf{h}}_u^l\Big),$$

$$\mathbf{R}(t - t_k) = \mathrm{diag}(\phi(t - t_k)) \cdot \mathbf{E}^l \cdot \mathrm{diag}\Big(\hat{\mathbf{h}}_u^l\Big)^{-1} \cdot \mathrm{diag}\Big(\tilde{\mathbf{h}}_u^l\Big) \cdot \mathbf{B}^l \cdot \mathrm{diag}\big(\mathbf{h}_u^l\big)^{-1} \cdot \mathbf{R}\Big(\tilde{\mathbf{h}}_u^l\Big).$$

Thus,
$$\mathbf{1}^\top \mathbf{R}\big(\mathbf{h}_u^l\big) = \mathbf{1}^\top \mathbf{R}\big(\mathbf{h}_u^{l-1}\big) + \sum_{e_k = (v, u, t_k) \in \mathcal{N}_u^n([0,t])} \big[\mathbf{1}^\top \mathbf{R}\big(\mathbf{h}_v^{l-1}\big) + \mathbf{1}^\top \mathbf{R}(\mathbf{x}_{e_k}) + \mathbf{1}^\top \mathbf{R}(t - t_k)\big]. \tag{18}$$

$\square$

### A.4. Derivation of applying LRP to $f_{\mathrm{emb}}$, when $f_{\mathrm{emb}}$ is the graph attention function

If the $f_{\mathrm{emb}}$ is graph attention function, let $\hat{\mathbf{h}}_v^l = [\mathbf{h}_v^{l-1} \| \mathbf{x}_{e_k} \| \phi(t - t_k)]$ denote the neighbor embedding. Then, $\mathbf{h}_{\mathrm{neighbor}}^l$ is the set of embeddings of neighboring nodes at time step $t$ and layer $l$, constructed by stacking $\hat{\mathbf{h}}_v^l$ for each $e_k = (v, u, t_k) \in \mathcal{N}_u^n[0, t]$. The Eq. (4) becomes:

$$\hat{\mathbf{h}}_v^l = [\mathbf{h}_v^{l-1} \| \mathbf{x}_{e_k} \| \phi(t - t_k)], \quad \mathbf{h}_{\mathrm{neighor}}^{t,l} = \begin{bmatrix} \cdots \\ \hat{\mathbf{h}}_v^l \\ \cdots \end{bmatrix}_{(v, u, t_k) \in \mathcal{N}_u^n[0,t]}, \quad \mathbf{K}^l = \mathbf{h}_{\mathrm{neighor}}^l \mathbf{W}_K^l,$$

$$\mathbf{V}^l = \mathbf{K}^l = \mathbf{h}_{\mathrm{neighor}}^l \mathbf{W}_K^l, \quad \mathbf{q}^l = \Big(\mathbf{h}_u^{l-1} \| \phi(0)\Big) \mathbf{W}_q^l, \quad \mathbf{P}^l = \frac{\mathbf{q}^l (\mathbf{K}^l)^\top}{\sqrt{d_m}},$$

$$\hat{\mathbf{P}}^l = \mathrm{softmax}(\mathbf{P}^l), \quad \mathbf{O}^l = \hat{\mathbf{P}}^l \mathbf{V}^l, \quad \tilde{\mathbf{h}}_u^l = \mathbf{O}^l \mathbf{W}_O^l, \quad \mathbf{h}_u^l = \Big(\mathbf{h}_u^{l-1} \| \tilde{\mathbf{h}}_u^l\Big) \mathbf{W}_2^l \tag{19}$$

Where the dimensions are $\hat{\mathbf{h}}_v^l \in \mathbb{R}^{1 \times (d_m + d_e + d_t)}$, $\mathbf{h}_{\mathrm{neighbor}}^l \in \mathbb{R}^{n \times (d_m + d_e + d_t)}$, $\mathbf{W}_K^l \in \mathbb{R}^{(d_m + d_e + d_t) \times d_m}$, $\mathbf{K}^l \in \mathbb{R}^{n \times d_m}$, $\mathbf{V}^l \in \mathbb{R}^{n \times d_m}$, $\mathbf{W}_q^l \in \mathbb{R}^{(d_m + d_t) \times d_m}$, $\mathbf{q}^l \in \mathbb{R}^{1 \times d_m}$, $\mathbf{P}^l \in \mathbb{R}^{1 \times n}$, $\mathbf{O}^l \in \mathbb{R}^{1 \times d_m}$, $\mathbf{W}_O^l \in \mathbb{R}^{d_m \times d_m}$, $\tilde{\mathbf{h}}_u^l \in \mathbb{R}^{1 \times d_m}$, $\mathbf{W}_2^l \in \mathbb{R}^{2d_m \times d_m}$.

According to the Lemma 4.1, we can obtain the following equation:

$$\mathbf{R}\big(\mathbf{h}_u^{l-1}\big) = \mathrm{diag}\big(\mathbf{h}_u^{l-1}\big) \cdot \mathbf{A}^l \cdot \mathrm{diag}\big(\mathbf{h}_u^l\big)^{-1} \cdot \mathbf{R}\big(\mathbf{h}_u^l\big), \quad \mathbf{R}\big(\tilde{\mathbf{h}}_u^l\big) = \mathrm{diag}\big(\tilde{\mathbf{h}}_u^l\big) \cdot \mathbf{B}^l \cdot \mathrm{diag}\big(\mathbf{h}_u^l\big)^{-1} \cdot \mathbf{R}\big(\mathbf{h}_u^l\big),$$

$$\mathbf{R}\big(\mathbf{O}^l\big) = \mathrm{diag}\big(\mathbf{O}^l\big) \cdot \mathbf{W}_O^l \cdot \mathrm{diag}\big(\tilde{\mathbf{h}}_u^l\big)^{-1} \cdot \mathbf{R}\big(\tilde{\mathbf{h}}_u^l\big) \tag{20}$$

Where $\mathbf{R}\big(\mathbf{O}^l\big) \in \mathbb{R}^{1 \times d_m \times d_m}$ denote the contribution to $\mathbf{z}_u^t$.

The matrix multiplication $\mathbf{O}^l = \hat{\mathbf{P}}^l \mathbf{V}^l$ is treated as a bilinear operation using the AttnLRP method (Achtibat et al., 2024). In this approach, when assigning contribution values to $\hat{\mathbf{P}}^l$, $\mathbf{V}^l$ is treated as a fixed parameters, and vice versa. Let $\mathbf{O}_{\mathrm{ratio}}^l = \mathrm{diag}\big(\hat{\mathbf{P}}^l\big) \cdot \mathbf{V}^l \cdot \mathrm{diag}\big(\mathbf{O}^l\big)^{-1}$ and $\mathbf{O}_{\mathrm{ratio}}^l[i]$ denote the $i$-th row of $\mathbf{O}_{\mathrm{ratio}}^l$, we can obtain the following equations:

$$\mathbf{O}_{\mathrm{ratio}}^l = \mathrm{diag}\big(\hat{\mathbf{P}}^l\big) \cdot \mathbf{V}^l \cdot \mathrm{diag}\big(\mathbf{O}^l\big)^{-1} \in \mathbb{R}^{n \times d_m}, \quad \mathbf{R}\big(\hat{\mathbf{P}}^l\big) = \frac{1}{2}\mathbf{O}_{\mathrm{ratio}}^l \cdot \mathbf{R}\big(\mathbf{O}^l\big)$$

$$\mathbf{O}_{\mathrm{expanded}}^l = \begin{bmatrix} \mathrm{diag}(\mathbf{O}_{\mathrm{ratio}}^l[1]) \\ \cdots \\ \mathrm{diag}(\mathbf{O}_{\mathrm{ratio}}^l[n]) \end{bmatrix} \in \mathbb{R}^{n \times d_m \times d_m}, \quad \mathbf{R}\big(\mathbf{V}^l\big) = \frac{1}{2}\mathbf{O}_{\mathrm{expanded}}^l \cdot \mathbf{R}\big(\mathbf{O}^l\big) \tag{21}$$

Where $\mathrm{diag}$ denotes the diagonalization operator that maps a vector to a diagonal matrix. $\mathbf{R}\big(\mathbf{V}^l\big) \in \mathbb{R}^{n \times d_m \times d_m}$, $\mathbf{R}\big(\hat{\mathbf{P}}^l\big) \in \mathbb{R}^{1 \times n \times d_m}$ denote the contribution to $\mathbf{z}_u^t$.

We continue to decompose contribution of $\mathbf{V}^l$. Let $\mathbf{W}_K^l = \begin{bmatrix} \mathbf{W}_{KC}^l \\ \mathbf{W}_{KD}^l \\ \mathbf{W}_{KE}^l \end{bmatrix} \in \mathbb{R}^{(d_m+d_e+d_t) \times d_m}$, where $\mathbf{W}_{KC}^l \in \mathbb{R}^{d_m \times d_m}, \mathbf{W}_{KD}^l \in \mathbb{R}^{d_e \times d_m}, \mathbf{W}_{KE}^l \in \mathbb{R}^{d_t \times d_m}$. Supposing that $\mathbf{h}_{\mathrm{neighbor}}^l[i] = \hat{\mathbf{h}}_v^l$, then

$$\mathbf{R}\big(\mathbf{h}_v^{l-1}\big) = \mathrm{diag}\big(\mathbf{h}_v^{l-1}\big) \cdot \mathbf{W}_{KC}^l \cdot \mathrm{diag}\big(\mathbf{V}^l[i]\big)^{-1} \cdot \big(\mathbf{R}\big(\mathbf{V}^l\big)[i]\big)$$

$$\mathbf{R}(\mathbf{x}_{e_k}) = \mathrm{diag}(\mathbf{x}_{e_k}) \cdot \mathbf{W}_{KD}^l \cdot \mathrm{diag}\big(\mathbf{V}^l[i]\big)^{-1} \cdot \big(\mathbf{R}\big(\mathbf{V}^l\big)[i]\big), \tag{22}$$

$$\mathbf{R}(\phi(t - t_k)) = \mathrm{diag}(\phi(t - t_k)) \cdot \mathbf{W}_{KE}^l \cdot \mathrm{diag}\big(\mathbf{V}^l[i]\big)^{-1} \cdot \big(\mathbf{R}\big(\mathbf{V}^l\big)[i]\big) \tag{23}$$

We decompose contribution of $\hat{\mathbf{P}}^l$. For the softmax function, we use the identity propagation rule, i.e. $\mathbf{R}\big(\hat{\mathbf{P}}^l\big) = \mathbf{R}\big(\mathbf{P}^l\big)$. Similarly, we use the AttnLRP method to handle the matrix multiplication. Let $\mathbf{P}_{\mathrm{ratio}}^l[i]$ denote the $i$-th row of $\mathbf{P}_{\mathrm{ratio}}^l$

$$\mathbf{P}_{\mathrm{ratio}}^l = \mathrm{diag}\big(\mathbf{q}^l\big) \cdot \big(\mathbf{K}^l\big)^\top \cdot \mathrm{diag}\big(\mathbf{P}^l\big)^{-1} \in \mathbb{R}^{d_m \times n}, \quad \mathbf{R}\big(\mathbf{q}^l\big) = \frac{1}{2\sqrt{d_m}}\mathbf{P}_{\mathrm{ratio}}^l \cdot \mathbf{R}\big(\mathbf{P}^l\big)$$

$$\mathbf{P}_{\mathrm{expanded}}^l = \begin{bmatrix} \mathrm{diag}(\mathbf{P}_{\mathrm{ratio}}^l[1]) \\ \cdots \\ \mathrm{diag}(\mathbf{P}_{\mathrm{ratio}}^l[n]) \end{bmatrix} \in \mathbb{R}^{d_m \times n \times n}, \quad \mathbf{R}\big(\mathbf{K}^l\big) = \frac{1}{2\sqrt{d_m}}\big(\mathbf{P}_{\mathrm{expanded}}^l \cdot \mathbf{R}\big(\mathbf{P}^l\big)\big)^\top \tag{24}$$

Where $\mathrm{diag}$ denotes the diagonalization operator that maps a vector to a diagonal matrix. $\mathbf{R}\big(\mathbf{K}^l\big) \in \mathbb{R}^{n \times d_m \times d_m}$, $\mathbf{R}\big(\mathbf{q}^l\big) \in \mathbb{R}^{1 \times n \times d_m}$ denote the contribution to $\mathbf{z}_u^t$.

Let $\mathbf{W}_q^l = \begin{bmatrix} \mathbf{W}_{qa}^l \\ \mathbf{W}_{qb}^l \end{bmatrix} \in \mathbb{R}^{(d_m+d_t) \times d_m}$, where $\mathbf{W}_{qa}^l \in \mathbb{R}^{d_m \times d_m}, \mathbf{W}_{qb}^l \in \mathbb{R}^{d_t \times d_m}$. Supposing that $\mathbf{h}_{\mathrm{neighbor}}^l[i] = \hat{\mathbf{h}}_v^l$. Let $R\big(\mathbf{h}_u^{l-1}\big) \in \mathbb{R}^{d_m \times d_m}$, then we attribute the $\mathbf{R}\big(\mathbf{q}^l\big)$ and $\mathbf{R}\big(\mathbf{K}^l\big)$ to $\mathbf{h}_v^{l-1}$, $\mathbf{x}_{e_k}$ and $\phi(t - t_k)$:

$$\mathbf{R}\big(\mathbf{h}_u^{l-1}\big) = \mathrm{diag}\big(\mathbf{h}_u^{l-1}\big) \cdot \mathbf{W}_{qa}^l \cdot \mathrm{diag}\big(\mathbf{q}^l\big)^{-1} \cdot \mathbf{R}\big(\mathbf{q}^l\big)$$

$$\mathbf{R}(\phi(0)) = \mathrm{diag}(\phi(0)) \cdot \mathbf{W}_{qb}^l \cdot \mathrm{diag}\big(\mathbf{q}^l\big)^{-1} \cdot \mathbf{R}\big(\mathbf{q}^l\big)$$

$$\mathbf{R}\big(\mathbf{h}_v^{l-1}\big) = \mathrm{diag}\big(\mathbf{h}_v^{l-1}\big) \cdot \mathbf{W}_{KC}^l \cdot \mathrm{diag}\big(\mathbf{K}^l[i]\big)^{-1} \cdot \big(\mathbf{R}\big(\mathbf{K}^l\big)[i]\big)$$

$$\mathbf{R}(\mathbf{x}_{e_k}) = \mathrm{diag}(\mathbf{x}_{e_k}) \cdot \mathbf{W}_{KD}^l \cdot \mathrm{diag}\big(\mathbf{K}^l[i]\big)^{-1} \cdot \big(\mathbf{R}\big(\mathbf{K}^l\big)[i]\big)$$

$$\mathbf{R}(\phi(t - t_k)) = \mathrm{diag}(\phi(t - t_k)) \cdot \mathbf{W}_{KE}^l \cdot \mathrm{diag}\big(\mathbf{K}^l[i]\big)^{-1} \cdot \big(\mathbf{R}\big(\mathbf{K}^l\big)[i]\big) \tag{25}$$

We suppose that the $i$-th element in $\mathcal{N}_u^n[0,t]$ is node $v$. We can obtain the contributions of $\mathbf{h}_u^{l-1}$, $\mathbf{h}_v^{l-1}$, $\mathbf{x}_{e_k}$, $\phi(t-t_k)$ and $\phi(0)$ to $\mathbf{z}_u^t$, denoted as $\mathbf{R}(\mathbf{h}_u^{l-1}) \in \mathbb{R}^{d_m \times d_m}$, $\mathbf{R}(\mathbf{h}_v^{l-1}) \in \mathbb{R}^{d_m \times d_m}$, $\mathbf{R}(\mathbf{x}_{e_k}) \in \mathbb{R}^{d_e \times d_m}$ and $\mathbf{R}(\phi(t-t_k)) \in \mathbb{R}^{d_t \times d_m}$:

$$
\begin{aligned}
\mathbf{R}(\phi(0)) = {}& \frac{1}{4\sqrt{d_m}} \, \mathrm{diag}(\phi(0)) \cdot \mathbf{W}_{qb}^l \cdot \mathrm{diag}(\mathbf{q}^l)^{-1} \cdot \mathrm{diag}(\mathbf{q}^l) \cdot (\mathbf{K}^l)^\top \cdot \mathrm{diag}(\mathbf{P}^l)^{-1} \\
& \cdot \mathrm{diag}(\hat{\mathbf{P}}^l) \cdot \mathbf{V}^l \cdot \mathrm{diag}(\mathbf{O}^l)^{-1} \cdot \mathrm{diag}(\mathbf{O}^l) \cdot \mathbf{W}_O^l \cdot \mathrm{diag}(\tilde{\mathbf{h}}_u^l)^{-1} \\
& \cdot \mathrm{diag}(\tilde{\mathbf{h}}_u^l) \cdot \mathbf{B}^l \cdot \mathrm{diag}(\mathbf{h}_u^l)^{-1} \cdot \mathbf{R}(\mathbf{h}_u^l) \\
\mathbf{R}(\mathbf{x}_{e_k}) = {}& \frac{1}{4\sqrt{d_m}} \, \mathrm{diag}(\mathbf{x}_{e_k}) \cdot \mathbf{W}_{KD}^l \cdot \mathrm{diag}(\mathbf{K}^l[i])^{-1} \cdot \mathrm{diag}(\mathbf{K}^l[i]) \cdot (\mathbf{q}^l)^\top \cdot (\mathbf{P}^l[i])^{-1} \\
& \cdot \left[ \mathrm{diag}(\hat{\mathbf{P}}^l) \cdot \mathbf{V}^l \cdot \mathrm{diag}(\mathbf{O}^l)^{-1} \cdot \mathrm{diag}(\mathbf{O}^l) \cdot \mathbf{W}_O^l \cdot \mathrm{diag}(\tilde{\mathbf{h}}_u^l)^{-1} \right. \\
& \left. \quad \cdot \mathrm{diag}(\tilde{\mathbf{h}}_u^l) \cdot \mathbf{B}^l \cdot \mathrm{diag}(\mathbf{h}_u^l)^{-1} \cdot \mathbf{R}(\mathbf{h}_u^l) \right]_{i,:} \\
& + \frac{1}{2} \, \mathrm{diag}(\mathbf{x}_{e_k}) \cdot \mathbf{W}_{KD}^l \cdot \mathrm{diag}(\mathbf{V}^l[i])^{-1} \cdot \mathrm{diag}(\mathbf{V}^l[i]) \cdot \left( \hat{\mathbf{P}}^l[i] \mathbf{I}_{d_m} \right) \\
& \cdot \mathrm{diag}(\mathbf{O}^l)^{-1} \cdot \mathrm{diag}(\mathbf{O}^l) \cdot \mathbf{W}_O^l \cdot \mathrm{diag}(\tilde{\mathbf{h}}_u^l)^{-1} \cdot \mathrm{diag}(\tilde{\mathbf{h}}_u^l) \\
& \cdot \mathbf{B}^l \cdot \mathrm{diag}(\mathbf{h}_u^l)^{-1} \cdot \mathbf{R}(\mathbf{h}_u^l) \\
\mathbf{R}(\phi(t-t_k)) = {}& \frac{1}{4\sqrt{d_m}} \, \mathrm{diag}(\phi(t-t_k)) \cdot \mathbf{W}_{KE}^l \cdot \mathrm{diag}(\mathbf{K}^l[i])^{-1} \cdot \mathrm{diag}(\mathbf{K}^l[i]) \cdot (\mathbf{q}^l)^\top \cdot (\mathbf{P}^l[i])^{-1} \\
& \cdot \left[ \mathrm{diag}(\hat{\mathbf{P}}^l) \cdot \mathbf{V}^l \cdot \mathrm{diag}(\mathbf{O}^l)^{-1} \cdot \mathrm{diag}(\mathbf{O}^l) \cdot \mathbf{W}_O^l \cdot \mathrm{diag}(\tilde{\mathbf{h}}_u^l)^{-1} \right. \\
& \left. \quad \cdot \mathrm{diag}(\tilde{\mathbf{h}}_u^l) \cdot \mathbf{B}^l \cdot \mathrm{diag}(\mathbf{h}_u^l)^{-1} \cdot \mathbf{R}(\mathbf{h}_u^l) \right]_{i,:} \\
& + \frac{1}{2} \, \mathrm{diag}(\phi(t-t_k)) \cdot \mathbf{W}_{KE}^l \cdot \mathrm{diag}(\mathbf{V}^l[i])^{-1} \cdot \mathrm{diag}(\mathbf{V}^l[i]) \cdot \left( \hat{\mathbf{P}}^l[i] \mathbf{I}_{d_m} \right) \\
& \cdot \mathrm{diag}(\mathbf{O}^l)^{-1} \cdot \mathrm{diag}(\mathbf{O}^l) \cdot \mathbf{W}_O^l \cdot \mathrm{diag}(\tilde{\mathbf{h}}_u^l)^{-1} \cdot \mathrm{diag}(\tilde{\mathbf{h}}_u^l) \\
& \cdot \mathbf{B}^l \cdot \mathrm{diag}(\mathbf{h}_u^l)^{-1} \cdot \mathbf{R}(\mathbf{h}_u^l) \\
\mathbf{R}(\mathbf{h}_v^{l-1}) = {}& \frac{1}{4\sqrt{d_m}} \, \mathrm{diag}(\mathbf{h}_v^{l-1}) \cdot \mathbf{W}_{KC}^l \cdot \mathrm{diag}(\mathbf{K}^l[i])^{-1} \cdot \mathrm{diag}(\mathbf{K}^l[i]) \cdot (\mathbf{q}^l)^\top \cdot (\mathbf{P}^l[i])^{-1} \\
& \cdot \left[ \mathrm{diag}(\hat{\mathbf{P}}^l) \cdot \mathbf{V}^l \cdot \mathrm{diag}(\mathbf{O}^l)^{-1} \cdot \mathrm{diag}(\mathbf{O}^l) \cdot \mathbf{W}_O^l \cdot \mathrm{diag}(\tilde{\mathbf{h}}_u^l)^{-1} \right. \\
& \left. \quad \cdot \mathrm{diag}(\tilde{\mathbf{h}}_u^l) \cdot \mathbf{B}^l \cdot \mathrm{diag}(\mathbf{h}_u^l)^{-1} \cdot \mathbf{R}(\mathbf{h}_u^l) \right]_{i,:} \\
& + \frac{1}{2} \, \mathrm{diag}(\mathbf{h}_v^{l-1}) \cdot \mathbf{W}_{KC}^l \cdot \mathrm{diag}(\mathbf{V}^l[i])^{-1} \cdot \mathrm{diag}(\mathbf{V}^l[i]) \cdot \left( \hat{\mathbf{P}}^l[i] \mathbf{I}_{d_m} \right) \\
& \cdot \mathrm{diag}(\mathbf{O}^l)^{-1} \cdot \mathrm{diag}(\mathbf{O}^l) \cdot \mathbf{W}_O^l \cdot \mathrm{diag}(\tilde{\mathbf{h}}_u^l)^{-1} \cdot \mathrm{diag}(\tilde{\mathbf{h}}_u^l) \\
& \cdot \mathbf{B}^l \cdot \mathrm{diag}(\mathbf{h}_u^l)^{-1} \cdot \mathbf{R}(\mathbf{h}_u^l) \\
\mathbf{R}(\mathbf{h}_u^{l-1}) = {}& \mathrm{diag}(\mathbf{h}_u^{l-1}) \cdot \mathbf{A}^l \cdot \mathrm{diag}(\mathbf{h}_u^l)^{-1} \cdot \mathbf{R}(\mathbf{h}_u^l) \\
& + \frac{1}{4\sqrt{d_m}} \, \mathrm{diag}(\mathbf{h}_u^{l-1}) \cdot \mathbf{W}_{qa}^l \cdot \mathrm{diag}(\mathbf{q}^l)^{-1} \cdot \mathrm{diag}(\mathbf{q}^l) \cdot (\mathbf{K}^l)^\top \cdot \mathrm{diag}(\mathbf{P}^l)^{-1} \\
& \cdot \mathrm{diag}(\hat{\mathbf{P}}^l) \cdot \mathbf{V}^l \cdot \mathrm{diag}(\mathbf{O}^l)^{-1} \cdot \mathrm{diag}(\mathbf{O}^l) \cdot \mathbf{W}_O^l \cdot \mathrm{diag}(\tilde{\mathbf{h}}_u^l)^{-1} \\
& \cdot \mathrm{diag}(\tilde{\mathbf{h}}_u^l) \cdot \mathbf{B}^l \cdot \mathrm{diag}(\mathbf{h}_u^l)^{-1} \cdot \mathbf{R}(\mathbf{h}_u^l)
\end{aligned}
\tag{26}
$$

### A.5. Proof of proposition 4.3

*Proof.* If the $f_{\text{update}}$ is the GRU function, the Eq. (3) can be rewritten as

$$
\begin{aligned}
\text{Reset gate:} \quad & \mathbf{r}_u^t = \sigma\big(\bar{\mathbf{m}}_u^t \mathbf{W}_r + \mathbf{s}_u^{t-} \mathbf{U}_r^t + \mathbf{b}_r\big), \\
\text{Update gate:} \quad & \mathbf{g}_u^t = \sigma\big(\bar{\mathbf{m}}_u^t \mathbf{W}_g + \mathbf{s}_u^{t-} \mathbf{U}_g + \mathbf{b}_g\big), \\
\text{Candidate state:} \quad & \tilde{\mathbf{s}}_u^t = \tanh\big(\bar{\mathbf{m}}_u^t \mathbf{W}_h + \mathbf{r}_u^t \odot (\mathbf{s}_u^{t-} \mathbf{U}_h) + \mathbf{b}_h\big), \\
\text{Updated state:} \quad & \mathbf{s}_u^t = (1 - \mathbf{g}_u^t) \odot \mathbf{s}_u^{t-} + \mathbf{g}_u^t \odot \tilde{\mathbf{s}}_u^t.
\end{aligned}
$$

Where, $\mathbf{W}_r, \mathbf{W}_g, \mathbf{W}_h, \mathbf{U}_r, \mathbf{U}_g, \mathbf{U}_h, \mathbf{b}_r, \mathbf{b}_g, \mathbf{b}_h$ are the parameters in the GRU.

In GRU computation, multiplicative interactions occur when a gate neuron modulates a signal neuron, e.g., $\mathbf{g}_u^t[i] \cdot \mathbf{s}_u^{t-}[i]$. Unlike linear mappings, such interactions pose challenges for relevance redistribution. A widely adopted strategy is the signal-take-all rule, which assigns all relevance to the signal neuron and none to the gate. This reflects the view that the gate controls the flow of information, but is not information itself (Wu et al., 2022). Thus,

$$
\mathbf{R}\big(\mathbf{s}_u^{t-}\big) = \text{diag}\bigg(\frac{(1 - \mathbf{g}_u^t) \odot \mathbf{s}_u^{t-}}{\mathbf{s}_u^t}\bigg) \cdot \mathbf{R}\big(\mathbf{s}_u^t\big), \quad \mathbf{R}\big(\tilde{\mathbf{s}}_u^t\big) = \text{diag}\bigg(\frac{\mathbf{g}_u^t \odot \tilde{\mathbf{s}}_u^t}{\mathbf{s}_u^t}\bigg) \cdot \mathbf{R}\big(\mathbf{s}_u^t\big)
$$

$$
\mathbf{1}^\top \mathbf{R}\big(\mathbf{s}_u^t\big) = \mathbf{1}^\top \mathbf{R}\big(\mathbf{s}_u^{t-}\big) + \mathbf{1}^\top \mathbf{R}\big(\tilde{\mathbf{s}}_u^t\big) \tag{27}
$$

We continue to decompose $\mathbf{R}(\tilde{\mathbf{s}}_u^t)$ and let $\mathbf{A} = \bar{\mathbf{m}}_u^t \mathbf{W}_h + \mathbf{r}_u^t \odot (\mathbf{s}_u^{t-} \mathbf{U}_h)$:

$$
\mathbf{R}\big(\bar{\mathbf{m}}_u^t\big) = \text{diag}\big(\bar{\mathbf{m}}_u^t\big) \cdot \mathbf{W}_h \cdot \text{diag}(\mathbf{A})^{-1} \cdot \mathbf{R}\big(\tilde{\mathbf{s}}_u^t\big), \quad \mathbf{R}\big(\mathbf{s}_u^{t-}\big) = \text{diag}\big(\mathbf{r}_u^t \odot \mathbf{s}_u^{t-}\big) \cdot \mathbf{U}_h \cdot \text{diag}(\mathbf{A})^{-1} \cdot \mathbf{R}\big(\tilde{\mathbf{s}}_u^t\big)
$$

$$
\mathbf{1}^\top \mathbf{R}\big(\tilde{\mathbf{s}}_u^t\big) = \mathbf{1}^\top \mathbf{R}\big(\bar{\mathbf{m}}_u^t\big) + \mathbf{1}^\top \mathbf{R}\big(\mathbf{s}_u^{t-}\big) \tag{28}
$$

Finally,

$$
\mathbf{R}\big(\mathbf{s}_u^{t-}\big) = \bigg(\text{diag}\bigg(\frac{(1 - \mathbf{g}_u^t) \odot \mathbf{s}_u^{t-}}{\mathbf{s}_u^t}\bigg) + \text{diag}\big(\mathbf{r}_u^t \odot \mathbf{s}_u^{t-}\big) \cdot \mathbf{U}_h \cdot \text{diag}(\mathbf{A})^{-1} \cdot \text{diag}\bigg(\frac{\mathbf{g}_u^t \odot \tilde{\mathbf{s}}_u^t}{\mathbf{s}_u^t}\bigg)\bigg) \cdot \mathbf{R}\big(\mathbf{s}_u^t\big)
$$

$$
\mathbf{R}\big(\bar{\mathbf{m}}_u^t\big) = \text{diag}\big(\bar{\mathbf{m}}_u^t\big) \cdot \mathbf{W}_h \cdot \text{diag}(\mathbf{A})^{-1} \cdot \text{diag}\bigg(\frac{\mathbf{g}_u^t \odot \tilde{\mathbf{s}}_u^t}{\mathbf{s}_u^t}\bigg) \cdot \mathbf{R}\big(\mathbf{s}_u^t\big),
$$

$$
\mathbf{R}\big(\bar{\mathbf{m}}_u^t\big) = [\mathbf{R}\big(\mathbf{s}_v^{t-}\big), \mathbf{R}\big(\mathbf{s}_u^{t-}\big), \mathbf{R}(\mathbf{x}_{e_k}), \mathbf{R}(t - t_k)]
$$

$$
\mathbf{1}^\top \mathbf{R}\big(\mathbf{s}_u^t\big) = \mathbf{1}^\top \mathbf{R}\big(\mathbf{s}_u^{t-}\big) + \mathbf{1}^\top \mathbf{R}\big(\bar{\mathbf{m}}_u^t\big) = \mathbf{1}^\top \mathbf{R}\big(\mathbf{s}_u^{t-}\big) + \mathbf{1}^\top \mathbf{R}\big(\mathbf{s}_v^{t-}\big) + \mathbf{1}^\top \mathbf{R}(\mathbf{x}_{e_k}) + \mathbf{1}^\top \mathbf{R}(t - t_k) \tag{29}
$$

Where $\text{diag}(\cdot)$ denotes the diagonalization operator that maps a vector to a diagonal matrix, $-$ represents element-wise division and $\cdot$ denotes matrix multiplication. $\mathbf{R}(\mathbf{s}_u^{t-}) \in \mathbb{R}^{d_m \times d_m}$ and $R(\bar{\mathbf{m}}_u^t) \in \mathbb{R}^{(2d_m + d_e + d_t) \times d_m}$. Splitting $\mathbf{R}(\bar{\mathbf{m}}_u^t)$ column-wise: the first $d_m$ columns correspond to $\mathbf{R}(\mathbf{s}_u^{t-}) \in \mathbb{R}^{d_m \times d_m}$, the next $d_m$ to $2d_m$ columns to $\mathbf{R}(\mathbf{s}_u^{t-}) \in \mathbb{R}^{d_m \times d_m}$, the following $d_e$ columns to $\mathbf{R}(\mathbf{x}_{e_k}) \in \mathbb{R}^{d_e \times d_m}$, and the remaining columns to $\mathbf{R}(t - t_k) \in \mathbb{R}^{d_t \times d_m}$. $\qquad\square$

### A.6. Derivation of LRP Applied to RNN function

If the $f_{\text{update}}$ is the RNN function, the Eq. (3) can be rewritten as

$$
\mathbf{s}_u^t = \tanh\big(\bar{\mathbf{m}}_u^t \mathbf{W}_h + \mathbf{s}_u^{t-} \mathbf{U}_h + \mathbf{b}_h\big),
$$

where $\mathbf{W}_h, \mathbf{U}_h, \mathbf{b}_h$ are the parameters in the RNN.

Similarly,

$$
\mathbf{R}\big(\bar{\mathbf{m}}_u^t\big) = \text{diag}\big(\bar{\mathbf{m}}_u^t\big) \cdot \mathbf{W}_h \cdot \text{diag}\big(\bar{\mathbf{m}}_u^t \mathbf{W}_h + \mathbf{s}_u^{t-} \mathbf{U}_h\big)^{-1} \cdot \mathbf{R}\big(\mathbf{s}_u^t\big),
$$

$$
\mathbf{R}\big(\mathbf{s}_u^{t-}\big) = \text{diag}\big(\mathbf{s}_u^{t-}\big) \cdot \mathbf{U}_h \cdot \text{diag}\big(\bar{\mathbf{m}}_u^t \mathbf{W}_h + \mathbf{s}_u^{t-} \mathbf{U}_h\big)^{-1} \cdot \mathbf{R}\big(\mathbf{s}_u^t\big),
$$

$$
\mathbf{R}\big(\bar{\mathbf{m}}_u^t\big) = \begin{bmatrix} \mathbf{R}\big(\mathbf{s}_v^{t-}\big) \\ \mathbf{R}\big(\mathbf{s}_u^{t-}\big) \\ \mathbf{R}(\mathbf{x}_{e_k}) \\ \mathbf{R}(\phi(t - t_k)) \end{bmatrix}. \tag{30}
$$

Where $\text{diag}(\cdot)$ denotes the diagonalization operator that maps a vector to a diagonal matrix, $-$ represents element-wise division and $\cdot$ denotes matrix multiplication. $\mathbf{R}(\mathbf{s}_u^{t-}) \in \mathbb{R}^{d_m \times d_m}$ and $R(\bar{\mathbf{m}}_u^t) \in \mathbb{R}^{(2d_m+d_e+d_t) \times d_m}$. $\mathbf{R}(\mathbf{s}_v^{t-}) \in \mathbb{R}^{d_m \times d_m}$, $\mathbf{R}(\mathbf{s}_u^{t-}) \in \mathbb{R}^{d_m \times d_m}$, $\mathbf{R}(\mathbf{x}_{e_k}) \in \mathbb{R}^{d_e \times d_m}$, and $\mathbf{R}(\phi(t-t_k)) \in \mathbb{R}^{d_t \times d_m}$ denote the contribution of $\mathbf{s}_v^{t-}$, $\mathbf{s}_u^{t-}$, $\mathbf{x}_{e_k}$ and $\phi(t-t_k)$ to $\mathbf{z}_u^t$, respectively.

## A.7. The derivation of selecting important events

The derivation process of Eq. (14) is

$$\text{KL}(p_2 \parallel p_1) = p_2 \log\left(\frac{p_2}{p_1}\right) + (1-p_2)\log\left(\frac{1-p_2}{1-p_1}\right) = \sigma(z_2)\left(z_2 - \text{sp}(z_2) - z_1 - \text{sp}(z_1)\right)$$
$$+(1-\sigma(z_2))\left(-\text{sp}(z_2)+\text{sp}(z_1)\right) = \sigma(z_2)(z_2 - z_1) - \text{sp}(z_2) + \text{sp}(z_1), \tag{31}$$

In Eq. (14), we derived the KL divergence between $p_2$ and $p_1$, where $p_1$ and $p_2$ are arbitrary probabilities obtained by applying the sigmoid function to logits. Now we let $p_2$ be the predicted probability on the original dynamic graph, $p_2 = \hat{y}_{v,u}(\mathcal{E}(t)) = \sigma(z_2)$, and $p_1$ be the predicted probability after retaining only a subset of critical events and removing the others. Based on the event-contribution matrix $\mathbf{C}^t$, we approximate the perturbed logit $z_1$ as, $z_1 = \sum_{l=1}^{d_c} d_l \mathbf{C}_{e_l}^t$, where $d_c$ is the number of candidate events, and $\mathbf{d} \in \{0,1\}^{d_c}$ is a selection vector whose entry $d_l$ indicates whether event $e_l$ is selected. Substituting $z_1$ into Eq. (14) gives

$$\text{KL}(p_2 \parallel p_1) = \sigma(z_2)(z_2 - z_1) - \text{sp}(z_2) + \text{sp}(z_1)$$
$$= \sigma(z_2)z_2 - \hat{y}_{v,u}\left(\mathcal{E}(t)\right) \sum_{l=1}^{d_c} d_l \mathbf{C}_{e_l}^t - \text{sp}(z_2) + \text{sp}(\sum_{l=1}^{d_c} d_l \mathbf{C}_{e_l}^t)$$

Since $\sigma(z_2)z_2$ and $\text{sp}(z_2)$ do not depend on $\mathbf{d}$, they are constant with respect to the selection and can be dropped from the optimization objective. Minimizing the remaining KL term over $\mathbf{d}$ yields Eq. (15), which we use to select a small set of key events that best preserves the original prediction.

## A.8. The example of Lemma 4.1

Let $\mathbf{x} = [1,2,3]$, $\mathbf{W} = \begin{bmatrix} 1 & 2 & 3 \\ 4 & 5 & 6 \\ 7 & 8 & 9 \end{bmatrix}$, and $\mathbf{y} = [30, 36, 42]$, then

$$\mathbf{P} = \begin{bmatrix} 1 & 0 & 0 \\ 0 & 2 & 0 \\ 0 & 0 & 3 \end{bmatrix} \cdot \begin{bmatrix} 1 & 2 & 3 \\ 4 & 5 & 6 \\ 7 & 8 & 9 \end{bmatrix} \cdot \begin{bmatrix} \frac{1}{30} & 0 & 0 \\ 0 & \frac{1}{36} & 0 \\ 0 & 0 & \frac{1}{42} \end{bmatrix} \approx \begin{bmatrix} 0.03 & 0.05 & 0.07 \\ 0.27 & 0.28 & 0.28 \\ 0.70 & 0.67 & 0.65 \end{bmatrix}.$$

Because $\mathbf{1}^\top \mathbf{P} = \mathbf{1}^\top$, we have

$$\mathbf{1}^\top \cdot \mathbf{R}(\mathbf{x}) = \mathbf{1}^\top \cdot \mathbf{P} \cdot \mathbf{R}(\mathbf{y}) = \mathbf{1}^\top \cdot \mathbf{R}(\mathbf{y}).$$

## A.9. The example of Proposition 4.3

Following the GRU equations, we first decompose $\mathbf{R}(\mathbf{s}_u^t)$ into $\mathbf{R}(\mathbf{s}_u^{t-})$ and $\mathbf{R}(\tilde{\mathbf{s}}_u^t)$. Then, $\mathbf{R}(\tilde{\mathbf{s}}_u^t)$ is further decomposed into $\mathbf{R}(\mathbf{s}_u^{t-})$ and $\mathbf{R}(\bar{\mathbf{m}}_u^t)$. Finally, $\mathbf{R}(\bar{\mathbf{m}}_u^t)$ is attributed to $\mathbf{R}(\mathbf{s}_v^{t-})$, $\mathbf{R}(\mathbf{s}_u^{t-})$, $\mathbf{R}(t-t_k)$, and $\mathbf{R}(\mathbf{x}_{e_k})$. The overview of the decomposition is provided in Figure 7. We provide a toy running example when the $f_{\text{update}}$ is GRU function. Suppose that $\mathbf{s}_v^{t-} = [0.4, 0.2]$, $\mathbf{s}_u^{t-} = [0.1, 0.3]$, $\mathbf{x}_{e_k} = [0.5]$, and $t - t_k = [0.6]$. Then,

$$\bar{\mathbf{m}}_u^t = [\mathbf{s}_v^{t-} \| \mathbf{s}_u^{t-} \| \mathbf{x}_{e_k} \| t - t_k] = [0.4, 0.2, 0.1, 0.3, 0.5, 0.6].$$

The GRU parameters are

$$\mathbf{W}_r = \begin{bmatrix} 0.1 & 0 \\ 0 & 0.1 \\ 0.5 & 0 \\ 0 & 0.5 \\ 0.1 & 0 \\ 0 & 0.1 \end{bmatrix}, \quad \mathbf{W}_g = \begin{bmatrix} 0.2 & 0 \\ 0 & 0.1 \\ 0.1 & 0 \\ 0 & 0.1 \\ 0.1 & 0 \\ 0 & 0.2 \end{bmatrix}, \quad \mathbf{W}_h = \begin{bmatrix} 0.1 & 0 \\ 0 & 0.1 \\ 0.05 & 0 \\ 0 & 0.05 \\ 0.05 & 0 \\ 0 & 0.05 \end{bmatrix},$$

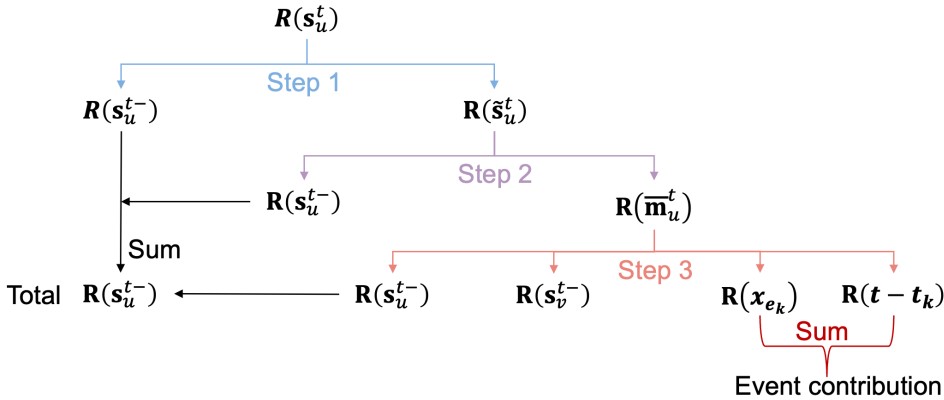

*Figure 7.* The overview of the LRP decomposition when the $f_{\text{update}}$ is GRU function

$$\mathbf{U}_r = \mathbf{U}_g = \mathbf{U}_h = \begin{bmatrix} 0.1 & 0 \\ 0 & 0.1 \end{bmatrix}, \quad \mathbf{b}_r = \mathbf{b}_g = \mathbf{b}_h = [0,0], \quad \mathbf{R}(\mathbf{s}_u^t) = \begin{bmatrix} 1 & 2 \\ 3 & 4 \end{bmatrix}.$$

**Forward propagation:**

$$\mathbf{r}_u^t = [0.526, 0.531], \quad \mathbf{g}_u^t = [0.537, 0.549], \quad \tilde{\mathbf{s}}_u^t = [0.075, 0.080], \quad \mathbf{s}_u^t = [0.086, 0.179].$$

**Calculation process:**

$$(1)\mathbf{P}_1 = \operatorname{diag}\left(\frac{(1 - \mathbf{g}_u^t) \odot \mathbf{s}_u^{t-}}{\mathbf{s}_u^t}\right) = \begin{bmatrix} 0.53 & 0 \\ 0 & 0.75 \end{bmatrix}, \mathbf{P}_2 = \operatorname{diag}\left(\frac{\mathbf{g}_u^t \odot \tilde{\mathbf{s}}_u^t}{\mathbf{s}_u^t}\right) = \begin{bmatrix} 0.47 & 0 \\ 0 & 0.25 \end{bmatrix}, \quad \mathbf{1}^\top \mathbf{P}_1 + \mathbf{1}^\top \mathbf{P}_2 = [1, 1].$$

$$\mathbf{R}(\mathbf{s}_u^{t-}) = \mathbf{P}_1 \cdot \mathbf{R}(\mathbf{s}_u^t) = \begin{bmatrix} 0.53 & 1.06 \\ 2.25 & 3 \end{bmatrix}, \mathbf{R}(\tilde{\mathbf{s}}_u^t) = \mathbf{P}_2 \cdot \mathbf{R}(\mathbf{s}_u^t) = \begin{bmatrix} 0.47 & 0.94 \\ 0.75 & 1 \end{bmatrix}, \mathbf{1}^\top \mathbf{R}(\mathbf{s}_u^t) = \mathbf{1}^\top \mathbf{R}(\mathbf{s}_u^{t-}) + \mathbf{1}^\top \mathbf{R}(\tilde{\mathbf{s}}_u^t).$$

$$(2)\mathbf{P}_3 = \operatorname{diag}(\bar{\mathbf{m}}_u^t) \cdot \mathbf{W}_h \cdot \operatorname{diag}(\bar{\mathbf{m}}_u^t \mathbf{W}_h + \mathbf{r}_u^t \odot (\mathbf{s}_u^{t-} \mathbf{U}_h))^{-1} = \begin{bmatrix} 0.53 & 0 \\ 0 & 0.25 \\ 0.07 & 0 \\ 0 & 0.18 \\ 0.33 & 0 \\ 0 & 0.37 \end{bmatrix},$$

$$\mathbf{P}_4 = \operatorname{diag}(\mathbf{r}_u^t \odot \mathbf{s}_u^{t-}) \cdot \mathbf{U}_h \cdot \operatorname{diag}(\bar{\mathbf{m}}_u^t \mathbf{W}_h + \mathbf{r}_u^t \odot (\mathbf{s}_u^{t-} \mathbf{U}_h))^{-1} = \begin{bmatrix} 0.07 & 0 \\ 0 & 0.2 \end{bmatrix}, \quad \mathbf{1}^\top \mathbf{P}_3 + \mathbf{1}^\top \mathbf{P}_4 = [1, 1].$$

$$\mathbf{R}(\bar{\mathbf{m}}_u^t) = \mathbf{P}_3 \cdot \mathbf{R}(\tilde{\mathbf{s}}_u^t) = \begin{bmatrix} 0.25 & 0.5 \\ 0.18 & 0.25 \\ 0.03 & 0.06 \\ 0.14 & 0.19 \\ 0.16 & 0.31 \\ 0.28 & 0.37 \end{bmatrix}, \mathbf{R}(\mathbf{s}_u^{t-}) = \mathbf{P}_4 \cdot \mathbf{R}(\tilde{\mathbf{s}}_u^t) = \begin{bmatrix} 0.03 & 0.06 \\ 0.15 & 0.2 \end{bmatrix}, \mathbf{1}^\top \mathbf{R}(\tilde{\mathbf{s}}_u^t) = \mathbf{1}^\top \mathbf{R}(\bar{\mathbf{m}}_u^t) + \mathbf{1}^\top \mathbf{R}(\mathbf{s}_u^{t-}).$$

(3) $\mathbf{R}(\bar{\mathbf{m}}_u^t)(i, j)$ represents the contribution of the $i$-th element in $\bar{\mathbf{m}}_u^t$ to the $j$-th column element of logits.

Thus,

$$\mathbf{R}(\mathbf{s}_v^{t-}) = \begin{bmatrix} 0.25 & 0.5 \\ 0.18 & 0.25 \end{bmatrix}, \quad \mathbf{R}(\mathbf{s}_u^{t-}) = \begin{bmatrix} 0.03 & 0.06 \\ 0.14 & 0.19 \end{bmatrix},$$

$$\mathbf{R}(\mathbf{x}_{e_k}) = [0.16, 0.31], \quad \mathbf{R}(t - t_k) = [0.28, 0.37].$$

The event contribution is

$$\mathbf{1}^\top \mathbf{R}(\mathbf{x}_{e_k}) + \mathbf{1}^\top \mathbf{R}(t - t_k) = 1.12.$$

The total contribution of $\mathbf{s}_u^{t-}$ is

$$\mathbf{R}(\mathbf{s}_u^{t-}) = \begin{bmatrix} 0.53 & 1.06 \\ 2.25 & 3 \end{bmatrix} + \begin{bmatrix} 0.03 & 0.06 \\ 0.15 & 0.2 \end{bmatrix} + \begin{bmatrix} 0.03 & 0.06 \\ 0.14 & 0.19 \end{bmatrix} = \begin{bmatrix} 0.59 & 1.18 \\ 2.54 & 3.39 \end{bmatrix},$$

and

$$\mathbf{1}^\top \mathbf{R}(\mathbf{s}_u^{t-}) + \mathbf{1}^\top \mathbf{R}(\mathbf{s}_v^{t-}) + \mathbf{1}^\top \mathbf{R}(\mathbf{x}_{e_k}) + \mathbf{1}^\top \mathbf{R}(t - t_k) = \mathbf{1}^\top \mathbf{R}(\mathbf{s}_u^t).$$

### A.10. The Algorithm for constructing the topology attribution tree

Algorithm 2 demonstrates the construction of a topology attribution tree and the propagation of relevance contributions.

---

**Algorithm 2** Construction of the topology attribution tree and relevance propagation

---

1: **Input**: target node $u$, node embedding $\mathbf{z}_u^t$, number of layers $L$
2: Initialize $\mathcal{T}_{\text{top}} \leftarrow \{(u, L)\}$ as a tree with root node, set frontier $\leftarrow \{(u, L)\}$.
3: Initialize contribution matrices: $\{\mathbf{M}_{p_0 \to u}^t \in \mathbb{R}^{d_m \times d_m} \leftarrow 0, |p_0 \in \mathcal{V}(t)\}$
4: Initialize contribution matrices: $\mathbf{C}_u^t \in \mathbb{R}^{|\mathcal{E}(t)| \times d_m} \leftarrow 0$ for all events
5: $p_l$ is the node at the layer $l$ and $p_{l+1}$ is the node at the layer $l + 1$.
6: next $\leftarrow \emptyset$
7: **for** $l = L - 1$ down to 0 **do**
8:     **for** each tuple $(p_{l+1}, l+1) \in$ frontier **do**
9:         **for** each event $e_l = (p_l, p_{l+1}, t_l) \in \mathcal{N}_{p_{l+1}}^n[0, t]$ **do**
10:             Insert $(p_l, l)$ and $(\mathbf{x}_{e_l}, l)$ as children of $(p_{l+1}, l+1)$
11:             next $\leftarrow$ next $\cup \{(p_l, l)\}$
12:             **if** $f_{\text{emb}}$ is graph sum function **then**
13:                 Compute $\mathbf{R}(\mathbf{h}_{p_l}^l)$, $\mathbf{R}\left(\mathbf{h}_{p_{l+1}}^{l+1}\right)$, $\mathbf{R}(\mathbf{x}_{e_l})$, and $\mathbf{R}(\phi)$ via Eq. (12)
14:             **else if** $f_{\text{emb}}$ is graph attention function **then**
15:                 Compute $\mathbf{R}\left(\mathbf{h}_{p_{l+1}}^{l+1}\right)$, $\mathbf{R}(\mathbf{h}_{p_l}^l)$, $\mathbf{R}(\mathbf{x}_{e_l})$, and $\mathbf{R}(\phi)$ via Eq. (26)
16:             **end if**
17:             $\mathbf{C}_u^t[e_l] \leftarrow \mathbf{C}_u^t[e_l] + \mathbf{1}^\top \mathbf{R}(\mathbf{x}_{e_l}) + \mathbf{1}^\top \mathbf{R}(\phi(t - t_l))$
18:             **if** $l = 0$ **then**
19:                 $\mathbf{M}_{p_0 \to u}^t \leftarrow \mathbf{M}_{p_0 \to u}^t + \mathbf{R}(\mathbf{h}_{p_0}^0)$
20:             **end if**
21:         **end for**
22:     **end for**
23:     frontier $\leftarrow$ next
24:     next $\leftarrow \emptyset$
25: **end for**
26: **Output**: $\mathcal{T}_{\text{top}}(u)$, the contribution matrix $\mathbf{C}_u^t$, and $\{\mathbf{M}_{p_0 \to u}^t | p_0 \text{ is a leaf node of } \mathcal{T}_{\text{top}}(u)\}$

---

### A.11. Evaluation metrics

$\text{Fidelity}_{\text{KL}} = \text{KL}(\hat{y}_u(\mathcal{E}(t)) \| \hat{y}_u(\mathcal{E}^*))$ (node property prediction) and $\text{Fidelity}_{\text{KL}} = \text{KL}(\hat{y}(\mathcal{E}(t)) \| \hat{y}(\mathcal{E}^*))$ (graph classification task) $\text{Fidelity}_{\text{prob}}$: $\text{Fidelity}_{\text{prob}} = |\hat{y}_{v,u}(\mathcal{E}(t)) - \hat{y}_{v,u}(\mathcal{E}^*)|$ (link prediction), $\text{Fidelity}_{\text{prob}} = |\hat{y}_u(\mathcal{E}(t)) - \hat{y}_u(\mathcal{E}^*)|$ (node property prediction) and $\text{Fidelity}_{\text{prob}} = |\hat{y}(\mathcal{E}(t)) - \hat{y}(\mathcal{E}^*)|$ (graph classification task) as defined in (Yuan et al., 2022).

## A.12. The selection for node property prediction task.

Similarly, for the node property prediction, the predicted probability is $\hat{\mathbf{y}} = \hat{\mathbf{y}}_u = \text{softmax}(f_{\text{mlp}}(\mathbf{z}_u^t))$. Let $\mathbf{z}$ denote the logits, and the contribution matrix of events is $\mathbf{C}_u^t$. The KL divergence is

$$\text{KL}(\hat{\mathbf{y}}(\mathcal{E}_1(t))\|\hat{\mathbf{y}}(\mathcal{E}_2(t))) = \sum_{k=1}^{c} \hat{y}_k(\mathcal{E}_2(t)) \log\Big[\frac{\hat{y}_k(\mathcal{E}_2(t))}{\hat{y}_k(\mathcal{E}_1(t))}\Big]$$

$$= \sum_{k=1}^{c} \hat{y}_k(\mathcal{E}_2(t))[z_k(\mathcal{E}_2(t)) - z_k(\mathcal{E}_1(t))] - \log(Z) = \sum_{k=1}^{c} \hat{y}_k(\mathcal{E}_2(t))\Delta z_k - \log(Z), \tag{32}$$

where $Z(\mathcal{E}_\tau(t)) = \sum_{k=1}^{c} \exp(z_k(\mathcal{E}_\tau(t)))$ for $\tau = 1, 2$, and $Z = \frac{Z(\mathcal{E}_2(t))}{Z(\mathcal{E}_1(t))}$. The objective function for node property prediction is:

$$\mathbf{d}^* = \underset{\mathbf{d}\in\{0,1\}^{d_c}, \|\mathbf{d}\|_1=n}{\arg\min} \sum_{k=1}^{c} \left( -\hat{y}_k(\mathcal{E}(t)) \sum_{l=1}^{|d_c|} d_l \mathbf{C}_{e_l}^t \right) + \log \sum_{k'=1}^{c} \exp \left( \sum_{l=1}^{|d_c|} d_l \mathbf{C}_{e_l}^t \right) \tag{33}$$

Solving this equation yields the most important events for the node property prediction task. The Algorithm shows the overall process of selecting important layer edges for node property prediction task.

---

**Algorithm 3** The overall framework for node property prediction

1: **Input**: target node $u$, time $t$, and the max depth of memory backtracking tree $T_L$
2: Compute $\mathcal{T}_{\text{top}}(u)$, $\{\mathbf{M}_{p_0 \to u}^t | p_0 \text{ is a leaf node of } \mathcal{T}_{\text{top}}(u)\}$ and $\mathbf{C}_u^t$ using Algorithm 2.      ▷ topology attribution
3: **for** each $\mathbf{M}_{p \to u}^t \in \{\mathbf{M}_{p_0 \to u}^t | p_0 \text{ is a leaf node of } \mathcal{T}_{\text{top}}(u)\}$ **do**
4:      Update $\mathbf{C}_u^t$ using Algorithm 5      ▷ Memory attribution
5: **end for**
6: Update $\mathbf{C}_u^t$ using Lemma 4.1, and let $\mathbf{C}_u^t \to \mathbf{C}^t$
7: Select important events $\mathcal{E}^*$ using Eq. (33)
8: **Output:** The important events $\mathcal{E}^*$

---

## A.13. The selection for graph classification prediction task.

For the graph classification task, the predicted probability is $\hat{\mathbf{y}} = [\hat{y}_1, \cdots, \hat{y}_c]$. The contribution matrix of events to node $u$ is $\mathbf{C}^t(u)$. The objective function for node property prediction is:

$$\mathbf{d}^* = \underset{\mathbf{d}\in\{0,1\}^{d_c}, \|\mathbf{d}\|_1=n}{\arg\min} \sum_{k=1}^{c} \left( -\hat{y}_k(\mathcal{E}(t)) \sum_{u\in\mathcal{V}(t)} \sum_{l=1}^{|d_c|} d_l \mathbf{C}_{e_l}^t(u) \right) + \log \sum_{k'=1}^{c} \exp \left( \sum_{u\in\mathcal{V}(t)} \sum_{l=1}^{|d_c|} d_l \mathbf{C}_{e_l}^t(u) \right) \tag{34}$$

Solving this equation yields the most important events for the graph classification task. The Algorithm shows the overall process of selecting important layer edges for graph classification task.

---

**Algorithm 4** The overall framework for graph classification task

1: **Input**: the node set $\mathcal{V}(t)$, time $t$, and the max depth of memory backtracking tree $T_L$
2: **for** $u \in \mathcal{V}(t)$ **do**
3:      Compute $\mathcal{T}_{\text{top}}(u)$, $\{\mathbf{M}_{p_0 \to u}^t | p_0 \text{ is a leaf node of } \mathcal{T}_{\text{top}}(u)\}$ and $\mathbf{C}_u^t$ using Algorithm 2.    ▷ topology attribution
4:      **for** each $\mathbf{M}_{p \to u}^t \in \{\mathbf{M}_{p_0 \to u}^t | p_0 \text{ is a leaf node of } \mathcal{T}_{\text{top}}(u)\}$ **do**
5:          Update $\mathbf{C}_u^t$ using Algorithm 5      ▷ Memory attribution
6:          $\mathbf{C}^t(u) = \mathbf{C}_u^t$
7:      **end for**
8: **end for**
9: Update $\mathbf{C}^t(u)$ using Lemma 4.1
10: Select important events $\mathcal{E}^*$ using Eq. (34)
11: **Output:** The important events $\mathcal{E}^*$

---

## A.14. The Algorithm for constructing the memory backtracking tree

---

**Algorithm 5** Construction of the memory backtracking tree and recursive relevance propagation

---

1: **Input**: target node $u$, node memory contributions $\mathbf{M}_{u \to u}^t \in \mathbb{R}^{d_m \times d_m}$, time $t$, $\mathbf{C}_{u_k}^t \in \mathbb{R}^{|\mathcal{E}(t)| \times d_m}$, and the max depth $T_L$
2: Initialize tree $\mathcal{T}_{\text{memory}} \leftarrow \{(u,t)\}$ with root $(u,t)$
3: **procedure DFS** (node: $u$, timestamp: $t$, layer: $T_l$, contribution matrix: $\mathbf{R}(\mathbf{s}_u^t)$)
4:     **if** $T_l \geq T_L$ **then**
5:         **return**
6:     **end if**
7:     Let $\mathcal{H}(u,t) = \{e_1, \ldots, e_m\}$ sorted by $t_1 < \cdots < t_m \leq t$
8:     **if** $\mathcal{H}(u,t) = \emptyset$ **then**
9:         **return**
10:    **end if**
11:    **for** $j = m$ **down to** $1$ **do**
12:       $e_j = (v_j, u, t_j)$
13:       Insert $(v_j, t_j)$, $(u, t_j)$ and $(e_j, t_j)$ as children of $(u,t)$ in $\mathcal{T}_{\text{memory}}$
14:       **if** $f_{\text{update}}$ is the GRU function **then**
15:          Compute $\mathbf{R}\left(\mathbf{s}_{v_j}^{t_j -}\right), \mathbf{R}\left(\mathbf{s}_u^{t_j -}\right), \mathbf{R}(\mathbf{x}_{e_j})$, and $\mathbf{R}(t - t_j)$ via Eq. (29)
16:       **else if** $f_{\text{update}}$ is the RNN function **then**
17:          Compute $\mathbf{R}\left(\mathbf{s}_{v_j}^{t_j -}\right), \mathbf{R}\left(\mathbf{s}_u^{t_j -}\right), \mathbf{R}(\mathbf{x}_{e_j})$, and $\mathbf{R}(t - t_j)$ via Eq. (30)
18:       **end if**
19:       $\mathbf{C}_u^t[e_j] \leftarrow \mathbf{C}_u^t[e_j] + \mathbf{1}^\top \mathbf{R}(\mathbf{x}_{e_j}) + \mathbf{1}^\top \mathbf{R}(\phi(t - t_j))$
20:       **procedure DFS** $\left(\text{node: } v_j, \text{timestamp: } t_j, \text{layer: } T_l + 1, \text{contribution matrix: } \mathbf{R}(\mathbf{s}_{v_j}^{t_j -})\right)$
21:       **procedure DFS** $\left(\text{node: } u, \text{timestamp: } t_j, \text{layer: } T_l + 1, \text{contribution matrix: } \mathbf{R}(\mathbf{s}_u^{t_j -})\right)$
22:    **end for**
23: **end procedure**
24: **procedure DFS** (node: $u$, timestamp: $t$, layer: $0$, contribution matrix: $\mathbf{M}_{u \to u}^t$)
25: **Output**: $\mathcal{T}_{\text{memory}}(u, t)$, contributions $\mathbf{C}_u^t$

---

## A.15. Dataset

We select several real-world temporal graph datasets for link prediction. These datasets cover a wide range of real-world applications and domains, including social networks, political networks, communication networks, etc. The brief introduction of the six datasets is listed as follows. Data statistics are given in Table 2

- Wikipedia (Kumar et al., 2019): A one-month interaction network with nodes as editors and pages, and edges as timestamped posting requests.

- Reddit (Kumar et al., 2019): Records one-month activity in subreddits, with nodes as users/posts and edges as timestamped posts. Edge features are 172-dimensional LIWC vectors from edit texts.

- Enron (Shetty & Adibi, 2004): Email interaction network of ENRON employees over three years, with no attributes.

- UCI (Panzarasa et al., 2009): Unattributed social network among UCI students, tracking online forum posts with second-level timestamps.

- tgbn-trade (Huang et al., 2023): International agriculture trade network between UN nations (1986-2016), with edges representing annual trade values.

- tgbn-genre (Huang et al., 2023): Bipartite network between users and music genres, with edge weights denoting genre preferences.

- tgbn-reddit (Huang et al., 2023): Interaction network between users and subreddits (2005-2019), with edges representing posts.

- HMDB51 (Kuehne et al., 2011): a large benchmark for human action recognition with 51 action categories and 6,766 annotated video clips collected from movies and web videos. We selected four categories: climbing, sit-ups, running, and pull-up, as the classification task.

- Penn Action (Zhang et al., 2013): a large-scale collection designed for human action recognition in video. It contains over 2,300 video clips with 15 different action categories, including running, jumping, and kicking. Each video is annotated with action labels, start and end times, and multiple frames per action.

*Table 2.* The details of datasets.

| Datasets | Domains | Nodes | Links |
|---|---|---|---|
| Wikipedia | Social | 9,227 | 157,474 |
| Reddit | Social | 10,984 | 672,447 |
| Enron | Communication | 184 | 125,235 |
| UCI | Social | 1,899 | 59,835 |
| tgbn-trade | trade | 255 | 468,245 |
| tgbn-genre | interact. | 1,505 | 17,858,395 |
| tgbn-reddit | social | 11,766 | 27,174,118 |

## A.16. Running time

We partition the computation time into three components: **topology time**, for constructing the topology attribution tree and computing neighbor contributions; **memory time**, for constructing the memory backtracking tree and computing contributions of historical events; and **selection time**, for solving Eq. (15) or Eq. (33). Figure 8 reports the average computation time on all datasets. Topology time is omitted for the enron, UCI, tgbn-trade, tgbn-genre and tgbn-reddit datasets because it accounts for less than 1% of the total runtime and is therefore not shown in the stacked bars.

For link prediction, **memory time** is the dominant cost. It increases with the memory depth $T$ but remains practical: when $T = 10$, constructing the memory backtracking tree and computing historical contributions takes at most 5 seconds. For node property prediction, **selection time** dominates because the output is higher-dimensional than in link prediction (where the logit is one-dimensional), making the final solving step more expensive. Even so, when $T = 10$, event selection takes at most 3 seconds, which is still acceptable.

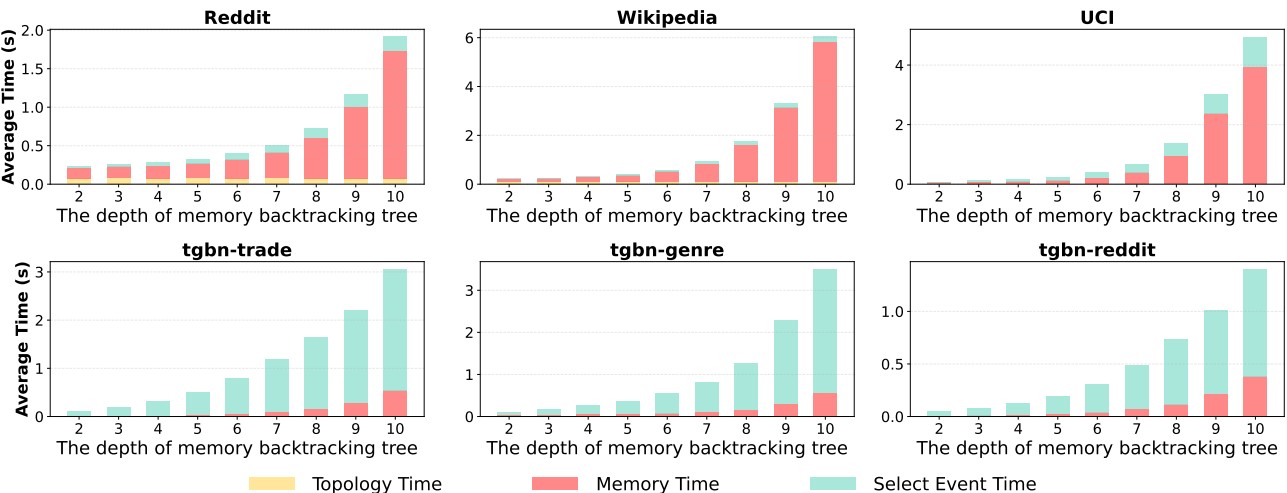

*Figure 8.* Running time decomposition: each figure represents a dataset. The x-axis represents the depth of the memory backtracking tree, and the y-axis represents the running time.

### A.17. The average and standard deviation of Fidelity$_{prob}$ and Fidelity$_{KL}$.

Tables 3 and 4 report the mean and standard deviation of Fidelity$_{KL}$ and Fidelity$_{prob}$ corresponding to Figures 5 and 6. We further performed a t-test between our method and the second-best baseline, and found statistically significant in 77% of the cases for Fidelity$_{KL}$ and in 74% of the cases for Fidelity$_{prob}$.

*Table 3.* The average and standard deviation of Fidelity$_{KL}$ under different sparsity level. $*$ indicates that our results are significantly different from the runner-up baseline under a t-test.

| Dataset | Ratio | MemExplainer | TempME | TGNNExplainer | GNNExplainer | PGExplainer | w/o memory | w/o topology | w/o selection |
|---|---|---|---|---|---|---|---|---|---|
| Wikipedia | 0.02 | **0.059**±0.113* | 0.103±0.216 | 0.120±0.236 | 0.146±0.294 | 0.110±0.193 | 0.079±0.132 | 0.088±0.162 | 0.104±0.234 |
| | 0.04 | **0.056**±0.104* | 0.095±0.186 | 0.124±0.235 | 0.141±0.280 | 0.105±0.188 | 0.079±0.142 | 0.083±0.168 | 0.098±0.239 |
| | 0.06 | **0.058**±0.114 | 0.090±0.182 | 0.128±0.270 | 0.142±0.307 | 0.105±0.192 | 0.077±0.142 | 0.070±0.142 | 0.094±0.236 |
| | 0.08 | **0.059**±0.116 | 0.081±0.158 | 0.119±0.265 | 0.150±0.313 | 0.107±0.189 | 0.072±0.133 | 0.070±0.143 | 0.091±0.226 |
| | 0.10 | **0.060**±0.119 | 0.077±0.149 | 0.123±0.251 | 0.146±0.337 | 0.120±0.237 | 0.072±0.142 | 0.068±0.140 | 0.088±0.221 |
| Reddit | 0.02 | **0.164**±0.257* | 0.341±0.501 | 0.494±0.553 | 0.436±0.534 | 0.534±0.570 | 0.359±0.439 | 0.583±0.628 | 0.285±0.378 |
| | 0.04 | **0.158**±0.252* | 0.325±0.516 | 0.472±0.574 | 0.428±0.549 | 0.506±0.560 | 0.315±0.414 | 0.801±0.675 | 0.242±0.353 |
| | 0.06 | **0.150**±0.258* | 0.316±0.584 | 0.445±0.555 | 0.418±0.547 | 0.474±0.542 | 0.303±0.407 | 0.942±0.665 | 0.219±0.331 |
| | 0.08 | **0.139**±0.253* | 0.297±0.586 | 0.425±0.529 | 0.391±0.512 | 0.458±0.536 | 0.303±0.446 | 0.942±0.665 | 0.193±0.276 |
| | 0.10 | **0.132**±0.260* | 0.264±0.509 | 0.397±0.480 | 0.387±0.527 | 0.439±0.529 | 0.291±0.410 | 0.942±0.665 | 0.176±0.286 |
| UCI | 0.02 | **0.060**±0.095* | 0.083±0.116 | 0.095±0.135 | 0.095±0.132 | 0.097±0.125 | 0.080±0.102 | 0.071±0.115 | 0.072±0.105 |
| | 0.04 | **0.049**±0.082* | 0.077±0.117 | 0.096±0.141 | 0.097±0.143 | 0.094±0.135 | 0.073±0.098 | 0.075±0.114 | 0.059±0.101 |
| | 0.06 | **0.044**±0.078 | 0.069±0.104 | 0.095±0.146 | 0.095±0.150 | 0.095±0.148 | 0.069±0.095 | 0.084±0.105 | 0.052±0.095 |
| | 0.08 | **0.041**±0.079 | 0.066±0.102 | 0.092±0.143 | 0.094±0.149 | 0.093±0.144 | 0.070±0.096 | 0.089±0.109 | 0.048±0.097 |
| | 0.10 | **0.039**±0.076 | 0.064±0.095 | 0.093±0.145 | 0.092±0.147 | 0.094±0.150 | 0.074±0.096 | 0.095±0.108 | 0.041±0.081 |
| Enron | 0.02 | **0.073**±0.160* | 0.138±0.230 | 0.204±0.218 | 0.187±0.220 | 0.201±0.213 | 0.177±0.225 | 0.120±0.212 | 0.146±0.224 |
| | 0.04 | **0.071**±0.164* | 0.122±0.225 | 0.196±0.225 | 0.183±0.219 | 0.201±0.215 | 0.177±0.227 | 0.101±0.212 | 0.129±0.228 |
| | 0.06 | **0.070**±0.140* | 0.111±0.223 | 0.185±0.225 | 0.183±0.226 | 0.196±0.235 | 0.177±0.245 | 0.085±0.188 | 0.107±0.228 |
| | 0.08 | **0.065**±0.137* | 0.105±0.229 | 0.183±0.228 | 0.181±0.228 | 0.195±0.236 | 0.180±0.245 | 0.078±0.214 | 0.101±0.260 |
| | 0.10 | **0.070**±0.143 | 0.100±0.227 | 0.180±0.230 | 0.179±0.228 | 0.197±0.237 | 0.184±0.251 | 0.076±0.215 | 0.099±0.261 |
| tgbn-trade | 0.02 | **0.153**±0.063 | 0.165±0.052 | 0.203±0.071 | 0.198±0.067 | 0.197±0.065 | 0.155±0.059 | 0.171±0.066 | 0.181±0.063 |
| | 0.04 | **0.143**±0.055* | 0.161±0.049 | 0.203±0.071 | 0.198±0.067 | 0.197±0.065 | 0.148±0.055 | 0.165±0.065 | 0.177±0.063 |
| | 0.06 | **0.137**±0.053* | 0.158±0.046 | 0.203±0.071 | 0.197±0.066 | 0.196±0.064 | 0.142±0.055 | 0.158±0.060 | 0.173±0.061 |
| | 0.08 | **0.130**±0.048* | 0.155±0.044 | 0.203±0.071 | 0.196±0.064 | 0.195±0.062 | 0.137±0.056 | 0.151±0.055 | 0.169±0.061 |
| | 0.10 | **0.125**±0.046* | 0.152±0.044 | 0.203±0.071 | 0.196±0.063 | 0.193±0.059 | 0.131±0.057 | 0.144±0.055 | 0.166±0.061 |
| tgbn-genre | 0.02 | **0.638**±0.133* | 0.705±0.095 | 0.745±0.103 | 0.751±0.099 | 0.760±0.119 | 0.714±0.122 | 0.671±0.114 | 0.729±0.118 |
| | 0.04 | **0.614**±0.132* | 0.687±0.101 | 0.751±0.103 | 0.751±0.097 | 0.754±0.119 | 0.709±0.123 | 0.656±0.120 | 0.721±0.124 |
| | 0.06 | **0.604**±0.133* | 0.678±0.104 | 0.744±0.104 | 0.750±0.096 | 0.752±0.117 | 0.706±0.125 | 0.645±0.121 | 0.705±0.129 |
| | 0.08 | **0.589**±0.128* | 0.666±0.107 | 0.738±0.107 | 0.749±0.094 | 0.752±0.117 | 0.694±0.126 | 0.628±0.121 | 0.700±0.129 |
| | 0.10 | **0.577**±0.125* | 0.659±0.106 | 0.735±0.107 | 0.748±0.094 | 0.755±0.119 | 0.690±0.127 | 0.618±0.118 | 0.694±0.130 |
| tgbn-reddit | 0.02 | **0.682**±0.414* | 1.090±0.727 | 1.732±0.686 | 1.535±0.674 | 1.872±0.769 | 0.850±0.676 | 0.858±0.610 | 0.979±0.525 |
| | 0.04 | **0.565**±0.368* | 0.967±0.714 | 1.722±0.721 | 1.693±0.721 | 1.871±0.766 | 0.773±0.659 | 0.734±0.577 | 0.912±0.491 |
| | 0.06 | **0.470**±0.325* | 0.868±0.703 | 1.768±0.729 | 1.708±0.719 | 1.869±0.761 | 0.719±0.663 | 0.635±0.536 | 0.835±0.454 |
| | 0.08 | **0.406**±0.291* | 0.788±0.695 | 1.825±0.755 | 1.722±0.722 | 1.834±0.745 | 0.672±0.651 | 0.548±0.504 | 0.799±0.447 |
| | 0.10 | **0.341**±0.254* | 0.726±0.693 | 1.766±0.773 | 1.767±0.727 | 1.806±0.731 | 0.647±0.646 | 0.477±0.457 | 0.748±0.434 |

### A.18. The performance of different $f_{emb}$ and $f_{update}$ in TGNs

In Figure 9 and Figure 10, we show the performance of Fidelity$_{prob}$ and Fidelity$_{KL}$ when the $f_{emb}$ is graph attention model. In Figure 11 and Figure 12, we report Fidelity$_{prob}$ and Fidelity$_{KL}$ for the setting where the memory updater $f_{update}$ is implemented as an RNN. Our method, MemExplainer, significantly outperform baseline explainers across all datasets for both metrics.

### A.19. Sensitivity analysis

In Figure 13 and Figure 14, we report Fidelity$_{prob}$ and Fidelity$_{KL}$ when the number of events from neighboring samples is fixed to $n = 20$. Our method, MemExplainer, outperforms all baseline methods across all datasets on both metrics. We conduct a sensitivity analysis on the maximum depth of the memory backtracking tree. On the Enron, UCI, tgbn-trade, and tgbn-genre datasets, we set the maximum depth of the memory backtracking tree from 2 to 10, selecting the same number of events for each target node or edge across all depths. The performance of FidelityKL and Fidelityprob is shown in Figures 15 and 16, respectively. As shown in these figures, in tgbn-trade and tgbn-genre datasets, Fidelity$_{KL}$ and Fidelity$_{prob}$

*Table 4.* The average and standard deviation of Fidelity$_{prob}$ under different sparsity level. ∗ indicates that our results are significantly different from the runner-up baseline under a t-test.

| Dataset | Ratio | MemExplainer | TempME | TGNNExplainer | GNNExplainer | PGExplainer | w/o memory | w/o topology | w/o selection |
|---|---|---|---|---|---|---|---|---|---|
| Wikipedia | 0.02 | **0.085**±0.108* | 0.122±0.144 | 0.131±0.150 | 0.144±0.164 | 0.129±0.142 | 0.106±0.114 | 0.110±0.127 | 0.116±0.147 |
| | 0.04 | **0.084**±0.105* | 0.118±0.140 | 0.133±0.149 | 0.141±0.157 | 0.123±0.139 | 0.104±0.112 | 0.105±0.128 | 0.108±0.144 |
| | 0.06 | **0.083**±0.106 | 0.113±0.136 | 0.130±0.153 | 0.138±0.160 | 0.121±0.140 | 0.101±0.114 | 0.094±0.119 | 0.106±0.141 |
| | 0.08 | **0.083**±0.106 | 0.107±0.129 | 0.123±0.147 | 0.139±0.164 | 0.123±0.141 | 0.098±0.108 | 0.091±0.122 | 0.103±0.139 |
| | 0.10 | **0.084**±0.108 | 0.103±0.125 | 0.124±0.147 | 0.136±0.157 | 0.126±0.147 | 0.096±0.109 | 0.090±0.121 | 0.100±0.137 |
| Reddit | 0.02 | **0.181**±0.173* | 0.279±0.228 | 0.354±0.249 | 0.321±0.249 | 0.373±0.256 | 0.293±0.242 | 0.379±0.285 | 0.260±0.215 |
| | 0.04 | **0.175**±0.171* | 0.264±0.229 | 0.335±0.252 | 0.312±0.252 | 0.357±0.255 | 0.268±0.231 | 0.474±0.300 | 0.232±0.203 |
| | 0.06 | **0.169**±0.167* | 0.247±0.228 | 0.320±0.254 | 0.305±0.250 | 0.339±0.256 | 0.256±0.231 | 0.539±0.285 | 0.218±0.192 |
| | 0.08 | **0.159**±0.162* | 0.232±0.224 | 0.310±0.253 | 0.292±0.248 | 0.330±0.256 | 0.248±0.234 | 0.539±0.285 | 0.204±0.183 |
| | 0.10 | **0.152**±0.158* | 0.219±0.212 | 0.299±0.245 | 0.287±0.249 | 0.319±0.255 | 0.245±0.233 | 0.539±0.285 | 0.188±0.181 |
| Enron | 0.02 | **0.115**±0.127* | 0.192±0.133 | 0.252±0.141 | 0.234±0.144 | 0.250±0.141 | 0.226±0.151 | 0.180±0.121 | 0.203±0.133 |
| | 0.04 | **0.111**±0.124* | 0.173±0.134 | 0.242±0.146 | 0.229±0.146 | 0.248±0.145 | 0.223±0.154 | 0.151±0.123 | 0.178±0.138 |
| | 0.06 | **0.117**±0.117* | 0.159±0.131 | 0.231±0.147 | 0.226±0.152 | 0.240±0.149 | 0.219±0.159 | 0.135±0.116 | 0.150±0.137 |
| | 0.08 | **0.104**±0.118* | 0.149±0.133 | 0.226±0.153 | 0.223±0.154 | 0.239±0.150 | 0.220±0.161 | 0.119±0.121 | 0.134±0.142 |
| | 0.10 | **0.111**±0.121 | 0.141±0.134 | 0.220±0.156 | 0.220±0.156 | 0.238±0.154 | 0.218±0.169 | 0.116±0.124 | 0.128±0.142 |
| UCI | 0.02 | **0.100**±0.101* | 0.127±0.116 | 0.135±0.121 | 0.136±0.122 | 0.140±0.120 | 0.125±0.104 | 0.111±0.112 | 0.115±0.108 |
| | 0.04 | **0.086**±0.091* | 0.119±0.117 | 0.133±0.126 | 0.133±0.128 | 0.133±0.123 | 0.115±0.102 | 0.117±0.114 | 0.096±0.103 |
| | 0.06 | **0.081**±0.087 | 0.109±0.112 | 0.129±0.127 | 0.128±0.129 | 0.129±0.127 | 0.110±0.101 | 0.130±0.113 | 0.087±0.097 |
| | 0.08 | **0.075**±0.085 | 0.105±0.111 | 0.125±0.125 | 0.126±0.129 | 0.126±0.127 | 0.110±0.102 | 0.137±0.114 | 0.079±0.096 |
| | 0.10 | **0.073**±0.083 | 0.103±0.109 | 0.126±0.127 | 0.124±0.127 | 0.125±0.128 | 0.117±0.105 | 0.145±0.111 | 0.073±0.087 |
| tgbn-trade | 0.02 | **0.072**±0.042 | 0.074±0.043 | 0.077±0.047 | 0.076±0.046 | 0.076±0.045 | 0.073±0.043 | 0.075±0.042 | 0.077±0.045 |
| | 0.04 | **0.070**±0.038* | 0.073±0.042 | 0.077±0.047 | 0.076±0.046 | 0.076±0.045 | 0.072±0.043 | 0.074±0.041 | 0.076±0.045 |
| | 0.06 | **0.069**±0.037* | 0.072±0.041 | 0.077±0.047 | 0.076±0.045 | 0.076±0.045 | 0.071±0.043 | 0.072±0.040 | 0.076±0.044 |
| | 0.08 | **0.067**±0.035* | 0.072±0.040 | 0.077±0.047 | 0.076±0.045 | 0.076±0.044 | 0.070±0.043 | 0.071±0.037 | 0.075±0.044 |
| | 0.10 | **0.066**±0.034* | 0.071±0.040 | 0.077±0.047 | 0.076±0.045 | 0.076±0.044 | 0.069±0.043 | 0.070±0.037 | 0.075±0.044 |
| tgbn-genre | 0.02 | **0.115**±0.017* | 0.118±0.017 | 0.119±0.017 | 0.119±0.017 | 0.120±0.018 | 0.118±0.018 | 0.117±0.017 | 0.119±0.018 |
| | 0.04 | **0.114**±0.017* | 0.117±0.016 | 0.119±0.017 | 0.119±0.017 | 0.119±0.018 | 0.118±0.018 | 0.116±0.018 | 0.118±0.018 |
| | 0.06 | **0.113**±0.017* | 0.117±0.016 | 0.119±0.017 | 0.119±0.017 | 0.119±0.018 | 0.118±0.018 | 0.115±0.018 | 0.118±0.018 |
| | 0.08 | **0.113**±0.016* | 0.116±0.016 | 0.119±0.017 | 0.119±0.017 | 0.119±0.018 | 0.117±0.018 | 0.115±0.018 | 0.118±0.018 |
| | 0.10 | **0.112**±0.016* | 0.116±0.016 | 0.118±0.017 | 0.119±0.017 | 0.119±0.018 | 0.117±0.018 | 0.114±0.018 | 0.117±0.018 |
| tgbn-reddit | 0.02 | **0.245**±0.088 | 0.295±0.108 | 0.374±0.073 | 0.357±0.078 | 0.380±0.074 | 0.255±0.114 | 0.265±0.103 | 0.296±0.085 |
| | 0.04 | **0.221**±0.086* | 0.276±0.113 | 0.371±0.076 | 0.368±0.078 | 0.380±0.073 | 0.239±0.118 | 0.243±0.103 | 0.287±0.084 |
| | 0.06 | **0.199**±0.082* | 0.257±0.116 | 0.374±0.076 | 0.370±0.077 | 0.380±0.074 | 0.227±0.121 | 0.224±0.103 | 0.276±0.084 |
| | 0.08 | **0.183**±0.078* | 0.241±0.120 | 0.377±0.075 | 0.371±0.076 | 0.379±0.074 | 0.216±0.123 | 0.204±0.103 | 0.270±0.085 |
| | 0.10 | **0.165**±0.073* | 0.226±0.123 | 0.371±0.078 | 0.375±0.075 | 0.377±0.074 | 0.210±0.123 | 0.189±0.100 | 0.261±0.085 |

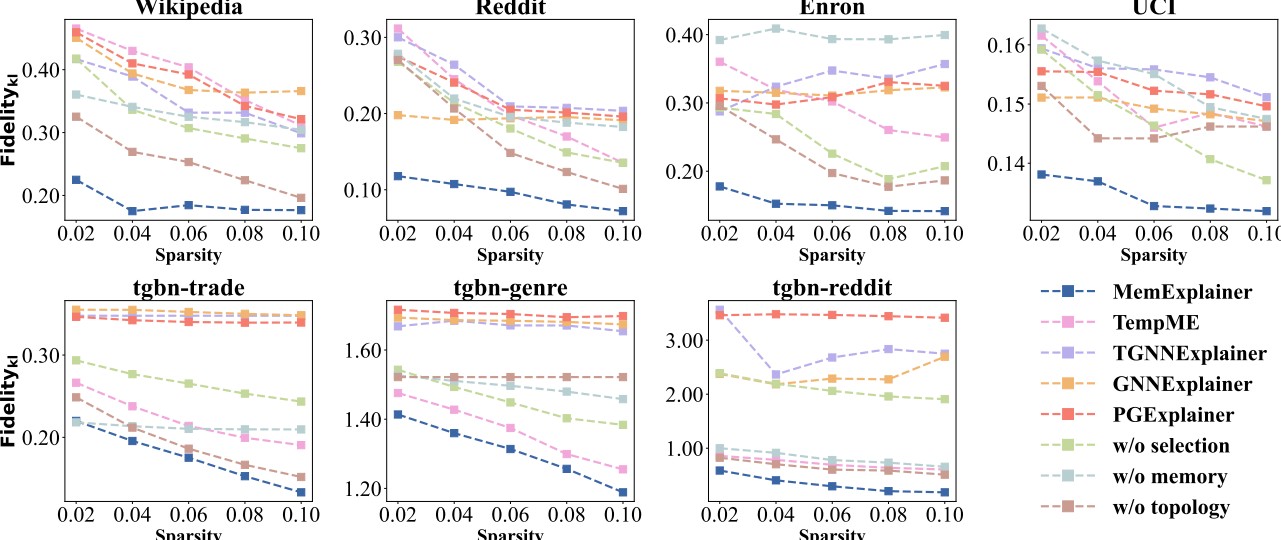

*Figure 9.* The performance of Fidelity$_{KL}$. Each figure corresponds to a different dataset. First and second rows represent link prediction and node property prediction, respectively. Lower value indicates better performance. The $f_{emb}$ is graph attention model.

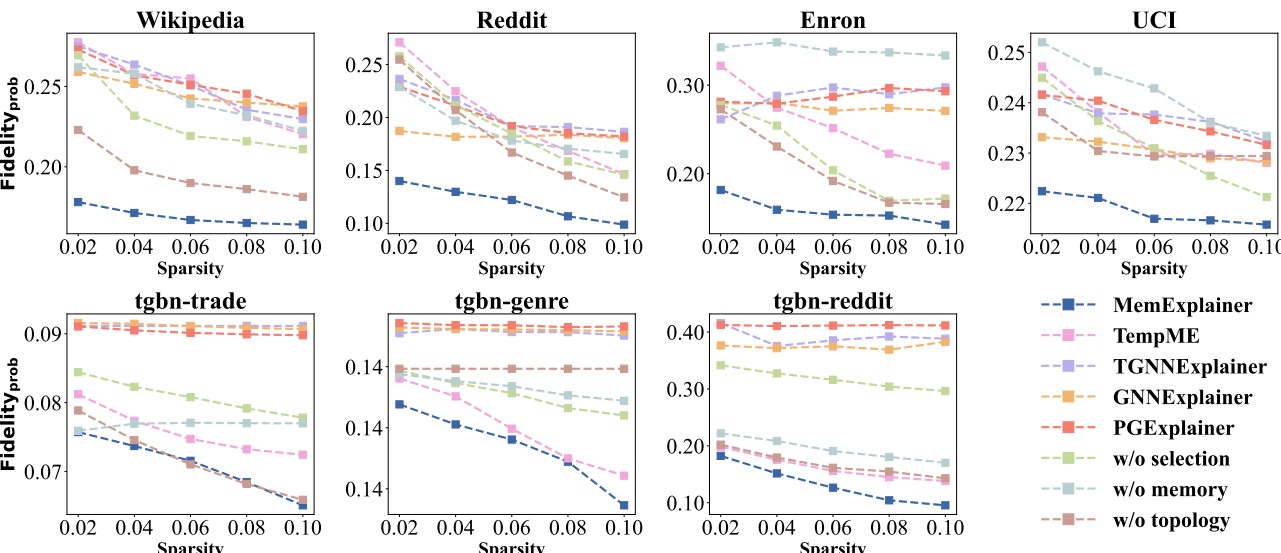

*Figure 10.* The performance of Fidelity$_{prob}$. Each figure corresponds to a different dataset. First and second rows represent link prediction and node property prediction, respectively. Lower value indicates better performance. The $f_{emb}$ is graph attention model.

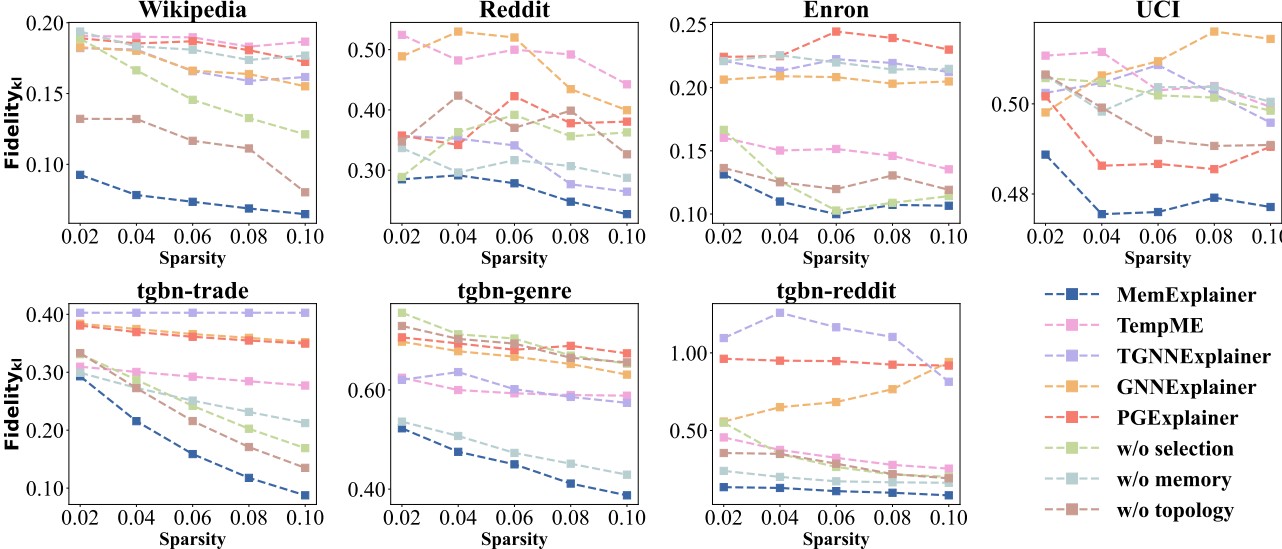

*Figure 11.* The performance of Fidelity$_{KL}$. Each figure corresponds to a different dataset. First and second rows represent link prediction and node property prediction, respectively. Lower value indicates better performance. The $f_{update}$ is RNN model.

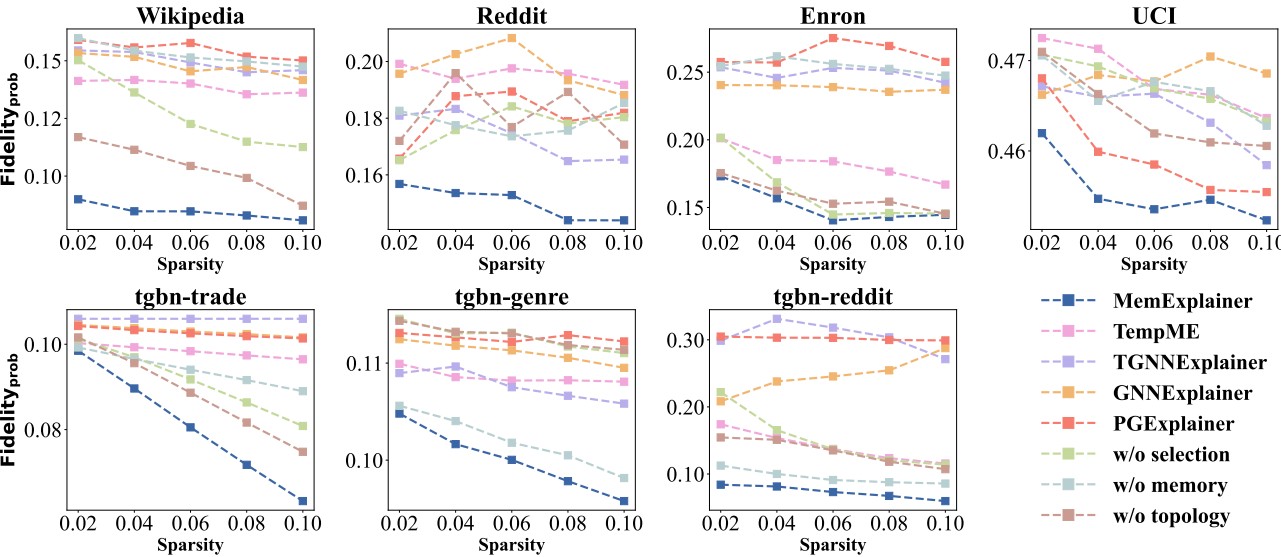

*Figure 12.* The performance of Fidelity$_{prob}$. Each figure corresponds to a different dataset. First and second rows represent link prediction and node property prediction, respectively. Lower value indicates better performance. The $f_{update}$ is RNN model.

initially decreases and then increases with increasing maximum depth. This trend occurs because deeper backtracking allows for more accurate tracing of historical events, resulting in more faithful explanations. However, this also increases the number of candidate explanation events, making the optimization problem more complex. The expanded search space complicates the identification of relevant events, leading to the decrease in fidelity. While on the Enron and UCI datasets, the lower depth can achieve better performance. The best performing on different datasets is different, which is due to the different characteristics of the datasets. The optimal maximum depth varies across different datasets, which can be attributed to the unique characteristics of each dataset.

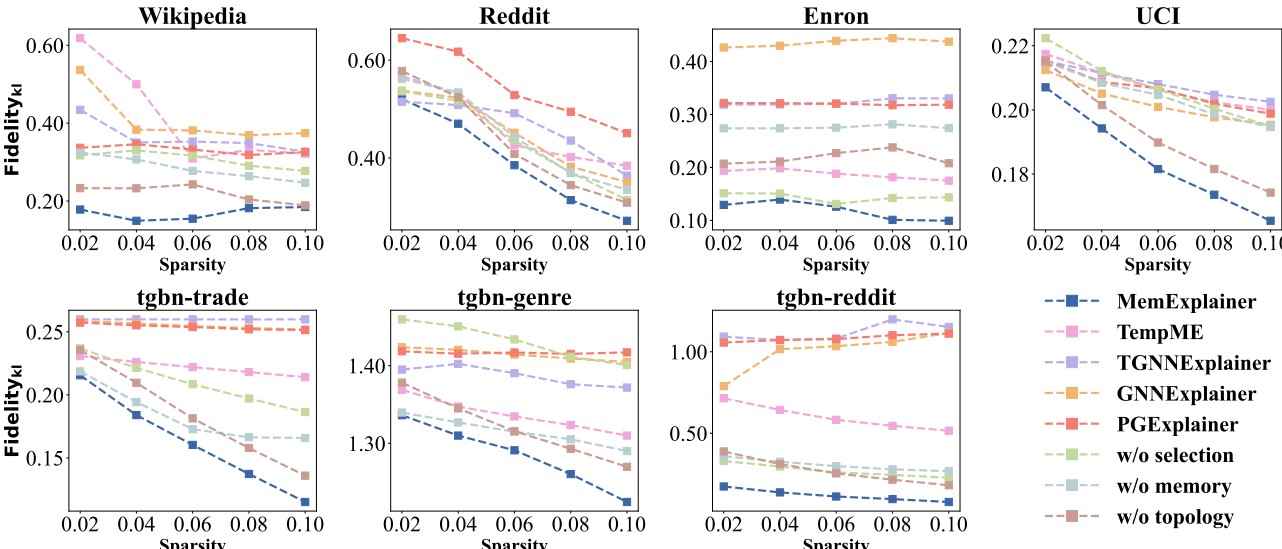

*Figure 13.* The performance of Fidelity$_{KL}$. Each figure corresponds to a different dataset. First and second rows represent link prediction and node property prediction, respectively. Lower value indicates better performance. The number of events from neighboring samples $n$ is 20.

We conduct a sensitivity analysis on the maximum depth of the memory backtracking tree. On the Enron, UCI, tgbn-trade, and tgbn-genre datasets, we set the maximum depth of the memory backtracking tree from 2 to 10, selecting the same number of events for each target node or edge across all depths. The performance of FidelityKL and Fidelityprob is shown in

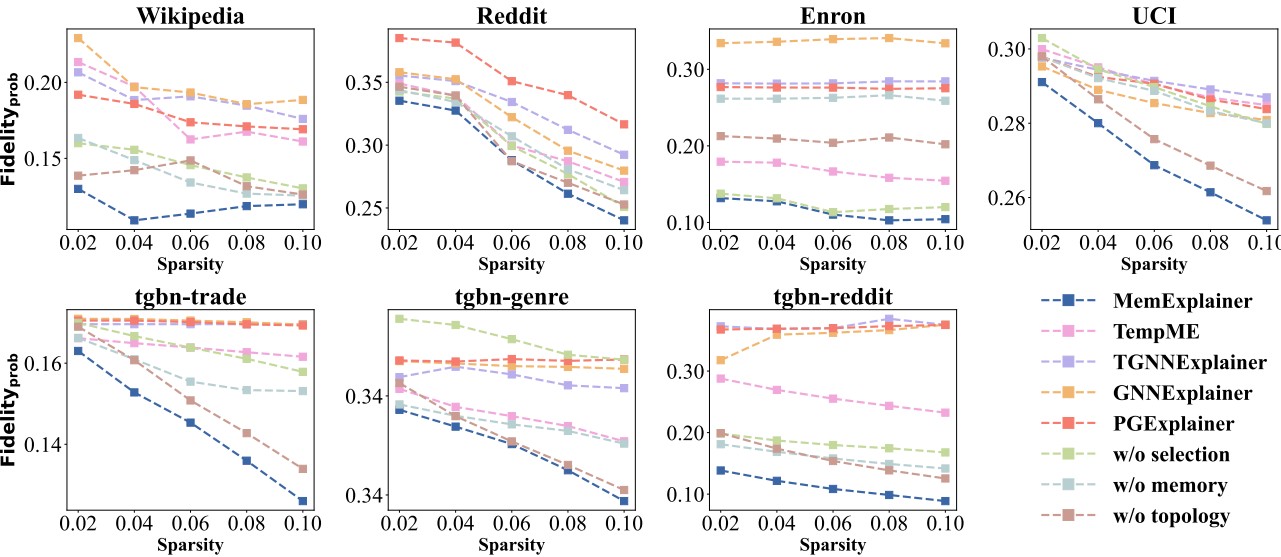

*Figure 14.* The performance of Fidelity$_{prob}$. Each figure corresponds to a different dataset. First and second rows represent link prediction and node property prediction, respectively. Lower value indicates better performance. The number of events from neighboring samples $n$ is 20.

Figures 15 and 16, respectively. As shown in these figures, in tgbn-trade and tgbn-genre datasets, Fidelity$_{KL}$ and Fidelity$_{prob}$ initially decreases and then increases with increasing maximum depth. This trend occurs because deeper backtracking allows for more accurate tracing of historical events, resulting in more faithful explanations. However, this also increases the number of candidate explanation events, making the optimization problem more complex. The expanded search space complicates the identification of relevant events, leading to the decrease in fidelity. While on the Enron and UCI datasets, the lower depth can achieve better performance. The best performing on different datasets is different, which is due to the different characteristics of the datasets. The optimal maximum depth varies across different datasets, which can be attributed to the unique characteristics of each dataset.

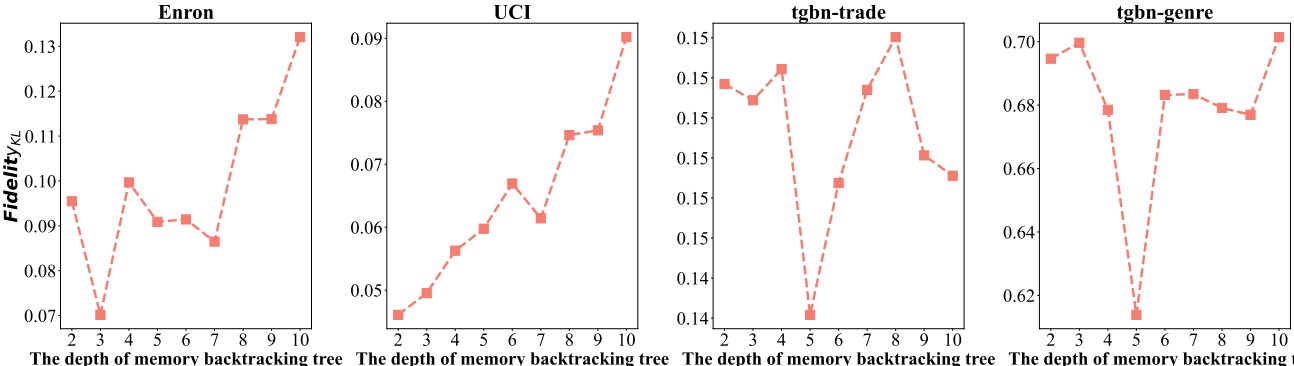

*Figure 15.* Performance of Fidelity$_{KL}$ at different maximum depths of the memory backtracking tree, with each figure representing a different dataset.

## A.20. Case study

We show the explanation results on pose-based action classification task. Table 8, Table 5, Table 6, and Table 7 show case studies for the pull-up, run, climb, and sit-up classes. For a fair comparison, all explanation methods operate on the same input sequence of skeleton graphs for each video. We also fix a **global edge budget**, i.e., the total number of key human-pose edges that each method can select. Since different methods select different explanation edges, these edges may appear on different frames, meaning the explanation results **may not** be frame-aligned across methods.

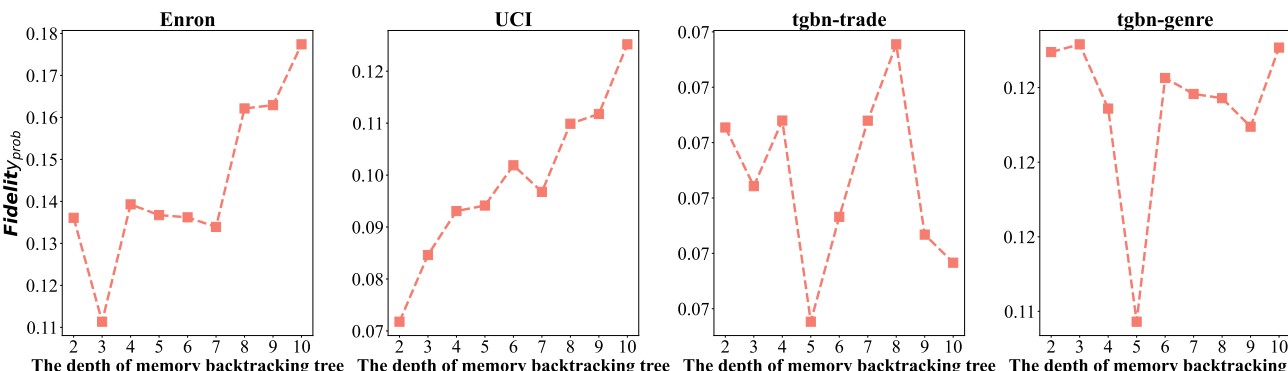

*Figure 16.* Performance of Fidelity$_{prob}$ at different maximum depths of the memory backtracking tree, with each figure representing a different dataset.

Our method typically select a much smaller subset of edges per frame, which leads to more visualized frames under the same total edge budget. For pull-ups, the yellow edges focus on the main kinematic chains (e.g., shoulder–elbow–wrist and hip–knee) during both lifting and lowering. For running, they concentrate on the lower limb chain (hip–knee–ankle) that drives forward motion, showing when the discriminative gait pattern appears. For climbing, our method mainly selects the supporting arm and leg that hold the body and control the center of mass, instead of uniformly highlighting the whole skeleton. For sit-ups, the selected edges lie along the torso–hip–knee chain, capturing how the upper body bends around relatively fixed legs.

In contrast, across all four actions, PGExplainer, GNNExplainer, and TGNNExplainer usually select most skeletal links in one frame, making it hard to see which joints actually drive the prediction. Under a fixed edge budget, this behavior naturally yields fewer visualized frames. TempME often selects edges that are unstable over time and misaligned with the key biomechanics of the movement. The selected edges do not form a consistent pattern across frames.

*Table 5.* Visualization of explanation results for the pull-up action with different methods. Green nodes denote joint (bone) nodes, red edges denote original connections between joints, yellow edges denote the selected important edges, and each image corresponds to one frame in the video.

| Methods | Explanation results |
|---|---|

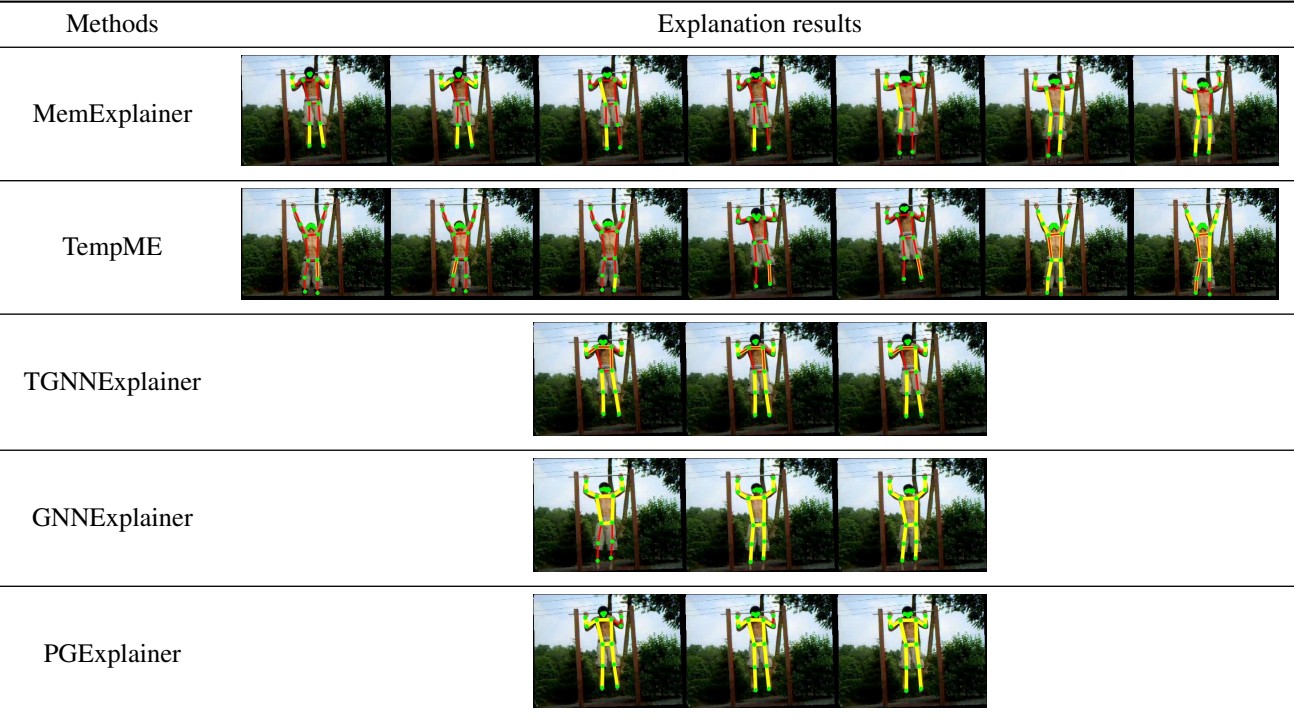

*Table 6.* Visualization of explanation results for the climbing action with different methods.Green nodes denote joint (bone) nodes, red edges denote original connections between joints, yellow edges denote the selected important edges, and each image corresponds to one frame in the video.

| Methods | Explanation results |
| --- | --- |
| MemExplainer |  |
| TempME |  |
| TGNNExplainer |  |
| GNNExplainer |  |
| PGExplainer |  |

*Table 7.* Visualization of explanation results for the sit-ups action with different methods.Green nodes denote joint (bone) nodes, red edges denote original connections between joints, yellow edges denote the selected important edges, and each image corresponds to one frame in the video.

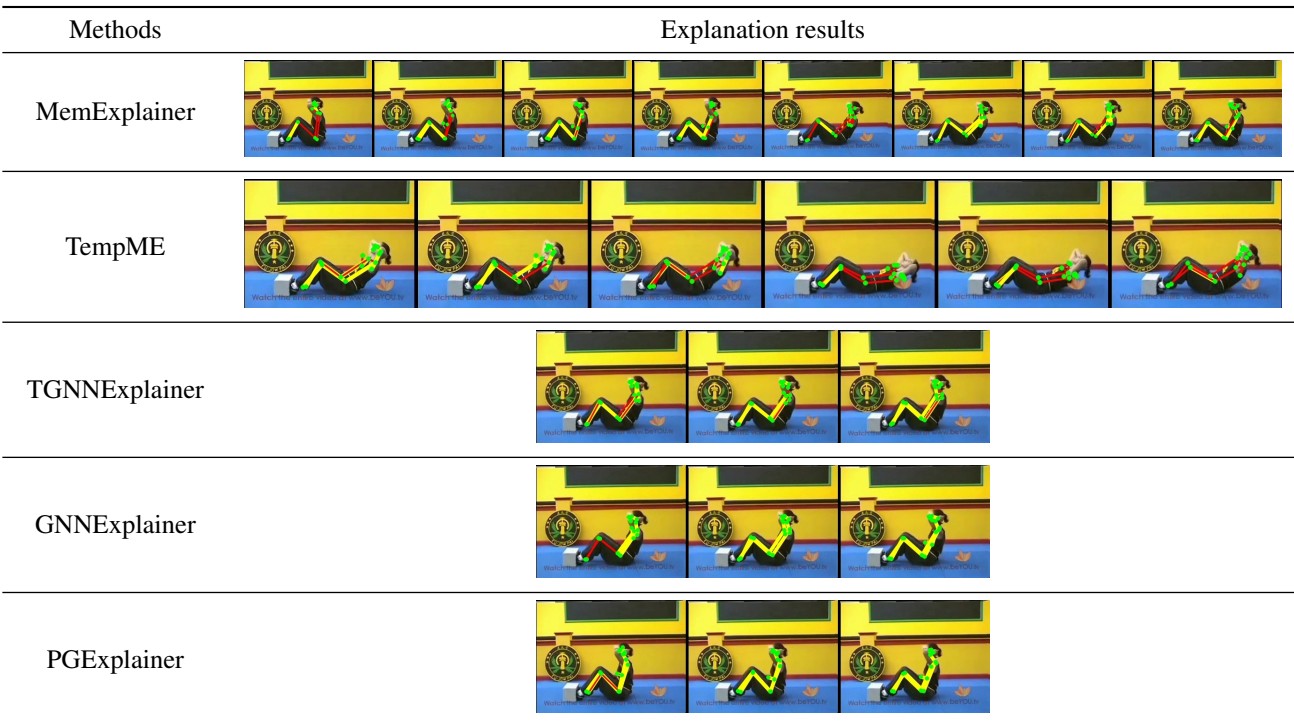

*Table 8.* Visualization of explanation results for the running action with different methods.Green nodes denote joint (bone) nodes, red edges denote connections between joints, yellow edges denote the selected important edges, and each image corresponds to one frame.

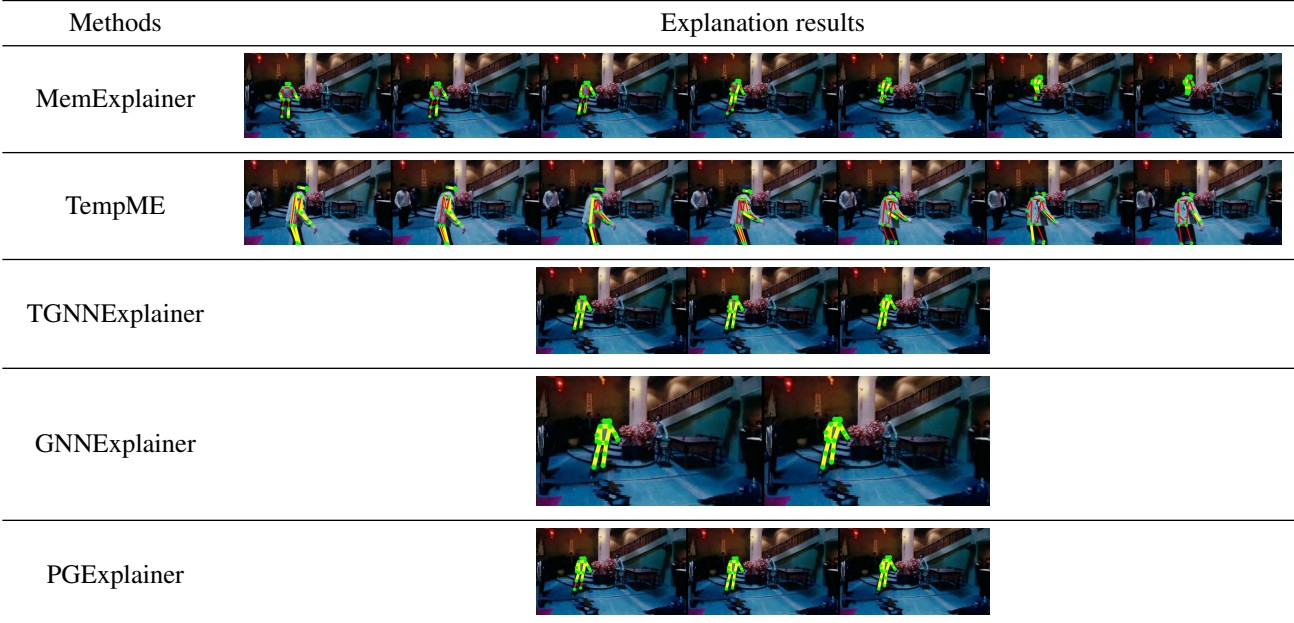

