# OpenReview forum: "Towards the Explainability of Temporal Graph Networks via Memory Backtracking and Topological Attribution"
_ICML.cc/2026/Conference — ICML 2026 spotlight_

### Official Review · Reviewer_uoJb · 2026-03-05

**Soundness:** 3
**Presentation:** 3
**Significance:** 3
**Originality:** 3
**Overall Recommendation:** 5
**Confidence:** 4

**Summary:**

This paper tackles a tricky problem in Temporal Graph Networks (TGNs): explaining exactly how the model's memory module drives its decisions. Even though memory is crucial for temporal reasoning, it’s usually overlooked in existing explanation methods. To solve this, the authors paired Layer-wise Relevance Propagation (LRP) with a clever dual-tree tracking system (using Topology Attribution and Memory Backtracking). What makes this approach stand out is its mathematical rigor—the relevance scores perfectly add up to match the model’s final output. They ran the framework through the wringer, testing it across nine datasets and three different tasks, and showed that it consistently outperforms current baselines.

**Compliance With Llm Reviewing Policy:**

Affirmed.

**Final Justification:**

My original rating is 4: weak accept, since some details of the proposed method and baselines are not clarified. After rebuttal, my concerns are basically resolved. Hence, I deicide to raise my rating.

**Key Questions For Authors:**

The memory backtracking concept could be extended to counterfactual explanations, which would be a natural and impactful next step. For example, what if this historical event had not occurred?

**Limitations:**

Limitations are not discussed (If I miss, feel free to point it out.).

**Strengths And Weaknesses:**

S1: The technical foundation of the paper is solid. The adoption of LRP for TGNs is well-motivated, and the conservation property provides a rigorous guarantee. Furthermore, this paper addresses a genuinely important problem: explaining why a temporal graph model made a prediction, with explicit focus on the memory component that encodes historical context.

S2: The experimental design is comprehensive in scope, covering diverse datasets from social, traffic, and financial fields, and different task types, which strengthens confidence in the method's general applicability. The improvement brought by the proposed MemExplainer is significant, compared with the SOTA temporal GNN explainers.

S2: The problem statement is crisp, and the distinction from static GNN explainers is clearly articulated. Key concepts (Memory Backtracking Tree, topology attribution) are defined with helpful formalism. Figures and tables are generally informative, with Table 8 providing a useful qualitative comparison of explanation quality.

- --
W1: The LRP propagation rules for non-linear memory update functions (e.g., GRU/LSTM gates) are not fully derived. While the authors reference standard LRP-ε and LRP-γ rules, the interaction with temporal memory states  are not formally bounded.

W2: While the breadth of evaluation is commendable, the paper lacks ablation studies on key design choices: (i) the maximum depth of the Memory Backtracking Tree, (ii) the sensitivity of the top-k selection hyper-parameter, and (iii) the impact of different TGN backbone architectures (e.g., TGN-Attn vs. TGN-GRU). Without these, it is difficult to isolate the contribution of the memory-backtracking component from other factors.

W3: The related work section could more explicitly discuss why LRP is preferable to Shapley-value-based methods for temporal attribution, e.g., computational efficiency and theoretical guarantees.

---

> ### Author Rebuttal · Authors · 2026-03-29
>
> Thank you for taking the time to review our paper!
>
> >**Q1:The LRP propagation rules for non-linear memory update functions are not fully derived. While the authors reference standard LRP-ε and LRP-γ rules, the interaction with temporal memory states are not formally bounded.**
>
> **A1**: Our LRP implementation for GRU follows [1]. The idea is the signal-take-all principle: the signal neuron receives all the relevance from the upper layer, while the gate neuron receives zero relevance. In our setting, the gate itself is not assigned relevance. This allocation method can also satisfy conservation. Empirically, even without gate relevance, our method selects more faithful historical events than the baselines.
>
> Besides, if one does not want the gate contribution to be fixed at zero, relevance can also be propagated into the gate via bilinear decomposition rules, as in [2]. For a bilinear term, the contribution of each factor is computed by treating the other as constant. The same idea can be applied to trace relevance into the gate. However, while incorporating gate relevance is possible, our current design is already theoretically sound and empirically outperforms the baselines.
>
> [1] Layer-wise relevance propagation for interpreting LSTM-RNN decisions in predictive maintenance.
>
> [2] Attnlrp: attention-aware layer-wise relevance propagation for transformers.
>
> >**Q2: the paper lacks ablation studies on key design choices.**
>
> **A2**:  **(i) maximum depth of the Memory Backtracking Tree**: we conduct a sensitivity analysis by varying the maximum depth from 2 to 10 while keeping the number of selected events for each target node or edge fixed across depths. The results are shown in **Figures 13 and 14** in Appendix. On tgbn-trade and tgbn-genre, both metrics first decrease and then increase as depth grows. Deeper backtracking traces more historical events, but also enlarges the search space, complicating event identification and reducing fidelity. On Enron and UCI, smaller depths work better. The optimal depth depends on dataset characteristics.
>
> **(ii) the sensitivity of the top-k selection hyper-parameter**: in the **Figures 4, 5, 7-12**, the x-axis represents the sparsity, defined as $\mathrm{Sparsity}=\frac{\lvert \mathcal{E}^* \rvert}{\lvert \mathcal{E}(t) \rvert}$, where $\lvert \mathcal{E}^*\rvert$ is the number of selected important events and and $\mathcal{E}(t)$ denotes the set of all events. Therefore, varying the sparsity is equivalent to varying the number of selected important events. The results show that our method outperforms the baselines across different sparsity levels.
>
> **(iii) the impact of different TGN architectures**: **Figures 7-8, 9-10, and 4-5** in the Appendix report results for TGN-Attn, TGN-RNN, and TGN-GRU, respectively. Across all backbone variants, our method significantly outperforms the baselines.
>
> >**Q3: The related work section could more explicitly discuss why LRP is preferable to Shapley-value-based methods for temporal attribution.**
>
> **A3**: Shapley-value-based methods are much more expensive in our setting. The Shapley-value-based method needs to consider all possible coalitions of the events, which is very time-consuming. In addition, computing the value function for each coalition requires a forward pass through the TGN model. This becomes prohibitive when the number of events is large. Approximate Shapley methods can reduce the cost, but their attribution scores generally do not satisfy the conservation property. In contrast, our event selection objective relies on conservation, and violating it may reduce explanation faithfulness. For these reasons, we choose LRP instead of Shapley-value-based methods.
>
> >**Q4: The memory backtracking concept could be extended to counterfactual explanations?**
>
> **A4**: One possible way to obtain counterfactual explanations is to compare the model’s predictions on the original temporal graph $G_0$ and a modified graph $G_1$ (e.g., after removing or perturbing some events). If the predictions differ, our method can be applied to both graphs to identify important historical events for each case. By comparing the relevance scores of historical events under $G_0$ and $G_1$, we can then identify the events that are most responsible for the prediction change. This comparison can be used to derive a counterfactual explanatory subgraph.
>
> >**Q5: Limitations are not discussed**
>
> **A5**: The limitation of our method is its computational cost on very large temporal graphs, where the memory backtracking tree can become deep and wide, increasing runtime and memory usage. A practical way is to limit the tree depth or prune branches with very small relevance scores. We will add this discussion to the limitation section.

---

> > ### Author Rebuttal · Reviewer_uoJb · 2026-04-03
> >
> > My questions are resolved and I am willing to raise my rating.

---

### Official Review · Reviewer_qhgd · 2026-03-05

**Soundness:** 3
**Presentation:** 2
**Significance:** 4
**Originality:** 2
**Overall Recommendation:** 5
**Confidence:** 3

**Summary:**

This paper studies explainability in Temporal Graph Networks (TGNs) and proposes a method called MemExplainer, which attributes model predictions to both neighboring events and historical events stored in node memories. The method introduces two key components: a topology attribution tree that decomposes the spatial contributions of neighboring events and node memories, and a memory backtracking tree that traces how historical events influence node memory updates over time. Experiments on nine temporal graph datasets across tasks including node property prediction, link prediction, and graph classification demonstrate improved fidelity metrics compared with several existing explainability baselines.

**Compliance With Llm Reviewing Policy:**

Affirmed.

**Final Justification:**

I'm happy to raise my rating from 4 to 5 since the author addressed my main concerns regarding mathematical formulas.

**Key Questions For Authors:**

* The memory backtracking tree traces historical events that contribute to node memories. For long temporal sequences with many events, this process may become computationally expensive. Could the authors clarify how the approach scales to large temporal graphs and whether pruning or approximation strategies are necessary?

* The method relies on Layer-wise Relevance Propagation to ensure conservation of contributions across layers. How sensitive are the explanations to the specific LRP formulation used, and would the approach remain effective if alternative attribution techniques such as Integrated Gradients or Shapley-based methods were used?

* Some notations are not very appropriate and appear before they are clearly defined, and the relationships between the proportional matrices $P(\cdot)$, contribution matrices $C^t$, and relevance terms $R(\cdot)$ are not always intuitive. In Eq.10, the Hadamard product of $x^\top \in \mathbb{R}^{d_1 \times 1}$ and $W \in \mathbb{R}^{d_1 \times d_2}$ does not seem to conform to the rules of operation. Could the authors clarify these definitions to improve the readability of the derivations?

**Limitations:**

Yes.

**Strengths And Weaknesses:**

## Strengths

* The paper addresses an important limitation of Temporal Graph Networks, namely their lack of interpretability regarding how historical events stored in memory influence predictions. By explicitly modeling both spatial interactions among neighboring events and temporal effects of historical memory updates, the proposed framework provides a structured approach to understanding TGN decision mechanisms.

* The proposed method attempts to trace the influence of historical events through memory updates using a memory backtracking tree. This design is conceptually appealing for temporal graph models where node memories accumulate information over time.

* The experimental section evaluates the method on several datasets covering different tasks such as link prediction, node classification, and graph classification. The results show improvements in explanation fidelity metrics compared with baseline explainers.

## Weaknesses

* The proposed framework introduces multiple components including topology attribution trees, memory backtracking trees, LRP-based contribution propagation, and an optimization procedure for event selection. The resulting pipeline is quite complex and may be difficult to implement or reproduce without extensive additional details.

*  The approach relies heavily on LRP and extends attribution to temporal memory updates. While the idea of tracing historical memory contributions is interesting, the methodological novelty mainly lies in adapting existing attribution techniques rather than introducing fundamentally new explanation principles.

* The notation is sometimes difficult to follow and not always consistently introduced before use. For example, the relationship between contribution matrices, proportional matrices, and the relevance propagation terms is not always clearly explained, and several intermediate symbols appear without intuitive interpretation. This makes the derivation of Eqs. (12)–(14) and the subsequent memory attribution steps harder to understand. A clearer notation table or a more consistent introduction of symbols would significantly improve readability.

---

> ### Author Rebuttal · Authors · 2026-03-29
>
> Thank you for taking the time to review our paper!
> >**Q1: The methodological novelty mainly lies in adapting existing attribution techniques rather than introducing new explanation principles.**
>
> **A1**: Tracing historical memory contributions in TGNs is more complex, and we propose the memory backtracking tree, which traces a node’s memory contribution from the prediction time back to historical events. In this sense, memory backtracking tree also serves as a new explanation principle. We further derive LRP for TGN, covering both the embedding module and the memory update process. While LRP provides the attribution foundation, the main novelty is the memory backtracking tree and the corresponding derivation that enables event tracing through the memory module.
>
> >**Q2: Would the approach remain effective if alternative attribution techniques such as Integrated Gradients or Shapley-based methods were used?**
>
> **A2**: **Shapley-value-based** methods are much more expensive in our setting because they must evaluate all possible coalitions of the events, and each evaluation requires a forward pass through the TGN model. This quickly becomes infeasible when the number of events is large. Approximate Shapley methods can reduce the cost, but their attribution scores usually do not satisfy the conservation property, which is essential for our event selection objective and explanation faithfulness. Similarly, **Integrated Gradients** is also less aligned with our objective, as it depends on a baseline choice and does not naturally enforce layer-wise conservation. In our setting, conservation is critical because the optimization objective for identifying important historical events relies on it.
>
> >**Q3: Could the authors clarify how the approach scales to large temporal graphs?**
>
> **A3**: In our framework, the memory backtracking tree is organized in a temporal manner: the root represents the contribution of the node’s memory vector at the prediction time, the first layer corresponds to the most recent events that directly updated this memory, and nodes closer to the leaves correspond to earlier historical events. Therefore, the tree depth directly controls how far backward in time the explanation traces from the prediction moment.
>
> For large temporal graphs, we can either **limit the depth** of the memory backtracking tree or use **pruning-based** approximations, such as keeping only the highest relevance scores of nodes or branches at each level during backtracking.
>
> >**Q4: The relationship between contribution matrices $C^t$, proportional matrices $P(\cdot)$, and the relevance propagation terms $R(\cdot)$ is not always clearly explained.**
>
> **A4**: Thank you for pointing this out, and we apologize for the lack of clarity in the current presentation.
>
> Consider one step in the topology attribution tree or the memory backtracking tree, where the parent node is a vector $y$ and a child node is a vector $x$, $R(y)$ denotes **the contribution of $y$ to the logits**, and $R(y)(i,j)$ denotes the contribution of the $i$-th element of $y$ to the $j$-th logit dimension. The proportional matrix $P$ describes **how contribution is redistributed from the parent node to the child node**, where $P(i,j)$ measures how much the $i$-th element of $x$ contributes to the $j$-th element of $y$.
> Then contribution of $x$ is computed as $R(x)=P \cdot R(y)$. The matrix $C^t$ denotes the **final event contribution matrix** obtained after all backward propagation steps are completed.
>
> From this perspective, the purpose of Eqs. (12)–(14) and Proposition 4.3 is to derive the proportional matrices $P$ for embedding module and the memory update module in TGN, so that relevance can be propagated layer by layer to obtain the final contribution matrix $C^t$.
>
> >**Q5: In Eq.10, the Hadamard product does not seem to conform to the rules of operation?**
>
> **A5**: We sincerely apologize for the confusion. For Eq. (10), we omitted the broadcasting operation in the notation, which made the Hadamard product appear dimensionally inconsistent. The correct expression is $P=\frac{\mathrm{broadcast}(x^\top)\odot W}{\mathrm{broadcast}(y)}$, the broadcasting operation expands $x^\top$ and $y$ to match the shape of W. To illustrate the meaning of Eq. (10), consider a simple example: supposing that x=[1,2,3], W=[[1,2,3],[4,5,6],[7,8,9]] y=xW=[30,36,42], $$P=\frac{\left[\begin{matrix}1&1&1\\\\2&2&2\\\\3&3&3\end{matrix}\right]\odot\left[\begin{matrix}1&2&3\\\\4&5&6\\\\7&8&9\end{matrix}\right]}{\left[\begin{matrix}30&36&42\\\\30&36&42\\\\30&36&42\end{matrix}\right]}=\left[\begin{matrix}0.03&0.05&0.07\\\\0.27&0.28&0.28\\\\0.7&0.67&0.65\end{matrix}\right]$$
> Each column of P sums to 1, which ensures conservation during relevance propagation. Therefore, $1^\top R(x)=1^\top P \cdot R(y)=1^\top R(y)$.
>
> >**Q6: The resulting pipeline may be difficult to implement or reproduce**
>
> **A6**: If the paper is accepted, we will release the full codebase to support reproducibility.

---

> > ### Author Rebuttal · Reviewer_qhgd · 2026-03-31
> >
> > Thank you for the clarifications in the response.
> >
> > However, regarding the mathematical notation, the use of a 'broadcasting operation' as a formal function in the equations still raises some concerns. I knew that 'broadcasting' is a widely understood engineering mechanism in frameworks like PyTorch (e.g., `torch.matmul`) or NumPy, but it is not a standard mathematical operator in linear algebra. To maintain the theoretical rigor of the manuscript, could the authors formalize this operation using standard mathematical notation? Alternatively, if a specific 'Broadcast' function must be used, please explicitly define its mathematical mapping in the text.
> >
> > As a minor follow-up regarding computational efficiency: although Appendix A.14 provides some running time results, a scalability comparison against other baseline models is still expected. Given the tight timeline right now, adding these comparison results to the revision would be sufficient.

---

> > > ### Author Response · Authors · 2026-04-02
> > >
> > > >**Q1: Could the authors formalize this operation using standard mathematical notation?**
> > >
> > > **A1**: Thank you for this helpful suggestion. we will replace the broadcasting-based expression with a standard linear algebra formulation in the manuscript. Specifically, let $x\in\mathbb{R}^{1 \times d_1}, W\in\mathbb{R}^{ d_1 \times d_2}$, and $y=xW\in\mathbb{R}^{1 \times d_2}$, then $$P=\operatorname{diag}(x) \cdot W \cdot \operatorname{diag}(y)^{-1}, R(x)=P\cdot R(y), \mathbf{1}^{\top}R(x)=\mathbf{1}^{\top}R(y).$$
> > > where $\operatorname{diag}(x)$ is the diagonal matrix, $\operatorname{diag}(y)^{-1}$ is the inverse of the diagonal matrix and $\cdot$ denotes the matrix multiplication.
> > >
> > > For example, for the same case, x=[1,2,3], W=[[1,2,3],[4,5,6],[7,8,9]], y=[30,36,42], then
> > > $P=\begin{bmatrix}1&0&0\\\\0&2&0\\\\0&0&3\end{bmatrix} \cdot \begin{bmatrix}1&2&3\\\\4&5&6\\\\7&8&9\end{bmatrix} \cdot \begin{bmatrix} \frac{1}{30}&0&0\\\\0&\frac{1}{36}&0\\\\0&0&\frac{1}{42}\end{bmatrix}= \begin{bmatrix}0.03&0.05&0.07\\\\0.27&0.28&0.28\\\\0.7&0.67&0.65\end{bmatrix}.$
> > >
> > > Because $ \mathbf{1}^{\top} P=\mathbf{1}^{\top}$, $\mathbf{1}^{\top}R(x)=\mathbf{1}^{\top}PR(y)=\mathbf{1}^{\top}R(y)$
> > >
> > > >**Q2: Although Appendix A.14 provides some running time results, a scalability comparison against other baseline models is still expected.**
> > >
> > > **A2**: Thank you for pointing this out. We conduct experiments to compare the runtime of our method (MemExplainer) with the baseline method on six different datasets. The table below reports the average computation time for a single target event or node on each dataset. The average computation time can also be seen in this [figure](https://anonymous.4open.science/r/test-872C/running_time.png)
> > >
> > > **Table 1**: Comparison of average computation time (in seconds) across six datasets. The best results are highlighted in **bold**.
> > >
> > > | Method | Reddit | Wikipedia | UCI | tgbn-trade | tgbn-genre | tgbn-reddit |
> > > |:---|:---:|:---:|:---:|:---:|:---:|:---:|
> > > | MemExplainer | **0.3295** | **0.4201** | 0.2435 | **0.4969** | **0.3569** | **0.1899** |
> > > | TempMe | 3.3872 | 1.8458 | 3.9529 | 1.6717 | 3.0322 | 1.4540 |
> > > | TGNNExplainer | 2.3454 | 1.0608 | 0.9995 | 2.2108 | 4.6266 | 4.1078 |
> > > | GNNExplainer | 0.6325 | 0.5425 | **0.2111** | 6.1971 | 3.3138 | 3.0257 |
> > > | PGExplainer | 4.7306 | 1.0672 | 0.7021 | 3.9570 | 5.7821 | 2.0882 |
> > >
> > > From the results in Table 1, we observe that our method is **faster** than baseline methods, except for GNNExplainer on the UCI dataset. However, the runtime gap between our method and GNNExplainer on UCI is relatively small. One reason is that TempMe, TGNNExplainer, GNNExplainer, and PGExplainer all **require training** during explanation. In contrast, our method only requires direct computation. In addition, the memory backtracking tree in our method only needs to be constructed once. After that, it can be **reused for explaining different events or nodes**. By contrast, the baseline methods usually need to resample the neighborhood subgraph for each target event or node and rerun the explanation process. This leads to higher time cost.

---

### Official Review · Reviewer_by8R · 2026-03-11

**Soundness:** 3
**Presentation:** 3
**Significance:** 4
**Originality:** 4
**Overall Recommendation:** 5
**Confidence:** 4

**Summary:**

This paper addresses a gap in the explainability of Temporal Graph Networks (TGNs). Existing explanation methods for TGNs tend to fix the memory vector of the nodes, effectively ignoring the memory module that is responsible for updating and recording a node's historical interactions. To solve this, the authors introduce a framework, referred to as MemExplainer, which attributes model predictions to both recent spatial interactions and historical events. There are three main contributions. First, the development of a topology attribution tree to quantify the spatial influence of neighbors and current node memories; second, the creation of a memory backtracking tree that traces how historical events shaped those memories; third, an optimization objective formulated via KL divergence to select the most important explanatory events.

**Compliance With Llm Reviewing Policy:**

Affirmed.

**Final Justification:**

I would like to thank the authors for their detailed rebuttal and the clear effort they put into addressing my concerns. Their thorough explanations resolved my questions regarding the clarity and technical soundness of the methodology. Because the rebuttal successfully cleared up these points, it fully reinforces my initial assessment of the paper's strengths, and I maintain my prior recommendation.

**Key Questions For Authors:**

1-In your sensitivity analysis, you note that for datasets like tgbn-trade, fidelity decreases and then increases with depth because a larger search space complicates event identification. Have you considered any regularization techniques during the KL divergence optimization to better handle this enlarged search space?
2-How does the choice of the number of depth TL in the memory backtracking tree affect the explanation for networks with very long-range historical dependencies?

**Limitations:**

Given that this method is designed to track historical behavioral dependencies in networks, including human communication and social media, it might have distinct implications for surveillance, biased profiling, and privacy. Authors might acknowledge this potential negative societal impact.

**Strengths And Weaknesses:**

For soundness, I think submission is technically robust. The empirical validation is a significant strength, utilizing nine distinct datasets ranging from human actions to communication networks. However, I want to highlight a potential weakness which lies in the computational complexity of the memory backtracking tree that grows exponentially. While authors cut the depth to manage this, their sensitivity analysis reveals that deeper backtracking sometimes reduces fidelity due to the expanded search space, which might indicates a limitation in identifying truly long-term dependencies. While they acknowledged "The optimal depth varies across datasets due to their distinct characteristics.", it might be better to elaborate on these characteristics and how model handles these variations.
For presentation, the paper is well-written and good structured. I can highlight minor proofreading issues, such as bath 2 instead of batch 2. In addition to this, especially in some part of papers, I need to frequently jump to appendix, which breaks flow.
For significance, from a social science perspective, I believe the significance is substantial. Understanding why a dynamic tie forms, whether it is an international trade agreement or a communication link within an organization like Enron, is reliant on historical path dependencies. This work provides a tool that could allow researchers to better interpret the temporal evolution of human and political behaviors.
For originality, I think combining spatial topology attribution with temporal memory backtracking to achieve a mathematically conserved explanation is a creative and justified approach to a complex problem.

---

> ### Author Rebuttal · Authors · 2026-03-30
>
> Thank you for taking the time to review our paper!
>
> >**Q1: However, I want to highlight a potential weakness which lies in the computational complexity of the memory backtracking tree that grows exponentially. While authors cut the depth to manage this, their sensitivity analysis reveals that deeper backtracking sometimes reduces fidelity due to the expanded search space, which might indicates a limitation in identifying truly long-term dependencies.**
>
> **A1**: Thank you for this insightful comment. We agree that this is a limitation of our current framework.
>
> The memory backtracking tree is organized temporally: the root represents the contribution of the node’s memory vector at the prediction time, the first layer corresponds to the most recent events that directly updated this memory, and nodes closer to the leaves correspond to increasingly earlier historical events. As a result, increasing the depth allows the method to trace farther back in time and potentially capture longer-range dependencies, but it also introduces more candidate historical events, which makes the subsequent optimization problem more difficult.
>
> To handle this issue, one possible strategy is to apply **pruning-based approximations** during backtracking, for example by keeping only the nodes or branches with the largest relevance scores at each level. This can reduce the search space while preserving the most influential paths. In addition, when the number of candidate historical events remains very large (e.g., more than 10000), a practical alternative is to use a **Top-K strategy** to directly select the most relevant events, instead of solving the full optimization problem.
>
> >**Q2: Have you considered any regularization techniques during the KL divergence optimization to better handle this enlarged search space?**
>
> **A2**: Thank you for this insightful suggestion. In our KL-divergence-based optimization, adding a standard regularization term **is generally not effective** because the optimization is already performed under a constraint: the number of selected important historical events is fixed to be equal to the given number. Under this constraint, regularizers do not further reduce the feasible search space in a meaningful way.
>
> In practice, when the candidate set of historical events becomes large, a more effective strategy is to simplify the optimization itself. One option is to **relax the binary selection variables** from {0,1} to continuous values in [0,1], which makes the optimization significantly more efficient. Another practical strategy is to **solve the optimization only over the Top-K historical events**. These approaches help control the enlarged search space more directly than adding regularization.
>
> >**Q3: How does the choice of the number of depth TL in the memory backtracking tree affect the explanation for networks with very long-range historical dependencies?**
>
> **A3**: Thank you for this important question. As we mentioned, in the memory backtracking tree, the first layer corresponds to the most recent events that directly updated root node memory, and nodes closer to the leaves correspond to earlier historical events. The depth $T_L$  determines how far the tree traces back through past memory updates.
>
> The existence of long-range historical dependencies **does not necessarily** imply that **early events contribute strongly to the current memory state**. Since the node’s memory vector is updated recurrently over time, the influence of earlier events on the current memory state may progressively weaken as more recent updates are incorporated. Therefore, earlier historical events may receive **smaller importance scores** than those events at later time steps, even when long-range dependencies exist in the data.
>
> Thus, in practice, a moderate depth such as $T_L$ in [10,15] is often sufficient to capture the main explanatory events, while using a much larger depth mainly introduces weak and noisy early historical branches.
>
> >**Q4: I can highlight minor proofreading issues, such as bath 2 instead of batch 2. In addition to this, especially in some part of papers, I need to frequently jump to appendix, which breaks flow.**
>
> **A4**: Thank you for pointing this out, and we apologize for the proofreading mistakes. We also apologize for the disruption in reading flow caused by frequent references to the appendix. We will carefully revise the presentation to improve readability and make the manuscript easier to follow.
>
> >**Q5: It might have distinct implications for surveillance, biased profiling, and privacy. Authors might acknowledge this potential negative societal impact.**
>
> **A5**:  In our work, all experiments are conducted on publicly available datasets, and our goal is to improve the explanations of temporal graph models rather than to support surveillance or profiling. We will add a brief discussion in the revised revision to acknowledge these considerations and emphasize the need for responsible use.

---

> > ### Author Rebuttal · Reviewer_by8R · 2026-04-04
> >
> > Thank you to the authors for the detailed and thoughtful rebuttal. I appreciate your candidness regarding the limitations of the memory backtracking tree and your willingness to add the necessary societal impact discussions and presentation fixes.
> >
> > While your responses theoretically address my concerns, I have a follow-up question regarding the practical implementation of your proposed solutions:
> >
> > In A1 and A2, you propose several compelling strategies to manage the enlarged search space, such as pruning-based approximations, continuous relaxation of the binary variables to [0,1], and utilizing a Top-K strategy.
> >
> > Question: Will you actually implement and evaluate these specific strategies (e.g., continuous relaxation or Top-K) in the revised manuscript, or will these be proposed as future work? If they have been implemented, could you briefly share how much the continuous relaxation impacts the fidelity of the explanations compared to the exact constrained optimization? If they are strictly future work, the exponential complexity remains a practical bottleneck for the current iteration of MemExplainer, which should/can be explicitly emphasized in the limitations section.

---

> > > ### Author Response · Authors · 2026-04-04
> > >
> > > Thank you for the thoughtful follow-up question.
> > >
> > > We would like to clarify that the continuous relaxation or solving the optimization over the Top-K historical events are currently not implemented and are proposed as **future directions** for improving scalability. In the current implementation, we reduce the practical runtime by limiting the depth of the Memory Backtracking Tree. In addition, the memory backtracking tree in our method only needs to be constructed once. After that, **it can be reused for explaining different events or nodes**.  This property further improves computational efficiency in practical applications. The results shown in Figures 4, 5, and 7–12 are all obtained with the memory backtracking tree depth fixed at 5. Even under this constrained search setting, our method consistently achieves better performance than all baselines.
> > >
> > >  We also compare the runtime of our method with baseline approaches. The table below reports the average computation time for a single target event or node on each dataset. The average computation time can also be seen in this [figure](https://anonymous.4open.science/r/test-872C/running_time.png)
> > >
> > > **Table 1**: Comparison of average computation time (in seconds) across six datasets. The best results are highlighted in **bold**.
> > >
> > > | Method | Reddit | Wikipedia | UCI | tgbn-trade | tgbn-genre | tgbn-reddit |
> > > |:---|:---:|:---:|:---:|:---:|:---:|:---:|
> > > | MemExplainer | **0.3295** | **0.4201** | 0.2435 | **0.4969** | **0.3569** | **0.1899** |
> > > | TempMe | 3.3872 | 1.8458 | 3.9529 | 1.6717 | 3.0322 | 1.4540 |
> > > | TGNNExplainer | 2.3454 | 1.0608 | 0.9995 | 2.2108 | 4.6266 | 4.1078 |
> > > | GNNExplainer | 0.6325 | 0.5425 | **0.2111** | 6.1971 | 3.3138 | 3.0257 |
> > > | PGExplainer | 4.7306 | 1.0672 | 0.7021 | 3.9570 | 5.7821 | 2.0882 |
> > >
> > > From the results in Table 1, we observe that our method is **faster** than baseline methods, except for GNNExplainer on the UCI dataset. However, the runtime gap between our method and GNNExplainer on UCI is relatively small. One reason is that TempMe, TGNNExplainer, GNNExplainer, and PGExplainer all **require training** during explanation. In contrast, our method only requires direct computation. Besides, the baseline methods usually need to resample the neighborhood subgraph for each target event or node and rerun the explanation process. This leads to higher time cost.
> > >
> > > Overall, **we acknowledge that the exponential worst-case complexity remains a limitation in the very large temporal graphs**. We will make this point explicit in the limitations section.

---

### Official Review · Reviewer_xYRG · 2026-03-13

**Soundness:** 3
**Presentation:** 2
**Significance:** 2
**Originality:** 3
**Overall Recommendation:** 5
**Confidence:** 3

**Summary:**

The paper tackles a critical gap in the explainability of Temporal Graph Networks (TGNs) by introducing a dual-tree attribution framework. Recognizing that traditional mask-based perturbation methods are computationally prohibitive for unrolled memory chains, existing explainers typically freeze the node memory vector, effectively reducing TGNs to standard GNNs with static features. To overcome this, the authors abandon perturbation in favor of adapting Layer-wise Relevance Propagation (LRP). This gradient-based backward attribution allows the framework to seamlessly backpropagate through the GRU/RNN update gates, tracing the current prediction back to the originating historical events. The method ensures a strict conservation property where the assigned event contributions perfectly reconstruct the output logits, culminating in a faithful, optimization-based event selection.

**Compliance With Llm Reviewing Policy:**

Affirmed.

**Final Justification:**

I am raising my score. The authors fully addressed my concerns by providing a concrete numerical toy example for the LRP operations and a clear, chronological explanation of the memory backtracking tree.

**Key Questions For Authors:**

1. Can the authors provide a minimal, step-by-step walkthrough (e.g., a toy temporal graph with 3 nodes over 3 timestamps) demonstrating exactly how relevance flows backward through the GRU gates to be assigned to a specific historical event?

**Limitations:**

No. The authors have not adequately discussed the limitations or the potential negative societal impacts of their work. The current Impact Statement (Section 7) is entirely dismissive. I suggest adding a dedicated "Limitations and Broader Impacts" paragraph or section in the final version.

**Strengths And Weaknesses:**

Strengths:
1. The authors correctly identify a fundamental flaw in prior work: fixing the memory vector artificially truncates the temporal dimension and ignores the core dynamic mechanism of TGNs. Unboxing the memory module successfully addresses a major bottleneck in dynamic graph interpretability.
2. The mathematical formulation of the attribution and backtracking trees is solid. The adaptation of LRP to maintain the conservation property (i.e., the sum of event contributions explicitly equals the output logits) provides a robust, mathematically grounded foundation for the method.
3. The experimental setup is highly thorough, benchmarking the proposed method against multiple baselines across nine datasets and three distinct downstream tasks. The qualitative case studies (e.g., the kinematic chain visualizations in pose-based action classification) convincingly demonstrate the method's practical utility and alignment with underlying biomechanics.

Weaknesses:
1. The methodology is mathematically dense but lacks a step-by-step running toy example. This makes the complex LRP operations through GRU gates difficult to follow for a general audience.
2. The core mechanism of "memory backtracking" is presented too abstractly. The authors rely heavily on equations without providing enough intuitive, high-level explanations to support the mathematical derivations.

---

> ### Author Rebuttal · Authors · 2026-03-31
>
> Thank you for taking the time to review our paper!
>
> >**Q1: The methodology lacks a step-by-step running toy example.**
>
> **A1**: The GRU function is shown in 4.3.1. and a high-level overview of the decomposition is provided in this [figure](https://anonymous.4open.science/r/test-872C/GRU_decomposition.png). Following the GRU equations, we first decompose $R(s_u^{t})$ into $R(s_u^{t-})$ and $R(\tilde s_u^{t-})$. Then, $R(\tilde s_u^{t-})$ is further decomposed into $R(s_u^{t-})$ and  $R(\bar m_u^t)$ . Finally, $R(\bar m_u^t)$ is attributed to $R(s_u^{t-}), R(s_v^{t-}), R(t-t_k), R(x_{e_k}) $. We provide a toy running example. Supposing that $s_v^{t-}=[0.4,0.2]$, $s_u^{t-}=[0.1,0.3]$, $x_{e_k}=[0.5]$, and $t-t_k=[0.6]$. Then, $\bar m_u^t=[s_v^{t-}||s_u^{t-}||x_{e_k}|| t-t_k]=[0.4,0.2,0.1,0.3,0.5,0.6]$. The GRU parameters $W_r=\begin{bmatrix}
> 0.1&0\\\\0&0.1\\\\0.5&0\\\\0&0.5\\\\0.1&0\\\\0&0.1\end{bmatrix}$, $W_g=\begin{bmatrix}0.2&0\\\\0&0.1\\\\0.1&0\\\\0&0.1\\\\0.1&0 \\\\0& 0.2\end{bmatrix}$, $W_h=\begin{bmatrix}0.1&0\\\\0&0.1\\\\0.05&0\\\\0&0.05\\\\0.05&0\\\\0&0.05\end{bmatrix}$, $U_r=U_g=U_h=\begin{bmatrix}0.1&0\\\\0&0.1\end{bmatrix}$, $b_r=b_g=b_h=[0,0]$, and $R(s_u^t)=\begin{bmatrix}1&2\\\\3&4\end{bmatrix}$.
>
> **Forward propagation**: $r_u^t=[0.526,0.531]$, $g_u^t=[0.537,0.549]$, $\tilde s^t_u=[0.075,0.080]$, and $s^t_u=[0.086,0.179]$
>
> **Calculation process**:
>
> (1)$P_1=\operatorname{diag}(\frac{(1-g_u^t)\odot s_u^{t-}}{s_u^t})=\begin{bmatrix}0.53&0\\\\0&0.75\end{bmatrix}$, $P_2=\operatorname{diag}(\frac{g_u^t\odot s_u^{t-}}{\tilde s_u^t})=\begin{bmatrix}0.47&0\\\\0&0.25\end{bmatrix}$, and $\mathbf{1}^\top P_1 + \mathbf{1}^\top P_2 = [1,1]$. $R(s_u^{t-})
> =P_1 \cdot R(s_u^t)=\begin{bmatrix}0.53&1.06\\\\2.25&3\end{bmatrix}$, $R(\tilde s_u^{t-})
> =P_2 \cdot R(s_u^t)=\begin{bmatrix}0.47&0.94\\\\0.75&1\end{bmatrix}$, and $\mathbf{1}^\top R(s_u^t)=\mathbf{1}^\top R(s_u^{t-}) +\mathbf{1}^\top R(\tilde s_u^{t-})$
>
> (2) Let $A=\bar m_u^t W_h + r_u^t \odot (s_u^{t-} U_h)+b_h$, $P_3=\frac{\operatorname{broadcast}(\bar m_u^t)^\top)\odot W_h
> }{\operatorname{broadcast}(A)}=\begin{bmatrix}0.53&0\\\\0&0.25\\\\0.07&0\\\\0&0.18\\\\0.33&0\\\\0&0.37\end{bmatrix}$, $P_4=\frac{\operatorname{broadcast}(r_u^t \odot s_u^{t-})^\top \odot U_h}{\operatorname{broadcast}(A)}=\begin{bmatrix}0.07&0 \\\\0&0.2\end{bmatrix}$, and $\mathbf{1}^\top P_3 + \mathbf{1}^\top P_4 = [1,1]$. $R(\bar m_u^t)
> =P_3 \cdot R(\tilde s_u^t)=\begin{bmatrix}0.25&0.5\\\\0.18&0.25\\\\0.03&0.06\\\\0.14&0.19\\\\0.16&0.31\\\\0.28&0.37
> \end{bmatrix}$, $R(s_u^{t-})=P_4 \cdot R(\tilde s_u^t)=\begin{bmatrix}0.03&0.06\\\\0.15&0.2\end{bmatrix}$, $\mathbf{1}^\top R(\tilde s_u^t)=\mathbf{1}^\top R(\bar m_u^t)+\mathbf{1}^\top R(s_u^{t-})$.
>
> (3) $R(\bar{m}_u^t)(i,j)$ represents the contribution of the i-th element in $\bar m_u^t$ to the j-th column element of logits, thus, $$R(s_v^{t-})=\begin{bmatrix}0.25&0.5\\\\0.18&0.25\end{bmatrix}, R(s_u^{t-})=\begin{bmatrix}0.03&0.06\\\\0.14&0.19\end{bmatrix}, R(x_e)=[0.16,0.31], R(t-t_k)=[0.28,0.37].$$ The event contribution is $\mathbf{1}^\top R(x_e) + \mathbf{1}^\top R(t-t_k)$=1.12. The total contribution $R(s_u^{t-})=\begin{bmatrix}0.59&1.18\\\\2.54&3.39\end{bmatrix}
> $ and  $\mathbf{1}^\top R(s_u^{t-}) +\mathbf{1}^\top R(s_v^{t-}) + \mathbf{1}^\top R(x_e) + \mathbf{1}^\top R(t-t_k)=\mathbf{1}^\top R(s_u^{t})$
>
> >**Q2:The authors do not provide high-level explanations. Can the authors provide a step-by-step walkthrough?**
>
> **A2**: The memory backtracking tree is organized chronologically: the root corresponds to the target node’s memory at the prediction time, the first layer contains the most recent events that directly updated this memory, and deeper layers trace earlier historical events.
>
> We illustrate this with a toy example in this [figure](https://anonymous.4open.science/r/test-872C/toy_example.png). As shown in Fig. (b), events are processed in batches, and memory updates are carried forward to subsequent batches. Let $s_u^t$ denote the memory vector of node $u$ at time t. Fig. (c) shows the memory update process of nodes 2 and 3. For node 3, its memory is updated from $s_3^{t_0}$ after receiving the message from event $e_1$, remains unchanged at $t_2$, is updated again by event $e_3$ to $s_3^{t_3}$, and stays unchanged at $t_4$. Node 2 follows a similar process.
>
> Fig. (d) shows the memory backtracking tree for node 3. Starting from the $R(s_3^{t_4})$, we first trace it back to the most recent event $e_3$, and we can obtain the tree in Fig. (d)(1). The $R(s_3^{t_2}), R(s_2^{t_2})$, and $R(e_3)$ can be obtained  as described in the previous question (Q1). Then, $R(s_3^{t_2})$ is further traced back to event $e_1$, as shown in Fig. (d)(2), and $R(s_2^{t_2})$ is traced back to event $e_2$, as shown in Fig. (d)(3). By recursively expanding the tree in this way, we obtain the contributions of all relevant historical events, satisfying $\mathbf 1^\top R(e_1)+\mathbf 1^\top R(e_2)+\mathbf 1^\top R(e_3) =\mathbf 1^\top R(s_3^{t_4})$.

---

> > ### Author Rebuttal · Reviewer_xYRG · 2026-04-03
> >
> > My questions are resolved, and I am willing to raise my rating.

---

### Decision · Program_Chairs · 2026-04-30

**Decision:**

Accept (spotlight)

**Comment:**

The article addresses a real gap in temporal graph explainability by opening the memory module, which is exactly the part that prior methods tend to freeze or ignore.

The reviewers find the technical idea clear and meaningful: combining topology attribution with memory backtracking so that current predictions can be traced to concrete historical events. The evaluation is broad across datasets and task types, and the rebuttal materially improved the paper by clarifying the derivations, notation, toy examples, and runtime concerns.

The remaining weaknesses are mostly about presentation (abstract concepts and running toy examples) and scalability limits, not about the core contribution. Hence, I am comfortable supporting acceptance.